# Changes in glacier dynamics in the northern Antarctic Peninsula since 1985

Thorsten Seehaus[1], Alison J. Cook[2], Aline B. Silva[3], Matthias Braun[1]

[1] Institute of Geography, Friedrich-Alexander-University Erlangen-Nuremberg, Wetterkreuz 15, 91058 Erlangen, Germany

5    [2] Department of Geography, Durham University, South Road, Durham DH1 3LE, United Kingdom

[3] Laboratório de Monitoramento da Criosfera, Universidade Federal do Rio Grande, Av. Itália, km 8, 96203-900, Rio Grande - RS Brazil

Correspondence to:  Thorsten Seehaus (*thorsten.seehaus@fau.de*)

**Abstract.** The climatic conditions along the northern Antarctic Peninsula have shown significant changes within the last 50
10   years. Here we present a comprehensive analysis of temporally and spatially detailed observations of the changes in ice dynamics along both the east and west coastlines of the northern Antarctic Peninsula. Temporal evolutions of glacier area (1985-2015) and ice surface velocity (1992-2014) are derived from a broad multi-mission remote sensing database for 74 glacier basins on the northern Antarctic Peninsula (<65° S along the west coast and north of the Seal Nunataks on the east coast). A recession of the glaciers by 238.81 km² is found for the period 1985-2015, of which the glaciers affected by ice
shelf disintegration showed the largest retreat by 208.59 km². Glaciers on the east coast north of the former Prince Gustav Ice Shelf extent in 1986 receded by only 21.07 km² (1985-2015) and decelerated by about 58% on average (1992-2014). A dramatic acceleration after ice shelf disintegration with a subsequent deceleration is observed at most former ice shelf tributaries on the east coast, combined with a significant frontal retreat. In 2014, the flow speed of the former ice shelf tributaries was 26% higher than before 1996. Along the west coast the average flow speeds of the glaciers increased by 41%.
However, the glaciers on the western Antarctic Peninsula revealed a strong spatial variability of the changes in ice dynamics. By applying a hierarchical cluster analysis, we show that this is associated with the geometric parameters of the individual glacier basins (Hypsometric Indexes, maximum surface elevation of the basin, flux gate to catchment size ratio). The heterogeneous spatial pattern of ice dynamic evolutions at the northern Antarctic Peninsula shows that temporally and spatially detailed observations as well as further monitoring are necessary to fully understand glacier change in regions with
such strong topographic and climatic variances.

## 1 Introduction

During the last century, the northern Antarctic Peninsula (AP) and its outlying islands have undergone significant warming (Turner et al., 2005), leading to substantial glaciological changes. Skvarca et al. (1998) reported a significant increase in surface air temperatures at the north-eastern AP in the period 1960-1997 and correlated it with the recession of the Larsen
and Prince Gustav Ice shelves (Fig. 1) and the observed retreat of tidewater glaciers on James Ross Island in the period

1975-1995 (Skvarca et al., 1995). However, a recent cooling trend on the AP was revealed by Oliva et al. (2017) and Turner et al. (2016) since the late 1990s. Shepherd et al. (2012) compiled a comprehensive glacier mass balance database of the polar ice sheets. The authors estimated a mass loss on the whole AP (<73° S) of -36±10 Gt a$^{-1}$ for the period 2005-2010, which corresponds to 35% of the total mass loss of Antarctica. A projection of sea level rise contribution by the AP ice sheet amounts to 7-16 mm sea-level equivalent by 2100 and 10-25 mm by 2200 (Barrand et al., 2013a). However, along the western AP and on the higher elevation areas an increase in snow accumulation in the late 20th century was derived from ice cores (e.g. at Palmer Land, 73.59° S, 70.36° W, Thomas et al., 2008; Detroit Plateau, 64.08°S, 59.68° W, Potocki et al., 2011; at Bruce Plateau, 66.03°S, 64.07°W, Goodwind, 2013) and climate models (e.g. Dee et al., 2011), whereas Van Wessem et al. (2016) obtained insignificant trends in precipitation.

Numerous ice shelves along the AP have retreated widely (e.g. Müller, Wilkins, Wordie) or disintegrated in recent decades (e.g. Larsen A in 1995, Larsen B in 2002) (Braun and Humbert, 2009; Cook and Vaughan, 2010; Doake and Vaughan, 1991; Rack et al., 1998; Rack and Rott, 2003; Wendt et al., 2010). As a consequence to the reduced buttressing, former tributary glaciers showed increased ice discharge and frontal retreat (e.g. De Angelis and Skvarca, 2003; Rack and Rott, 2004; Rignot et al., 2004; Seehaus et al., 2015; Wendt et al., 2010). For the northern AP (<66° S), a mass loss rate of -24.9±7.8 Gt a$^{-1}$ was reported by Scambos et al. (2014) for the period 2003-2008, indicating that major ice mass depletion happened at the northern AP, especially along the eastern side where numerous glaciers have been affected by ice shelf collapses. Seehaus et al. (2015, 2016) quantified the ice loss of former ice shelf tributaries. Mass loss rates of -2.14±0.21 Gt a$^{-1}$ (1995-2014) and -1.16±0.16 Gt a$^{-1}$ (1993-2014) were found at Dinsmoor-Bombardier-Edgeworth Glacier System and Sjögren-Inlet glaciers, respectively. Glaciers that were not terminating in an ice shelf also showed considerable changes. Cook et al. (2005, 2014) have analyzed the variations of tidewater glacier fronts since the 1940s. The authors reported that 90% of the observed glaciers retreated, which they partly attributed to atmospheric warming. A more recent study revealed a mid-ocean warming along the southwestern coast of the AP, forcing the glacier retreat in this region (Cook et al., 2016). Pritchard and Vaughan (2007) observed an acceleration of ice flow by ~12% along the west coast of the AP (1995-2005) and linked it to frontal retreat and dynamic thinning of the tidewater glaciers. Observations by Kunz et al. (2012) support this supposition. They analyzed surface elevation changes of 12 glaciers on the western AP based on stereoscopic digital elevation models (DEM) over the period 1947-2010. Frontal surface lowering was found at all glaciers. Glacier-wide surface lowering was observed by various author groups (e.g. Berthier et al., 2012; Rott et al., 2014; Scambos et al., 2014) at former ice shelf tributaries along the north-eastern AP. The collected observations suggest that the ice masses on the AP are contributing to sea level rise and show that glaciers' response to climate change on the AP is not homogeneous and that more detailed knowledge of various aspects on the glacier changes are required. Previous studies often cover a specific period or area, or focus on one particular aspect of glacier change. By now, the availability of remote sensing data time series data and other datasets in this region facilitates the comprehensive analysis of glacier change. Therefore, we study the changes in glacier extent in

combination with detailed investigations on ice dynamics as well as other derived geometrical attributes of glaciers on the northern AP (<65° S along the west coast and north of the Seal Nunataks on the east coast, Fig. 1b colored polygons) between 1985 and 2015. We analyze various multi-mission remote sensing datasets in order to obtain methodologically consistent and temporally detailed time series of ice dynamic changes of 74 glacier basins. The observations are individually discussed for the subregions, considering the different atmospheric, glaciological and oceanic conditions and changes.

## 2 Study site

The AP is the northern-most region of Antarctica and stretches from 63-75°S (Huber et al., 2017). It covers only 3% of the entire continent in area, but receives 13% of the total mass input (Van Lipzig et al., 2002, 2004). The AP's mountain chain (typically 1500-2000 m high) acts as an orographic barrier for the circumpolar westerly air streams leading to very high precipitation values on the west coast and on the plateau region of up to 5000 mm we $a^{-1}$, as well as frequent foehn type wind occurrences on the east coast (Cape et al., 2015, Marshall et al., 2006, Van Wessem et al. 2016). The foehn events are characterized by strong winds and high air temperatures. Consequently, the climatic mass balance ($b_{clim}$) shows a strong gradient across the mountain chain (Turner, 2002; Van Wessem et al., 2016). Aside from those that are ice shelf tributaries, almost all glaciers on the AP are marine terminating, and the majority of the glacier catchments extend up to the high elevation plateau regions (north to south: Laclavère, Louis Philippe, Detroit, Herbert, Foster, Forbidden, Bruce, Avery, Hemimont, Dyer). Typically the AP plateau is separated from the outlet glaciers by escarpments and ice-falls. Glaciers on the west coast drain into the Bellingshausen Sea and on the east coast into the Weddell Sea. Since the 1980s, the ice shelves along the east coast have substantially recessed and disintegrated (Larsen Inlet in 1987-89, Prince Gustav and Larsen A in 1995 and Larsen B in 2002) (Cook and Vaughan, 2010; Rott et al., 1996; Skvarca et al., 1999), which Scambos et al. (2003) attributed to higher summer air temperatures and surface melt. A more recent study by Holland et al. (2015) discovered that significant thinning of the Larsen C Ice Shelf is caused by basal melting and that ungrounding from an ice rise and frontal recession could trigger its collapse. The northern AP has a maritime climate and is the only region of Antarctica that frequently experiences widespread surface melt (Barrand et al., 2013b; Rau and Braun, 2002).

Our study site stretches approximately 330 km from the northern tip of the AP mainland southwards to Drygalski Glacier on the east coast and Grubb Glacier on the west coast (Fig. 1). This facilitates the analyses of the temporal evolution (~20 years) of the response of tributary glaciers to ice shelf disintegration at the former Larsen A and Prince Gustav ice shelves on the east coast, the investigation of glaciers north of the former Prince Gustav Ice Shelf, where no information on change in ice flow is currently available, and the comparison with temporal variations in ice dynamics along the west coast at the same latitude. The study site covers ~11,000 km² (~11% of the whole AP including islands, Cook et al., 2014; Huber et al., 2017) with elevations stretching from sea level up to 2220 m. The glacier basin delineations are based on the Antarctic Digital Database ADD 6.0 (Cook et al., 2014). Glacier names are taken from the Global Land Ice Measurements from Space

(GLIMS) project database. The local GLIMS glacier IDs (e.g. TPE62, LAB2) are used for unnamed glaciers and further missing glacier basin names are replaced with the ADD 6.0 glacier IDs. Neighboring basins with coalescing ice flow at the termini are merged (many are already merged in the ADD 6.0), as the delineation of the individual glacier sections is not always possible and the width can vary temporally (due to changes in mass flux of the individual glaciers). In these cases, the names of the glaciers are also merged (e.g. Sikorsky-Breguet-Gregory – SBG, see Table 1 for abbreviations of glacier names). Due to the sparse data coverage (fewer than three good quality velocity measurements), no time series analysis of the glaciers at the northern tip of the AP or at some capes and peninsulas (e.g. Sobral Peninsula, Cape Longing) is possible. Therefore, the northern-most analyzed catchments are Broad-Valley Glacier on the east coast and TPE8 Glacier on the west coast, resulting in 74 studied glacier basins. Furthermore, the study site is divided into three sectors, taking into account the different climatic settings and drainage orientation as well as former ice shelf extent: sector "West" - glaciers on the west coast, draining into the Bransfield and Gerlache Strait; sector "East" – glaciers on the east coast, draining into the Prince Gustav Channel; and sector "East-Ice-Shelf" – glaciers on the east coast, that were former tributaries to the Larsen A, Larsen Inlet and Prince Gustav Ice Shelf.

## 3 Data & Methods

A large number of various remote sensing datasets are analyzed in order to obtain temporally and spatially detailed information on changes in ice dynamics in the study area. Glacier area changes are derived from satellite and aerial imagery. Repeat-pass Synthetic Aperture Radar (SAR) satellite acquisitions are used to compute surface velocity fields in order to obtain information on changes in glacier flow speed. Auxiliary data from sources such as a digital elevation model and glacier inventory are included in the further analyses and discussion of the results.

### 3.1 Area changes

Changes in glacier area are derived by differencing glacier outlines from various epochs. All observed glaciers are tidewater glaciers and only area changes along the calving front were considered. Information on the positions of the glacier fronts are taken from Cook et al. (2014), and are available for the whole AP in the ADD 6.0 (1945-2010). This coastal-change inventory is based on manually digitized ice front positions using imagery from various satellites (e.g. Landsat, ERS) and aerial photo campaigns. This dataset is updated (up to 2015) and gaps are filled by manual mapping of the ice front positions based on SAR and optical satellite images. Consistent with Cook et al. (2014), the ice-front positions are assigned to 5-year intervals in order to analyze temporal trends in glacier area changes in the period 1985-2015. Before 1985, only sparse information on ice front positions for the whole study site is available, and the coverage by SAR data for analyzing glacier flow starts in 1992. Additionally, the analysis of the area changes for the Larsen A and Prince Gustav Ice Shelf tributaries is limited to the period 1995-2015, as the ice shelves disintegrated in 1995.

The uncertainties of the glacier change measurements strongly depend on the specifications of the imagery used (e.g. spatial resolution, geodetic accuracies) as well as the methods used. To each record in the coastal-change inventory from the ADD 6.0, a reliability rating is assigned according to Ferrigno et al. (2006). The rating ranges from 1 to 5 (reliability within 60 m to 1 km) and takes into account errors due to manual digitization and interpretation (see Ferrigno et al., 2006 for a detailed description). This approach is also applied on the updated ice-front positions. Nearly all mapped ice fronts in the area studied have a good reliability rating of 1 (76%) and 2 (21%). Only a few glacier fronts (3%) have a rating of 3. No ice fronts with reliability ratings of 4 and 5 are mapped in the study area.

## 3.2 Surface velocities

Surface velocity maps are derived from repeat-pass Synthetic Aperture Radar (SAR) acquisitions. SAR image time series of the satellite missions ERS-1/2, Envisat, RadarSAT-1, ALOS, TerraSAR-X (TSX) and TanDEM-X (TDX) are analyzed, covering the period 1992-2014. Specifications of the SAR sensors are listed in Table 2. The large number of SAR images was provided by the German Aerospace Center (DLR), the European Space Agency (ESA) and the Alaska Satellite Facility (ASF). To obtain displacement fields for the glaciers, the widely used and well approved intensity offset tracking method is applied on co-registered single look complex SAR image pairs (Strozzi et al., 2002). In order to improve the co-registration of the image pairs, we mask out fast moving and unstable regions such as outlet glaciers and the sea during the co-registration processes. Furthermore, single SAR image tiles acquired during the same satellite flyover are concatenated in the along-track direction. This helps to further improve the co-registration in coastal regions (by including more stable areas in the co-registration process) but also simplifies the analysis of the final results as no mosaicking of the results is needed. Image pairs with low quality co-registration are filtered out. A moving window technique (step-size see Table 2) is used by the intensity offset tracking method to compute the cross-correlation function of each image patch and to derive its azimuth and slant range displacement. The resolution of the obtained displacement fields depends on the combination of the step-size and the resolution of the images in slant-range geometry. A resolution of the velocity fields of ~50 m for the high resolution sensors TSX, TDX and ~100 m for all other sensors was targeted. Less reliable offset measurements are filtered out by means of the signal-to-noise ratio of the normalized cross-correlation function. Moreover, we apply an additional filter algorithm based on a comparison of the magnitude and alignment of the displacement vector relative to its surrounding offset measurements. This technique removes more than 90% of incorrect measurements (Burgess et al., 2012). Finally, the displacement fields are transferred from slant range into ground range geometry, taking into account the contortion caused by the topography (topographic effects on the local SAR incidence angle). The results are then geocoded, orthorectified, resampled and converted into velocity fields (with 100 m pixel spacing for all sensors) by means of the time span between the SAR acquisitions. The mean date of the consecutive SAR acquisitions is assigned to each velocity field. The ASTER Global Digital Elevation Model of the Antarctic Peninsula (AP-DEM, Cook et al., 2012) is used as elevation reference. It is currently the best available digital elevation model of the Antarctic Peninsula. It has a mean elevation bias of -4 m (±25 m

RMSE) from ICESat data and horizontal accuracy better than 2 pixels. Since the accuracy varies regionally, Huber et al. 2017 estimated the uncertainty to be ±50 m for the AP-DEM, based on their experiences with other DEMs. Velocity data is analyzed close to the calving front (see further down) where the slope of the glaciers at the AP is typically quite low. Thus, the impact of the DEM accuracy on the velocity fields is insignificant (see Seehaus et al., 2015 supplemental material).

Depending on the displacement rate and resolution of the SAR sensor, the tracking window size needs to be adapted (de Lange et al. 2007). For the fast flowing central glacier sections, larger window sizes are needed since large displacements cannot be tracked by using small correlation patches. Small tracking window sizes are suitable for the slow moving lateral sections of the outlet glaciers. Wide parts of large tracking patches cover the stable area next to the glacier, which biases the tracking results towards lower velocities. Consequently, we compute surface velocity fields of the same image pairs for

different correlation patch sizes in order to get the best spatial coverage. Table 2 shows the different tracking window sizes for each sensor. The results of each image pair are stacked by starting with the results of smallest tracking window size and filling the gaps with the results of the next biggest tracking window size.

The accuracy of the velocity measurements strongly depends on the coregistration quality and the intensity offset tracking algorithm settings. The mismatch of the coregistration $\sigma_v^C$ is quantified by measuring the displacement on stable reference

areas close to the coast line, such as rock outcrops and nunataks. Based on the Bedmap2 (Fretwell et al., 2013) and ADD 6.0 rock outcrop masks, reference areas are defined and the median displacements magnitude of each velocity field is measured at these areas. The uncertainty of the tracking process $\sigma_v^T$ is estimated according to McNabb et al. (2012) and Seehaus et al. (2015) depending on accuracy of the tracking algorithm $C$, image resolution $dx$, oversampling factor $z$, time interval $dt$.

$$\sigma_v^T = \frac{Cdx}{zdt} \qquad (1)$$

The accuracy of the tracking algorithm is estimated to be 0.2 pixels and an oversampling factor $z=2$ is applied to tracking patches in order to improve the accuracy of the tracking process. Both independent error estimates are quadratically summed to compute the uncertainties of the individual velocity fields $\sigma_v$.

$$\sigma_v = \sqrt{(\sigma_v^T)^2 + (\sigma_v^C)^2} \qquad (2)$$

Two approaches to measure and analyze the temporal changes in ice flow of the studied glacier are evaluated (see also

Section S1 in the supplement).

First approach: An across glacier profile is defined (red lines in Fig. 1) close to the terminus of each basin, considering the maximum retreat state of the ice front in the observation period. The changes in the ice flow of the individual glaciers are analyzed by measuring the surface velocities along the profiles. In order to reduce the number of data gaps along the profile due to pixel size data voids in the velocity fields, the velocity data is extracted within a buffer zone of 200 m around the

profiles. The results are visually inspected in order to remove unreliable measurements, based on the magnitude and direction of ice flow along the profiles. Datasets with partial profile coverage, large data gaps or large-scale tracking errors are rejected. The resulting profile coverage by velocity measurements is in average 97% and data coverage of more than 93% is obtained for 90% of all extracted profiles. To minimize the impact of potential outliers (still remaining tracking errors), median velocities along the profiles are calculated and their temporal developments are plotted for each basin (Fig. S1-S74 in the supplement).

Second approach: The velocity values are picked at the location of maximum ice thickness at the across glacier profiles (taken from the first approach). Ice thickness is obtained from the ice thickness reconstruction of the AP by Huss and Farinotti (2014). By means of visual inspection of the velocity profiles obtained by the first approach, outliers in the measurements using the second approach are manually filtered out and the resulting evolution of the flow speeds of each glacier are plotted (Fig. S75-S148 in the supplement).

The glaciers are manually classified in six categories according to the temporal evolution of the ice flow speeds (see Table 3), since automatic classification attempts did not achieve satisfying results. Only glaciers with three or more observations and an observation period of more than 10 years are considered in the categorization, resulting in 74 categorized glacier basins (colored polygons in Fig. 1b). The GAMMA Remote Sensing software is used for processing of the SAR data.

### 3.3 Catchment geometries and settings

Glacier velocities and area change measurements provide information on the ice dynamics of the individual glaciers. To facilitate a better and comprehensive interpretation of these observations, additional attributes regarding the different geometries and settings of the glaciers are derived. In addition to glacier attributes derived by Huber et al. (2017), we calculated the Hypsometric Index and the ratio of the flux gate cross section divided by the glacier catchment area.

Mass input strongly affects the dynamics of a glacier. The climatic mass balance at the northern AP shows a strong spatial variability, with very high accumulation rates along the west coast (3769 mm we a$^{-1}$ on average in sector "West", 1992-2014, RACMO2.3), significantly lower values on the east coast (1119 mm we a$^{-1}$ on average in sector "East", 1992-2014, RACMO2.3) and an increase towards higher altitudes along both coast lines (Turner, 2002; Van Wessem et al. 2016). Consequently, the mass input depends on the basin orientation (east coast or west coast), elevation range and the hypsometry. For each glacier basin a Hypsometric Index (*HI*), defined by Jiskoot et al. (2009), is calculated by means of surface elevations from the AP-DEM. Based on this index the glaciers are grouped into the five categories according to Jiskoot et al. (2009), ranging from very top-heavy to very bottom heavy (Table 4). Moreover, the maximum elevations of the individual glacier catchments are derived from the AP-DEM, which represents the altitude range of the catchment, since all observed glaciers are marine terminating.

In order to characterize the catchment shape, the ratios (*FA*) of the flux gate cross sections divided by the glacier catchment areas are calculated. The flux gates are defined along the profiles used for the glacier flow analysis (Section 3.2). Lower values of *FA* indicate a channelized outflow (narrowing towards the glacier front), whereas higher *FA* ratios imply a broadening of the glacier towards the calving front. Ice thickness at the flux gates is taken from the AP Bedmap dataset from Huss and Farinotti (2014).

## 3.4 Cluster analysis

The glaciers in the sector "West" (Fig. 1, red shaded area) show a heterogeneous spatial pattern of ice dynamics as compared to the other sectors changes (Section 4.1, 4.2). In order to analyze the influence of the glacier geometries on the glaciological changes and to find similarities, a cluster analysis is carried out in sector "West". This is a proven method to classify glaciers based on a set of variables (Lai and Huang, 1989; Sagredo and Lowell, 2012). Variables of the glacier dynamics used are the derived area changes (in percent) and velocity changes (ratings of the categories, Table 3). Glaciers categorized as "stable" showed a temporal variability in flow speeds of less than 0.25 m d$^{-1}$. Therefore, we used the same rating for the velocity change categories "stable" and "fluctuating" to perform the cluster analysis. The glacier geometry parameters used are the Hypsometric Indexes *HI*, maximum surface elevation $h_{max}$ of the basin and the flux gate to catchment size ratio *FA*. The variables are standardized in the traditional way of calculating their standard scores (also known as z-scores or normal scores). It is done by subtracting the variables mean value and dividing by its standard deviation (Miligan and Cooper, 1988). Afterwards a dissimilarity matrix is calculated using the Euclidean distances between the observations (Deza and Deza, 2009). A hierarchical cluster analysis (Kaufman and Rousseeuw, 1990) is applied on the dissimilarities using Ward's minimum variance method (Ward, 1963). At the start, the most similar glaciers (samples) are grouped. The resulting clusters are iteratively joined based on their similarities until only one cluster is left, resulting in a dendrogram (see Section 4.4). The distances between the clusters are updated in each iteration step by applying the Lance-Williams algorithms (Lance and Williams, 1967).

## 4 Results

### 4.1 Area changes

Area changes relative to the measurements in the epoch 1985-1989 (1995-2000 for the former Larsen A and Prince Gustav Ice Shelf tributaries, see Section 5.2) of the observed glaciers are plotted in Fig. S1-S74 (supplement). The glaciers are classified in three groups based on the latest area change measurements, which are illustrated in Fig. 2: retreat (Fig. 2a, b, c, f) – loss of glacier area by frontal retreat; stable (Fig. 2e) – no significant area changes (within the error bars); advance (Fig. 2d) – gain of glacier area by frontal advance. In Fig. 3 the spatial distribution of the area change classification is illustrated. All glaciers along the east coast, including the former ice shelf tributaries, retreated, whereas along the west coast, numerous

glaciers show stable ice front positions and some glaciers even advanced. In total, 238.81 km² of glacier area was lost in the survey area in the period 1985-2015, which corresponds to a relative loss of 2.2%. All sectors show glacier area loss (Table 5), of which the area loss by 5.7% (208.59 km²) at sector "East-Ice-Shelves" clearly dominates. The glaciers in sector "West" and "East" recessed by 0.2% (9.14 km²) and 1.4% (21.07 km²), respectively. The temporal trends of total glacier area and area loss of all observed glaciers and of each sector are presented in Fig. 4. Catchment areas and changes between 1985 and 2015 of the individual basins are listed in Table S1 (supplement) and relative changes are illustrated in Fig. 5.

## 4.2 Surface velocities

A total of 282 stacked and filtered velocity fields are derived from the SAR acquisitions covering the period from 25th December, 1992 until 16th December, 2014. Figure S157-S160 (supplement) show exemplary velocity fields of the studied area obtained for ERS, Envisat, ALOS and TSX/TDX data. The average total uncertainty of the velocity fields amounts to $0.08 \pm 0.07$ m d$^{-1}$ and the values for each SAR sensor are provided in Table 2. In Table S3 (supplement) the error estimates of each velocity field are listed. The mean sample count to estimate the coregistration quality is 11717 and the average mismatch amounts to 0.07 m d$^{-1}$. The error caused by the tracking algorithm strongly varies depending on the source of the SAR data (sensor). A mean value of 0.05 m d$^{-1}$ is found. ERS image pairs with time intervals of one day have very large estimated tracking uncertainties, biased by the very short temporal baselines. Therefore, only the errors caused by the mismatch of the coregistration are considered in the total error computations of the seven ERS tracking results with one day temporal baselines.

All measured velocity profiles of the 74 observed glaciers are visually inspected and in total 2256 profile measurements (first approach) and 2736 point measurements (second approach) passed the quality check. The shortest observation period is 14.83 years at DBC31 Glacier, the average number of velocity measurements per glacier is 30.5 and 37.0 and the average observation period is 19.25 years ($\sigma$ = 2.06 years) and 19.21 years ($\sigma$ = 1.96 years) for the first and second measuring approach, respectively. Figure 2 shows by example the temporal evolution of the ice flow (using the first approach) for each velocity change category (see Table 3) and Fig. S149-S156 (supplement) show surface velocity profiles across the terminus for the same glaciers as well as for the small glacier catchments DGC14 and TPE61. For small and narrow glaciers, the capturing of the flow velocity gradients in the margins is still limited mainly by the sensor resolution, even applying different tracking window sizes (see Section 3.2).

The temporal evolution of the surface velocities at the termini of each glacier are depicted in the supplement (Fig. S1-S74 for the first approach, Fig. S75-148 for the second approach) and the related categories are listed in Table S1 and S2 (supplement).

For both velocity measuring approaches and each glacier, the flow velocities in the first $v_S$ and last year $v_E$ of the observation period as well as the absolute and relative change $dv$ is presented in Table S1 and S2 (supplement). The mean values of $v_S$, $v_E$ and $dv$ of all analyzed glaciers and for each sector are listed in Table 5. On average the ice flow in the whole studied area increased by 0.061 m/d (13%) and 0.071 m/d (7%) for the first and second approach respectively, but the average changes of the individual sectors are more pronounced. Along the west coast an average acceleration by 41% (0.177 m/d) and 44% (0.369 m/d) occurred and the former ice shelf tributaries on the east coast accelerated by 26% (0.118 m/d) and 41% (0.312 m/d) for both approaches respectively. In the sector "East" the glaciers decelerated resulting in a mean velocity change of -58% (-0.423 m/d) and -69% (-1.272 m/d) for the first and second approach respectively. The presented average flow speed change values are based on the observed changes of all glaciers in the respective sector (Table S1), ignoring the different size of the individual glaciers.

Detailed results and differences of both approaches to measure the glacier velocities are presented and discussed in the supplement Section S1. Based on this discussion, we decided to favor the first approach and its results are used for the subsequent analysis.

The spatial distribution of the categories is illustrated in Fig. 3. At nearly all glaciers in sector "East-Ice-Shelf" a peak in ice velocities is observed. In the sector "East", most glaciers showed a decrease in flow velocities in the observation period. The glaciers on the west coast show a more irregular distribution than along the east coast, but a local clustering of accelerating glaciers can be observed at Wilhelmina Bay. In order to analyze the quality of obtained velocity change signal, the ratio of the maximum measured velocity difference (maximum velocity minus minimum velocity) divided by the average error of the velocity measurements is calculated for each glacier. An average signal to noise ratio of 14.6 is found. At three glaciers (DGC14, DGC22 and Orel) a signal to noise ratio of less than 2 is observed. These glaciers are characterized as "stable", which justifies the low signal to noise ratio.

### 4.3 Catchment geometries and settings

The spatial distribution of Hypsometric Indexes and categories of the glacier basins is presented in Fig. 3 and the values are listed in Table S1 (supplement). The HI values range between -4.6 and 9.1 (mean: 0.88, $\sigma$: 2.10). No clear spatial distribution pattern can be identified, reflecting the heterogeneous topography of the AP. The maximum elevation of the catchments and the $FA$ factors are also listed in Table S1 (supplement).

### 4.4 Cluster analysis

The resulting dendrogram of the hierarchical cluster analysis is plotted in Fig. 6. Four groups are distinguished. The boxplots of each input variable are generated based on this grouping and are shown in Fig. 7. The characteristics of the groups are discussed in Section 5.3.

## 5 Discussion

Most of the observed glaciers (62%) retreated and only 8% advanced in the study period. These findings are comparable to the results of Cook et al. (2005, 2014, 2016). Only glaciers along the west coast showed stable or advancing calving fronts and all glaciers on the east coast receded since 1985. This heterogeneous area change pattern was also observed by Davies et al. (2012) on western Trinity Peninsula. Most significant retreat occurred in the sector "East-Ice-Shelf". In the period 1985-1995, the Larsen Inlet tributaries (APPE-glaciers) lost 45.0 km² of ice. After the disintegration of Prince Gustav and Larsen A Ice Shelf, the tributaries rapidly retreated in the period 1995-2005. The recession slowed down in the latest observation interval (2005-2010). This trend is comparable to detailed observations by Seehaus et al. (2015, 2016) at individual glaciers (DBE glaciers and Sjögren-Inlet glaciers). At sector "East" the highest area-loss is found in the earliest observation interval (1985-1990). Davies et al. (2012) also reported higher retreat rates for most of the glaciers in this sector in the period 1988-2001 than in the period 2001-2009. However, another significant recession is also found at sector "East" after 1995 (Fig. 4). Davies et al. (2012) and Hulbe et al. (2004) supposed that the disintegration of a nearby ice shelf affects the local climate. The air temperatures would rise due to the presence of more ice free water in summers. Thus, the higher retreat rates at sector "East" after 1995 could be indirectly caused by the disintegration of Prince Gustav and Larsen A Ice Shelf in Sector "East-Ice-Shelf". At Base Marambio, ~100 km east of this sector, approximately 2°C higher mean annual air temperatures were recorded in the period 1996-2005 as compared to the period 1986-1995 (Oliva et al., 2017). Unfortunately, no temperature data recorded within sector "East" is available covering this period that could be used to validate this hypothesis.

The average changes of flow velocities at each sector also vary strongly (Table 5) in the observation period 1992-2014. On the west coast an increase of 41% is found, whereas in sector "East" the glaciers slowed down by approximately 58% and at the ice shelf tributaries the ice flow increased on average by 26%. Pritchard and Vaughan (2007) reported an increase in mean flow rate of 7.8% in frame 4923 (the central and much of the northern part of sector "West") and 15.2% in frame 4941 (the southern part of sector "West") for the period 1992-2005 (frame numbers correspond to European Space Agency convention for identifying ERS coverage). This spatial trend corresponds to our observations, since most of the glaciers which accelerated are located at the southern end of sector "West". However, for the same observation period we derived a mean increase in flow velocity by 18.9 % in sector "West", which is an approximately 1.6 times higher acceleration. Pritchard and Vaughan (2007) estimated the mean velocity change by measuring the flow speed at profiles along the flow direction of the glacier, whereas we measured the velocity across glacier profiles at the terminus. If a tidewater glacier speeds up due to the destabilization of its front, the highest acceleration is found at the terminus (see Seehaus et al., 2015, Fig. 3). Consequently, the different profile locations explain the deviations between both studies.

In the following section the observed changes in the individual sectors are discussed in more detail.

## 5.1 East

The glaciers north of the former Prince Gustav Ice Shelf show a general deceleration. Eyrie, Russell East, TPE130, TPE31, TPE32, TPE34, and "2731" glaciers experienced a rapid decrease and, except "2731" Glacier, a subsequent stabilization or even gentle acceleration of flow velocities (Fig. S2, S6, S7 and S9-S12). A significant retreat followed by a stabilization or slight re-advance of the calving front position is also observed at these glaciers. According to Benn and Evans (1998), a small retreat of a glacier with an overdeepening behind its grounding line (i.e. where the bed slopes away from the ice front) can result in a rapid recession into the deepening fjord. The increased calving and retreat of the ice front cause stronger up-glacier driving stress, higher flow speed as well as glacier thinning and steepening (Meier and Post, 1987; Veen, 2002). The glacier front stabilizes when the grounding line reaches shallower bathymetry and ice flow also starts to slowdown. A delay between the front stabilization and slowdown can be caused by thinning and steepening of the glacier. Additionally, the accelerated ice flow can surpass the retreat rates and cause short-term glacier advances in the period of high flow speeds (e.g. Eyrie, Russel East, TPE130 and TPE32 glaciers, Fig. S6, S7, S9 and S11) (Meier and Post, 1987). This process can be initiated by climatic forcing (Benn and Evans, 1998). Significant higher surface air temperature at the north-eastern AP and a cooling trend in the 21$^{st}$ century was reported by Oliva et al. (2017) and Turner et al. (2016) (see Section 1). Hence, we assume that the initial recessions of the glaciers in sector "East" were forced by the warming observed by Oliva et al. (2017) and Skvarca et al. (1998) since the 1970s. Therefore, this initial frontal destabilization and retreat led to high flow speeds at the beginning of our ice dynamics time series (earliest velocity measurements from 1992) and the subsequently observed frontal stabilization (after 1985) caused the deceleration of the ice flow. The fjord geometry significantly affects the dynamics of the terminus of a tidewater glacier (Benn and Evans, 1998; Van der Veen, 2002). The tongues of Aitkenhead and "2707" glaciers are split into two branches by nunataks, resulting in rather complex fjord geometries. A retreat from pinning points (e.g. fjord narrowing) causes further rapid recession and higher flow speeds until the ice front reaches a new stable position as observed at "2707" and Aitkenhead Glacier (Fig. S1 and S3). At TPE10 Glacier (Fig. S8 and S82) a "peaked" flow velocity evolution is observed as at Aitkenhead Glacier (Fig. S3 and S77). No nunatak is present at the terminus, but small rock outcrops, indicating a shallow bedrock bump, are identified north of the center of the ice front by visual inspection of optical satellite imagery. Most probably, this shallow bedrock acts as a pinning point and prevents further retreat. The front of Broad Valley Glacier (Fig. S4) is located in a widening fjord. This geometry makes the glacier less vulnerable to frontal changes (Benn and Evans, 1998). Therefore, no significant changes in flow velocities are observed as a consequence of the frontal recession and re-advance.

Diplock and Victory glaciers (Fig. S5 and S13) show a decrease of flow speed during retreat (1995-2010) followed by an acceleration combined with frontal advance (2010-2015). Surge-type glaciers (tidewater as well as land terminating), found in various regions worldwide show similar behavior (Meier and Post, 1969; Sevestre and Benn, 2015). They are characterized by episodically rapid down-wasting, resulting in a frontal acceleration and strong advance. Regarding

tidewater glaciers the advance can be strongly compensated by increased calving rates in deepwater in front of the glacier. It is therefore possible that these glaciers may have experienced a surge cycle in our observation period; however, a longer time series analysis is necessary to prove this hypothesis.

## 5.2 East-Ice-Shelf

5    In the sector "East-Ice-Shelf" the tributary glaciers in the Larsen A embayment ("2558", Arron Icefall, DBE, Drygalski, LAB2, LAB32, TPE61 and TPE62; Fig. S14, S17, S19-S22, S25 and S26) and Sjögren-Inlet (Boydell, Sjögren and TPE114; Fig. S18, S23 and S24) lost the downstream Larsen A and Prince Gustav Ice Shelf in 1995. Nearly all glaciers showed a rapid and significant acceleration after ice shelf break up and a subsequent slow down. A gentle peak in flow speeds is obtained at LAB32 and TPE114 glaciers. They are classified as "stable", since the variations are below the threshold of 0.25

10   m d$^{-1}$, according to the categorization in Table 3. Dramatic speed up with subsequent deceleration of former ice shelf tributaries was reported by various authors; e.g. in this sector by Seehaus et al., (2015, 2016) at DBE and Sjögren-Inlet glaciers and further south at Larsen B embayment by Rott et al. (2011) and Wuite et al. (2015). The velocities reported by Rott et al. (2014) at Sjögren, Pyke, Edgeworth and Drygalski glaciers are generally higher than our findings. The authors measured the velocities at locations near the center of the glacier fronts, where the ice flow velocities are typically highest,

whereas we measured the median velocities at cross profiles close to the glacier fronts (Seehaus et al. 2015). The different approaches result in different absolute values (see also Section S1 in the supplement), but comparable temporal developments in glacier flow speeds are observed by both author groups. For example Rott et al. (2015) presented surface velocity measured along a central flow line of Drygalski Glacier. Figure S149 shows our surface velocity measurements across the terminus of Drygalski Glacier and Fig S94 velocity measurements at the maximum ice thickness across the

terminus profile. Both studies show comparable values (e.g. in 1995: this study ~2.7 m/d, Rott et al. (2015) ~2.8 m/d; in 2009: this study ~5.5 m/d, Rott et al. (2015) ~6.0 m/d) at the terminus.

Highest peak values of 6.3 m d$^{-1}$ are found at TPE61 Glacier in November 1995 and January 1996. Most glaciers (Arron Icefall, Drygalski, LAB2, TPE61, TPE62) strongly decelerated after the initial acceleration and show almost constant flow speeds in recent years, indicating that the glaciers adjusted to the new boundary conditions, albeit significant higher flow

speeds (compared to pre-ice-shelf-collapse conditions) can be observed at the central sections of the terminus (see Section S1 and Fig. S149 in the supplement). At "2558", Boydell, DBE and Sjögren glaciers the deceleration is ongoing and Boydell and DBE glaciers still show increased flow speeds at the glacier fronts. We suppose that these tributary glaciers show a prolonged response to ice shelf disintegration, caused by local settings (e.g. bedrock topography or fjord geometry), and are still adjusting to the new boundary conditions, as suggested by Seehaus et al. (2015, 2016).

In the 1980s, Prince Gustav Ice Shelf gradually retreated (see Fig. 1) and "2668" Glacier (Fig. S15) has not been buttressed by the ice shelf since the early 1990s. A deceleration is found in the period 2005-2010. Hence, this glacier may also have

experienced a speed up in the early 1990s due to the recession of Prince Gustav Ice Shelf in the 1980s. However, the earliest velocity measurement at "2668" Glacier is only available from February 1996.

The ice shelf in Larsen Inlet disintegrated in 1987-1988 and earliest velocity measurements are obtained in 1993. As for "2668" Glacier no sufficient cloud free coverage by Landsat imagery is available which facilitates the computation of surface velocities for the 1980s. The ice flow speeds at APPE glaciers (Fig. S16) are nearly stable with short term variations in the order of 0.2-0.5 m d$^{-1}$ between 1993 and 2014. Rott et al. (2014) also found nearly constant flow velocities at Pyke Glacier (part of the APPE basin, Table 1). The authors suggest that the ice flow of APPE glaciers was not strongly disturbed by the ice shelf removal due to the steep glacier surfaces and shallow seabed topography at the glacier fronts (Pudsey et al., 2001).

**5.3 West**

The glacier geometries differ strongly along the west coast. In the southern part of sector "West" the shoreline is more ragged and islands are near the coast. An impact of the islands on the climatic conditions at the AP mainland's coastline (e.g. orographic barrier) is not obvious (visual inspection of RACMO2.3 5.5 km grid cell model results (Van Wessem et al., 2016)). However, the climatic conditions on the AP show strong spatial and temporal variability (see Section 1.2 and 3.3). These factors cause the heterogeneous spatial pattern of area and flow speed changes in sector "West" as compared to the eastern sectors.

Kunz et al. (2012) observed thinning at the glacier termini along the western AP, by analyzing airborne and spaceborne stereo imagery in the period 1947-2010. Two of the twelve studied glaciers are located within our study area; Leonardo Glacier (1968-2010) and Rozier Glacier (1968-2010). An acceleration and terminus retreat can be caused by frontal thinning as shown by Benn et al. (2007). However, Benn et al. (2007) also point out that changes in ice thickness do not necessarily affect the ice flow and that calving front positions and ice dynamics are strongly dependent on the fjord and glacier geometries, derived from modeling results which have higher uncertainties especially for smaller basins.

The large number of glaciers in this sector is analyzed by means of a hierarchical cluster analysis (Section 3.4) and assorted into four groups based on the resulting dendrogram (Fig. 6). Boxplots of the individual input variables of each group are shown in Fig. 7. The correlation between the observed ice dynamics and the glacier geometries of each group are discussed in the following sections (see also Fig. 7).

**Group 1 (14 glaciers):**

Most glaciers experienced acceleration in the period 1992-2014. The majority of the glacier basins are "very top-heavy" or "top-heavy" (median $HI$ = -1.8), stretching from sea level up to 1892 m on average. The $b_{clim}$ increases toward higher

altitudes (Van Wessem et al., 2016) and highest values are found in the zone between 1000 and 1700 m a.s.l. Consequently these glaciers receive high mass input in their large high altitude accumulation areas. The accumulation is known to have significantly increased on the AP by 20% since 1850 (Thomas et al., 2008). Pritchard and Vaughan (2007) reported that only a small fraction of the acceleration can be attributed to glacier thickening due to increased mass input. Up-glacier thickening combined with frontal thinning (reported by Kunz et al., 2012) leads to a steepening of the glacier and an increase in driving stress, resulting in faster ice flow (Meier and Post, 1987) as observed in this study. Moreover, a thinning of the terminus reduces the effective basal stress of a tidewater glacier and facilitates faster ice flow (Pritchard and Vaughan, 2007). The flux gate cross sections to catchment size ratios are relatively small, indicating narrowing catchments towards the ice front. The channelized increased ice flow almost compensates for the increased calving rates (due to frontal thinning), resulting in an average recession of the glaciers by only 0.2% in the period 1985-2015. The high flow speeds may outweigh the calving and lead to ice-front advances as measured at Krebs and TPE46 Glacier. The glacier termini of this group are typically located in narrow fjords (Fig. 5) and are clustered in Charcot, Charlotte and Andvord Bay.

**Group 2 (19 glaciers)**

Glaciers of group 2 are spread all over sector "West", with a local clustering in Wilhelmina Bay. Group 2 shows similar $h_{max}$ and *FA* characteristics to group 1. Area changes are also quite small (-0.1%). Most of the glaciers experienced acceleration or show a "peaked" evolution of the flow velocities. In contrast to group 1 the catchments are in general "bottom-heavy" and some are even "very bottom-heavy". We assume that the constraints are similar to group 1 (increasing $b_{clim}$, frontal thinning and steepening). However, the additional mass accumulation in the upper areas is smaller due to the "bottom-heavy" glacier geometries. Consequently, the imbalance due to the frontal thinning and up-glacier mass gain is less pronounced as in group 1 and numerous glaciers ("peak" type) started to decelerate after the speed-up, indicating that these glaciers are adjusting to the new boundary conditions.

**Group 3 (13 glaciers)**

These basins typically show a "bottom-heavy" hypsometry and smaller elevation ranges (in average up to 1103 m a.s.l.). Thus, $b_{clim}$ is relatively low. The smaller mean ice thickness at the termini (161 m, compared to 211 m of all glaciers) of group 3 implies less interaction with the ocean, leading to a small average frontal retreat of ~0.1%. The low frontal ablation does not significantly affect the ice flow, probably due to the flat glacier topography and the low mass input. Consequently, the flow speed is in general stable or even slightly decreases in the observation period. Glaciers of group 3 usually face the open ocean, and do not terminate in narrow fjords (especially in the northern part, Trinity Peninsula).

**Group 4 (3 glaciers)**

All basins in this group have a "very bottom-heavy" hypsometry and an elevation range comparable to group 3 glaciers. The *FA* factors are in general higher than in group 3, implying that outflow of the catchments is less channelized and the glacier fronts are long compared to the catchment sizes. Therefore, the largest relative area changes, in average -5.1%, are found at glaciers in group 4. However, the absolute frontal retreat is small and does not significantly affect the glacier flow. Note: Group 4 consists of only three samples, limiting the significance.

## 6 Conclusions

Our analysis expands on previous work (Pritchard and Vaughan, 2007) on ice dynamic changes along the west coast of AP between TPE8 and Bagshawe-Grubb Glacier, both in regard to temporal coverage and analysis methods. It also spatially extends previous work on changes in ice dynamics along the east coast between Eyrie Bay and the Seal Nunataks. The spatially and temporally detailed analysis of changes in ice flow speeds (1992-2014) and ice front positions (1985-2015) reveal varying temporal evolution in glacier dynamics along the northern AP. The results are in general in line with findings of the previous studies; however, along the west coast a more accelerated glacier flow is determined and on the eastern side temporal evolution of ice dynamics of 21 glaciers is observed for the first time. A large variety of temporal variations in glacier dynamics were observed in our studied area and attributed to different forcing and boundary conditions.

On the east side all glacier fronts retreated in the study period (relative to 1985, relative to 1995 for former Larsen-A and Prince Gustav Ice Shelf tributaries, see also Section 5.2), with highest retreat rates observed at former tributaries of the Prince Gustav, Larsen Inlet and Larsen A ice shelves. Moreover, nearly all the glaciers affected by ice shelf disintegration showed similar temporal evolutions of ice velocities. The glaciers reacted with a strong acceleration to ice shelf break up followed by a deceleration, indicating that the glaciers adjusted or are still adjusting to the new boundary conditions. Glaciers on the east coast north of the former Prince Gustav Ice Shelf showed in general a significant deceleration and a reduction in frontal ablation. Based on the observed warming trend since the 1960s and the subsequent cooling since the mid-2000s in the northern AP, we suggest that the initial recession and speed up of the glaciers took place before the start of our observation and that the glaciers are now close to a new equilibrium.

The average flow speed of the glaciers along the west coast of the Antarctic Peninsula significantly increased in the observation period but the total frontal change is negligible. No general evolution in ice dynamics of the glaciers at the west coast is obvious. However, correlations between the changes in ice dynamics and the glacier geometries of the individual catchments are obtained by applying a hierarchical cluster analysis. Thus, the geometry of the individual glacier basin strongly affects the reaction of the glacier to external forcing.

We conclude that for regions with such a strong spatial variation in topographic and climatic parameters as the AP, it is impossible to derive a regional trend in glacier change by simply analyzing individual glaciers in this region. Therefore

further detailed observation of the glaciological changes along the AP is needed. Upcoming sensors hopefully facilitate the region wide measurement of recent surface elevation, since current estimates have got only partial coverage or have got some issues due to the complex topography of the AP. Moreover, future activities should link remote sensing derived ice dynamics and glacier extent with ocean parameters and ocean models, as well as regional climate models and ice dynamic models, in order to provide a better quantification of mass changes and physical processes leading to the observed changes.

*Author contributions.* T.S. designed the study, processed the SAR data, performed the data analysis and led the writing of the manuscript, in which he received support from all authors. A.C. and A.S. compiled and provided glacier front position data sets. M.B. initiated the project and coordinated the research.

*Competing interests.* The authors declare no competing financial interests.

*Acknowledgements.* This work was supported by the Deutsche Forschungsgemeinschaft (DFG) in the framework of the priority programme "Antarctic Research with comparative investigations in Arctic ice areas" by a grant to M.B. (BR 2105/9-1). MB and TS would like to thank the HGF Alliance "Remote Sensing of Earth System Dynamics" (HA-310) and Marie-Curie-Network International Research Staff Exchange Scheme IMCONet (EU FP7-PEOPLE-2012-IRSES) for additional support. Access to satellite data was kindly provided by various space agencies, e.g. under ESA AO 4032, DLR TerraSAR-X Background Mission Antarctic Peninsula & Ice Shelves, TSX AO LAN0013, TanDEM-X Mission TDX AO XTI_GLAC0264, ASF, GLIMS as well as NASA and USGS. We also thank Etienne Berthier and Olaf Eisen for editing and Jan Wuite and another anonymous referee for reviewing this paper.

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

## Figures

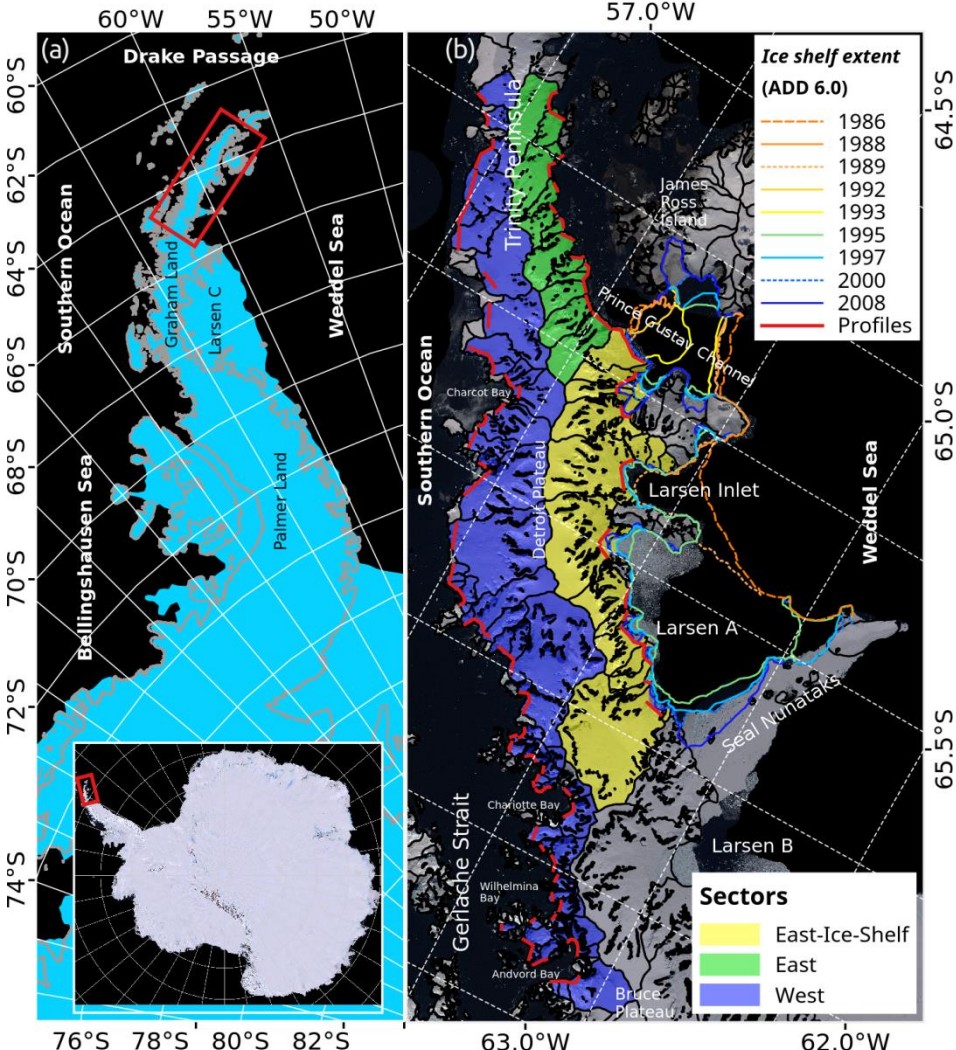

**Figure 1.** Panels **(a):** Location of study site on the Antarctic Peninsula and on the Antarctic continent (inset). Panel **(b)**: Separation of study site in 3 sectors and retreat states of Prince Gustav and Larsen A ice shelves. Red lines: profiles at glacier front for velocity measurements. Map base, Landsat LIMA Mosaic USGS, NASA, BAS, NSF, coastlines (ice shelf extent) and catchment delineations from SCAR Antarctic Digital Database 6.0.

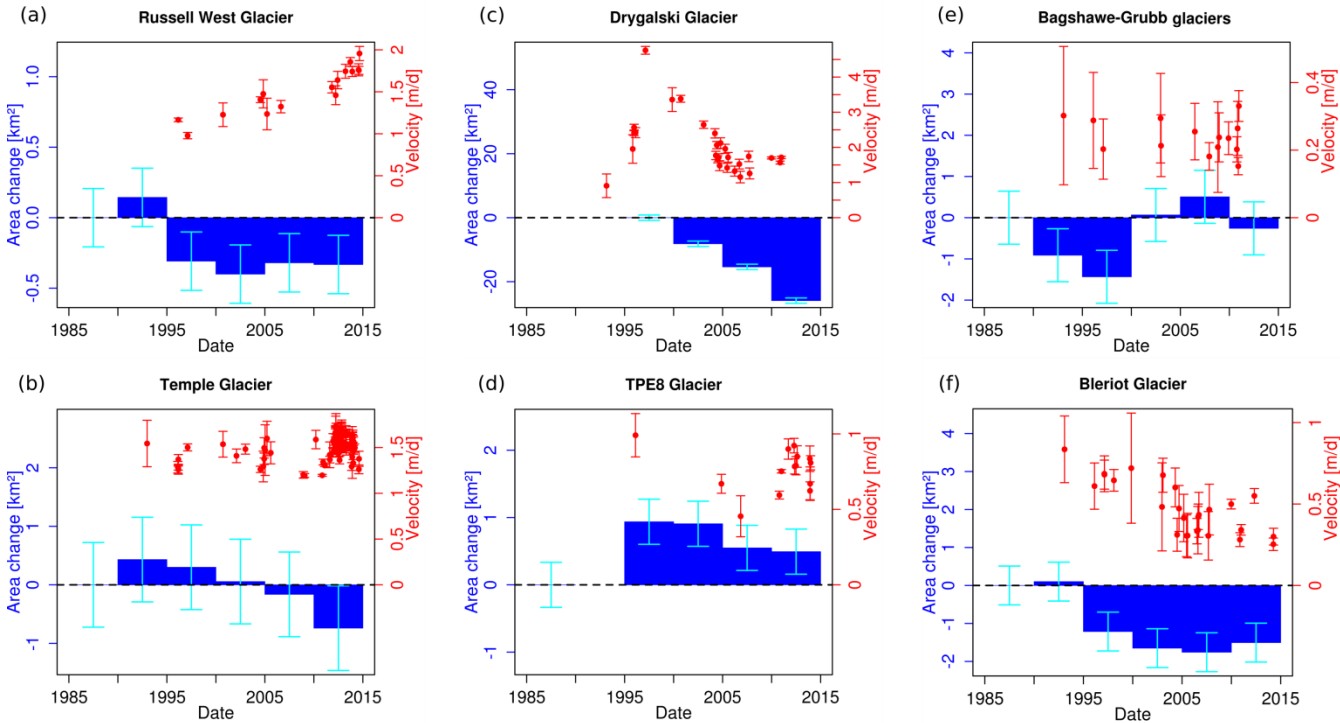

**Figure 2.** Temporal evolution of surface velocity (red, using first measuring approach) and area (blue) changes of selected glaciers in the study area for each velocity change category (see Table 3).

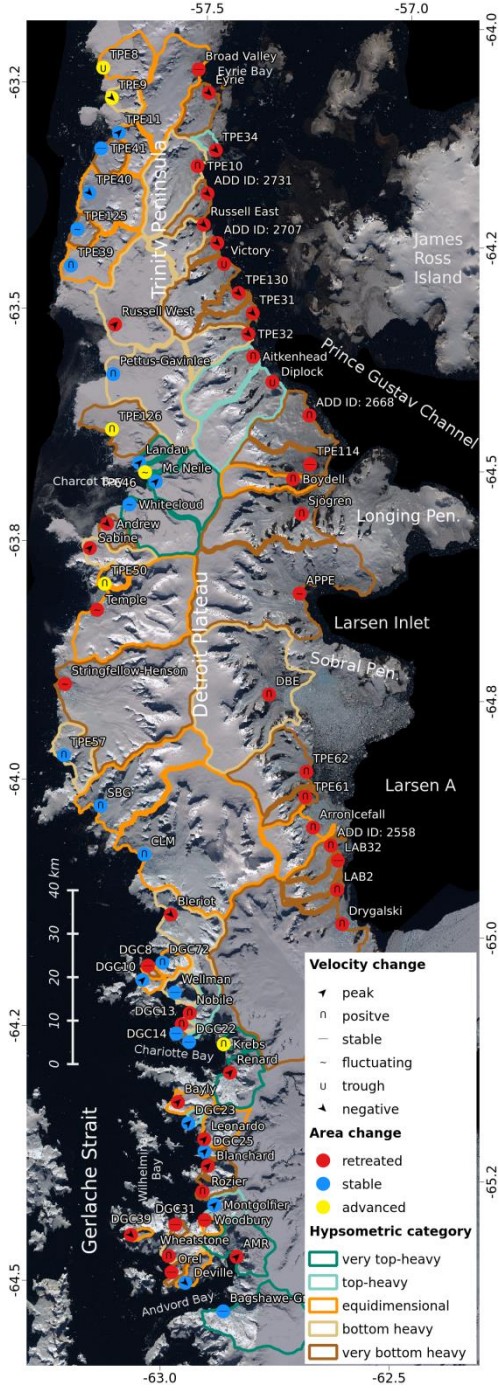

**Figure 3.** Categorizations of glaciers based on the temporal variations of area changes (dots) and flow velocities (symbols). Colors of catchment delineation indicate Hypsometric categories according to Jiskoot et al. (2009). Background: Landsat LIMA Mosaic USGS, NASA, BAS, NSF

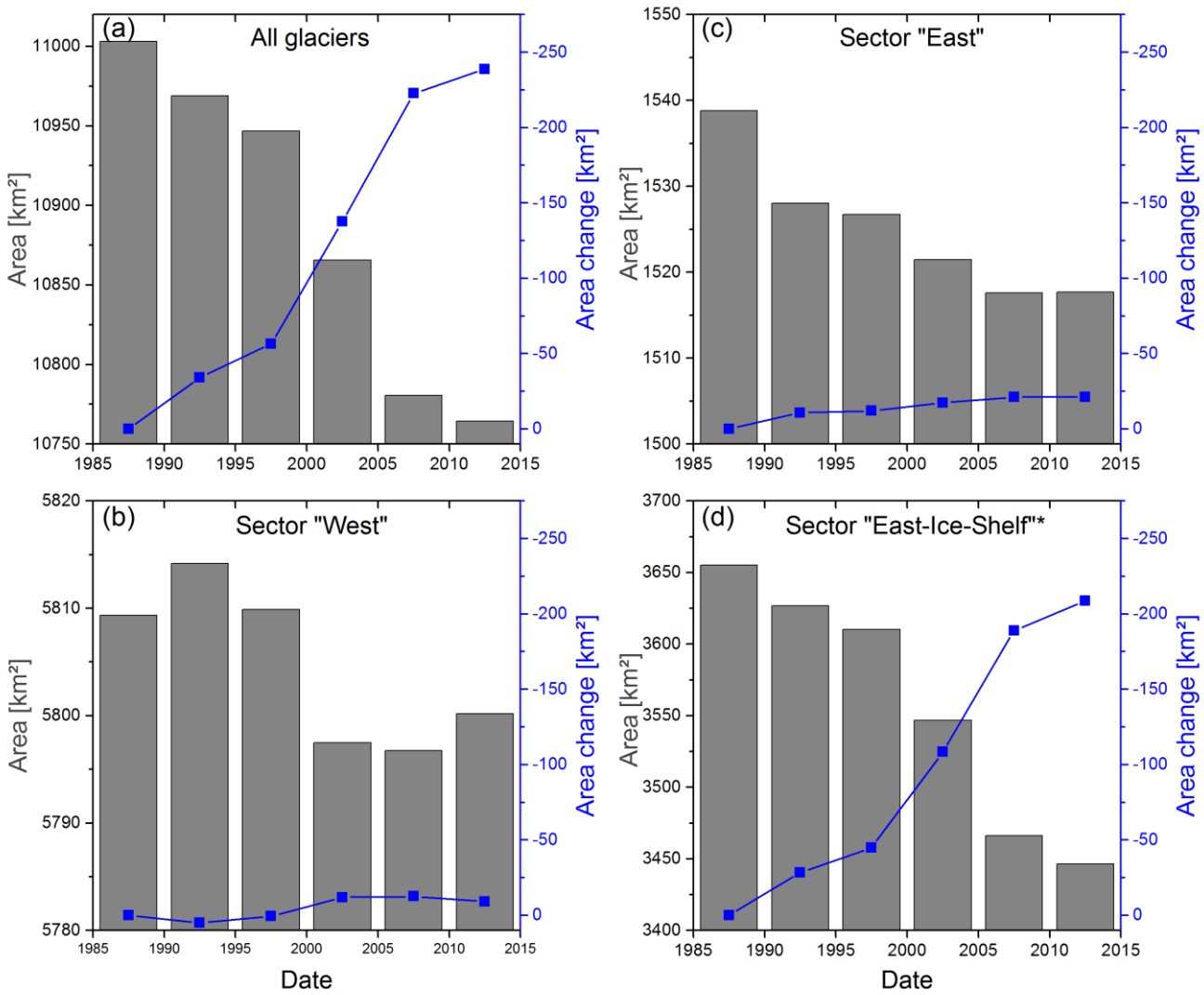

**Figure 4.** Total glacier area (gray bars) of the whole study site (Panel **(a)**) and of the individual sectors (Panels **(b)-(d)**) in the period 1985-2015. Changes in glacier area (blue points) are relative to the measurements in time interval 1985-1990. Note the different scaling of the left y-axes. *In sector "East-Ice-Shelf", area changes before 1995 are only measured at Larsen Inlet tributaries (APPE glaciers).

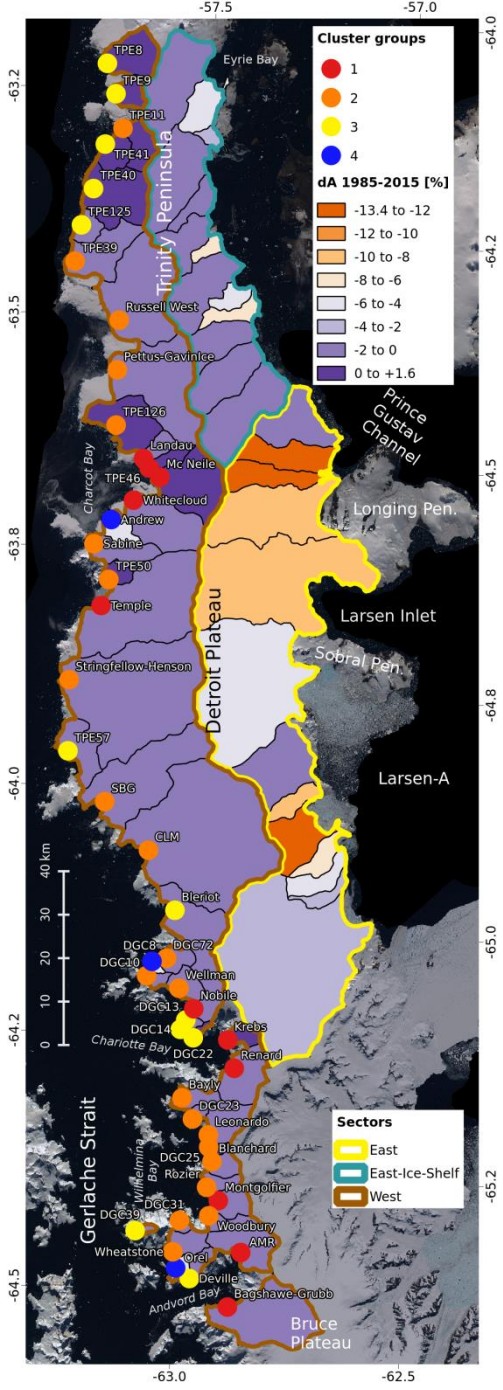

**Figure 5.** Spatial distribution of glacier types along the west coast. Glaciers are group based on a hierarchical cluster analysis (dots). In Section 5.3 the characteristics of the groups are discussed in detail. Individual glacier catchment colors: relative area change in the period 1985-2015. Colored polygon outlines: Boundaries of the three sectors. Background: Landsat LIMA Mosaic USGS, NASA, BAS, NSF

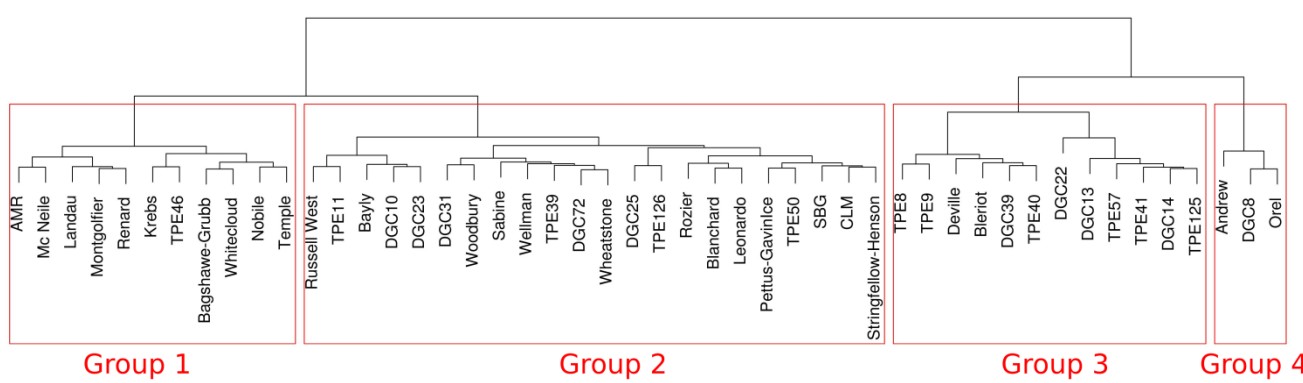

**Figure 6.** Dendrogram of hierarchical cluster analysis of glaciers in sector "West". The glaciers are assorted in four groups (red rectangles). See also Section 5.3.

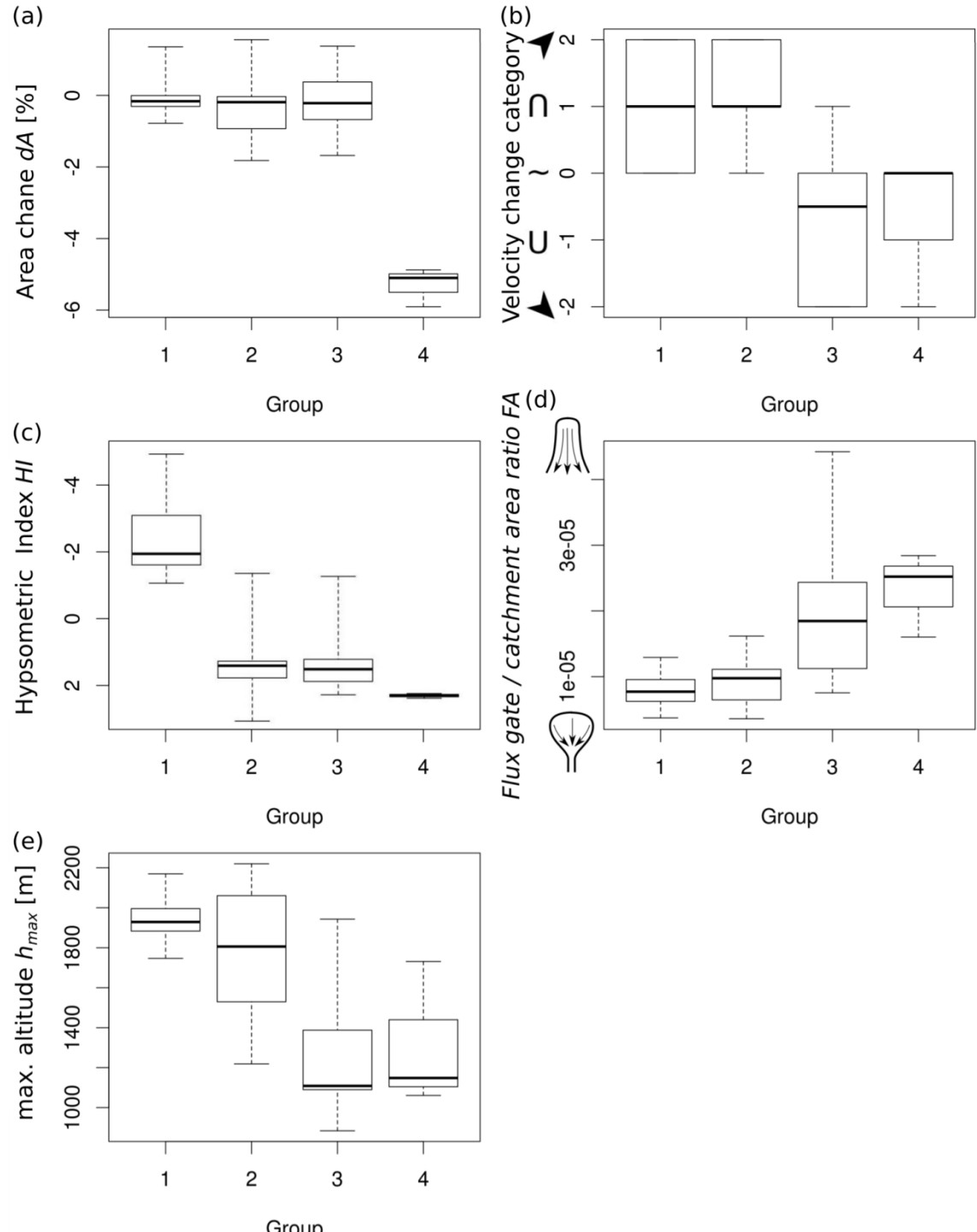

**Figure 7.** Boxplots of cluster analysis input variables (Sector "West") for each group. Whiskers extend to the most extreme data points. Panel **(b)**: The symbols used for the velocity change categories (see Table 3) are the same as in Fig. 3. Panel **(d)**: The pictograms illustrate the catchment shape (see Section 3.3).

**Tables**

5    **Table 1.** Abbreviations of glacier names

| Abbreviation | Glacier names |
|---|---|
| AMR | Arago-Moser-Rudolph |
| APPE | Albone-Pyke-Polaris-Eliason |
| CLM | Cayley-Lilienthal-Mouillard |
| DBE | Dinsmoor-Bombardier-Edgeworth |
| SBG | Sikorsky-Breguet-Gregory |

**Table 2.** Overview of SAR sensors and specifications used in this study.

| Platform | Sensor | Mode | SAR band | Repetition cycle [d] | Time interval | Ground range resolution [m]* | Tracking patch sizes [p x p]+ | Tracking step size [p x p]+ | Mean uncertainty of tracking results [m/d] |
|---|---|---|---|---|---|---|---|---|---|
| ERS-1/2 | SAR | IM | C band | 35/1 | 08. December 1992 <br><br> 02. April 2010 | 30 | 48x240 <br><br> 64x320 | 5x25 | 0.15±0.10 |
| RADARSAT 1 | SAR | ST | C band | 24 | 10. September 2000 <br><br> 03. September 2006 | 30 | 48x192 <br><br> 64x256 | 5x20 | 0.11±0.03 |
| Envisat | ASAR | IM | C band | 35 | 05. December 2003 <br><br> 16. August 2009 | 30 | 32x160 <br><br> 64x320 <br><br> 128x640 | 5x25 | 0.12±0.05 |
| ALOS | PALSAR | FBS | L band | 46 | 18. May 2006 <br><br> 17. March 2011 | 10 | 64x192 <br><br> 96x192 <br><br> 128x384 | 10x30 | 0.05±0.06 |
| TerraSAR-X TanDEM-X | SAR | SM | X band | 11 | 14. October 2008 <br><br> 22. December 2014 | 3 | 128x128 <br><br> 256x256 <br><br> 512x512 | 25x25 | 0.06±0.04 |

* nominal resolution; depending on the incidence angle.

+ intensity tracking parameters are provided in pixels [p] in slant range geometry.

**Table 3.** Description of velocity change categories.

| Category | Description | Rating[*] |
|---|---|---|
| positive | General increase of flow speed | 2 |
| peak | Increase of flow speed with subsequent deceleration | 1 |
| stable | Variability of measurements $< 0.25$ m d$^{-1}$ | 0 |
| fluctuating | Short term speed-ups and deceleration, no clear trend | 0 |
| trough | Decrease of flow speed with subsequent acceleration | -1 |
| negative | General decrease of flow speed | -2 |

[*]ratings used for cluster analysis Section 3.4

**Table 4.** Hypsometric Index and glacier basin category descriptions.

| Hypsometric Index $(HI)^*$ | Hypsometric categories | Number of Glaciers |
|---|---|---|
| $HI < -1.5$ | Very top-heavy | 8 |
| $-1.5 < HI < -1.2$ | Top-heavy | 7 |
| $-1.2 < HI < 1.2$ | Equidimensional | 18 |
| $1.2 < HI < 1.5$ | Bottom-heavy | 13 |
| $HI > 1.5$ | Very bottom-heavy | 28 |

[*]according to Jiskoot et al., (2009)

**Table 5.** Summary of observed parameters for each sector and all glaciers.

| Sector | East | East-Ice-Shelf | West | All glaciers |
|---|---|---|---|---|
| $N$ | 13 | 13 | 48 | 74 |
| $l_f$ [m] | 85114 | 127909 | 268763 | 481786 |
| $A_{1985\text{-}1990}$ [km²] | 1538.78 | 3655.13 | 5809.33 | 11003.23 |
| $A_{2010\text{-}2015}$ [km²] | 1517.71 | 3446.54 | 5800.18 | 10764.42 |
| $dA$ [km²] | -21.07 | -208.59 | -9.14 | -238.81 |
| $dt$ [a] | 18.22 | 19.05 | 19.58 | 19.25 |
| *First velocity measuring approach* | | | | |
| $v_S$ [m d$^{-1}$] | 0.729 | 0.480 | 0.428 | 0.490 |
| $v_E$ [m d$^{-1}$] | 0.306 | 0.562 | 0.605 | 0.545 |
| $dv$ [m d$^{-1}$] | -0.423 | 0.081 | 0.177 | 0.055 |
| $n_v$ | 277 | 550 | 1429 | 2256 |
| Second velocity measuring approach | | | | |
| $v_S$ [m d$^{-1}$] | 1.834 | 0.760 | 0.831 | 0.994 |
| $v_E$ [m d$^{-1}$] | 0.562 | 1.071 | 1.200 | 1.065 |
| $dv$ [m d$^{-1}$] | -1.272 | 0.312 | 0.369 | 0.071 |
| $n_v$ | 355 | 639 | 1742 | 2736 |

$N$ – number of studied glaciers

$l_f$ – length of ice front

A – glacier area in the respective period (subscript)*

5   dA – change in glacier area between 1985 and 2015*

dt - mean time period of velocity measurements

$v_S$ – mean of earliest velocity measurements (1992-1996)

$v_E$ – mean of latest velocity measurements (2010-2014)

dv – mean velocity change

5    $n_v$ – sum of velocity measurements in the observation period (dt)

*since 1995 for the former Larsen-A and Prince Gustav Ice Shelf tributaries (see Section 5.2)