# Peer review of "Changes in glacier dynamics in the northern Antarctic Peninsula since 1985"

_The Cryosphere, 2017_

## Referee Comment (RC1) · Anonymous Referee #1 · 20 Apr 2017

Review of "Changes in glacier dynamics at the northern Antarctic Peninsula since 1985" by Thorsten Seehaus, Alison Cook, Aline B. Silva, and Matthias Braun as submitted to The Cryosphere Discussions

General Comments from the paper for the Authors –

The authors are to be appreciated for assembling an extensive array of illuminating data sets for a fairly large portion of the Antarctic Peninsula. By extending and expanding a previous study (Seehaus et al., EPSL 2015), it is clear that the hope was to illuminate many more glacial basins in this area of ongoing response to climate change. The use of the 5- parameter cluster analysis was a brave attempt to derive common themes across the area. Unfortunately, the complexities of the areas being investigated and the shorter/irregular nature of the velocity data appear to have confounded confident

conclusions as the authors note on Page 14. A carefully edited paper with improved figures focusing on what is clearly known over the 1985 to 2015 area change period and the ~1992 to 2014 velocity data time frame will likely be publishable in TC.

Specific Comments from the text for the Authors –

Abstract (Page 1 Line 9): The first three sentences should emphasize that this study will attempt a comprehensive analysis rather than 'other analyses have been lacking/missing' or too focused on the shelf collapse glaciers. Page 1 Line 13: The <65° latitude limit would include some of the Larsen B's major tributary glaciers so a less ambiguous way of defining the basins chosen for study is needed here and in the Introduction. Page 1 Lines 15/16: Here and elsewhere the area changes need to be attributed to a specific year or by 'the end of the study period' or similar text. The Prince Gustav Channel ice shelf's northern limit is from what year? What is the standard deviation of the average velocity for those glaciers? 'Whereat' appears to be an archaic term. Page 1 Line 19: Similarly, what is the standard deviation of the average velocity?

1.0 Introduction –

Page 1 Line 29: It seems important to have the word 'estimated' before mass balance given that IMBIE was a 'consensus' report. Page 2 Line 9: Here and elsewhere it seems more appropriate to put references chronologically from early to later. Page 2 Line 23: 'The collected observations reported in these studies suggest' rather than 'the observations suggest'... Page 2 Line 28: 'methodologically' rather than 'methodically'

2.0 Study Site

Page 3: This section MUST explain why a region that is only about 25% of the total AP was chosen for study. This should also include why sections of even the 330 km long area are excluded. Vague phrasing such as 'apart from those that are ice shelf tributaries, nearly all glaciers on the AP are marine-terminating' doesn't explain why much

of the west coast + nearby major islands are excluded from this study. Page 3 Lines 3/4: 'high precipitation' and 'orographic barrier' could use numerical support. Does the whole selected study site act as the barrier or just the broad plateaus? Better graphics and labeling will help as noted further below. Page 3 Line 11: Order the shelf areas chronologically. Page 3 Line 12: The Scambos et al. (2003) sentence needs to be balanced with a more recent reference such as Holland et al. (2015). Page 3 Line 14: Insert 'frequently' before 'experiences melting'; other areas in Antarctica experience periodic melt events, especially a number of shelf areas (see just published work in Nature). Page 3 Line 16: 'Narrow' seems an odd choice given the adjacent/excluded islands and smaller peninsulas and the broad plateaus (named elsewhere) in the study area. Page 3 Line 20: Making composite glaciers because they have 'laterally connected termini' needs to be better justified given the Seehaus et al. (2015) paper on DBE. Page 3 Line 22: 'Sparse data coverage' needs to be clarified. Page 3 Line 24: The three sectors being defined by their 'different climatic settings' needs some additional justification. Some of the 'west' glaciers are shielded to some extent by large/high islands?

3.0 Data and Methods –

3.1 Area changes –

Page 4 Line ~1: I find sections that begin with no or abbreviated text frequently can be more clearly written. The 'Data and Methods' section needs an introductory paragraph that indicates why these specific data sets in the study are being utilized. Page 4 Lines 4/5: The two sentences can easily be merged with lines below them. Page 4 Lines 7/8: Distinguish sensors and satellites explicitly. Page 4 Line 13: Given the retreat processes for the PG Channel, is limiting all of the glaciers to 1995 appropriate? Page 4 Line 20: Were ratings of 4 and 5 not needed or was any such data discarded?

3.2 Surface velocities

Page 4 Line 24: Table 2 lacks SAR resolution information. Page 4 Line 28: Does

the mentioned masking eliminate glacier areas from having their full velocity patterns mapped? I think this and Line 30 could be clarified. Page 5 Line 7: Put a period after 'topography' and start the next sentence with 'The results are then geocoded...' Page 5 Lines 8-10: Some discussion of the limitations of the ASTER DEM is needed (this also potentially impacts the cluster analysis). Page 5 Line 11: Are there no reference for the text in this paragraph? Is this a unique approach or are there any similar analyses? Does any of this approach depend on the native resolution of the SAR sensor utilized (add column in Table S2)? Page 6 Line 1: Please give the time frame for when the terminus profiles were defined. The phrase "taking into account temporal changes' suggests there is a broad range of profile times rather than a consistent time. Page 6 Lines 2/3: The second sentence needs to be clarified. Page 6 Lines 7-9: Change text to 'three or more' rather than 'more than two' and discuss if 3 observations in 10 years is adequate to 'classify' a basin as in Table 3 (with potential impact to the cluster analysis). Clarify if any of the '74 basins' were specifically excluded or does this apply only to the smaller areas that appear to be excluded (see Figure 5). Also, a plot showing the number of velocity observations as a function of (named) basin size with indications of latitude may be useful given the 'sparse' coverage of the northern Trinity Peninsula (Page 3 Line 22).

3.3 Catchment geometries and settings

Page 6 Lines 12-14: It seems appropriate to mention this analysis and how/why it differs from the earlier work led by Cook (Huber et al., 2017) http://www.earth-syst-sci-data.net/9/115/2017/essd-9-115-2017.pdf Page 6 Line 17: Does accumulation increase with higher altitude on both sides? Does this apply mostly to the plateaus? Please clarify. Page 6 Line 20: Add the Jiskoot et al. reference(s) here, not just in Table 4. Page 6 Lines 23-25: These two sentences need some expansion, perhaps to include the impact of the DEM's uncertainty and or any issues in defining the flux gates. A plot would be better than just stating 'lower values indicate a channelized outflow'.

3.4 Cluster analysis –

Page 6 Line 26: Given that uncertainties in several of the five variables underlying the cluster analysis have not been explored, it is difficult to accept this approach. If this technique has been utilized practically in other similar glaciologic studies, please provide a reference(s). The standardization technique described (Page 7 Lines 2/3) could use some clarification and also a reference. Page 7 Lines 4-7: This is rather unclear and this technique could very much use an analogy or similar technique to make it clearer to the reader what is actually being done to 'sort the basins' into groups with common parameters.

4.0 Results

4.1 Area changes

Page 7 Line 8: This section also needs an introductory paragraph that summarizes what will be discussed in the sub sections. Page 7 Lines 10/11: Explain why these glaciers were chosen (all but one are from the 'West' region). It appears that they illustrate not just the three 'area change groups' but also the six 'velocity change groups' (Table 3). Is this correct? If using 'Figure' within a sentence, please spell it out. Use 'Fig.' as in (Fig. 3). Page 7 Line 16: Assume you mean '238 km2'. Also, see comments on Figure 4 that seem designed to greatly accentuate the '2.2%' loss between 1985 and 2015. Page 7 Line 17: You could usefully add the individual loss % values here.

4.2 Surface velocities

Page 7 Line 22: 'A total of' 282 etc... Page 7 Lines 23-26: Are the 'average' uncertainties of the velocity fields meaningful given the array of different sensors used? The text suggests not. Perhaps the average uncertainty of each sensor (and its standard deviation) could be stated instead and also added to Table 2? This information is too deeply buried in Table S2. Page 7 Lines 26-28: If these data are unreliable, explain how they were or were not used in the study and all the Figures S1-74? This is unclear. Also, was there any attempt to do curve fitting through the data that passed the quality criteria? Given the range of velocity (and area change) axes used, I find it very

difficult to visually assess (Page 8 Lines 1-3) the Table 3 categories. Page 8 Lines 6/7: The 'local clustering' should be identified even if it is explored further in the Discussion section (see comments on location indicators of Figures). Page 8 Line 9: Table S2 should be S1 and there is an error in one of the subscripts and 'd' should apparently be $\Delta$, here. Also see comments on Table 5. Page 8 Line 13: You might as well give the longest period for velocity and also the standard deviation.

4.3 Catchment geometries and settings

Page 8 Lines 15/16: The HI values are in Table S1, not S2, and appear to vary quite a bit more than in Jiskoot et al. (2009). Figure 3 is very difficult to read for both velocity and HI categories. Given that this section is 'Results', perhaps the unmapped areas should be mentioned.

4.4 Cluster analysis

Page 8 Lines 19-21: In part due to the preceding text (Lines 16/17) "No clear distribution pattern can be identified, reflecting the heterogeneous topography of the AP.", my concerns about the cluster analysis remain unresolved. The limited text here, regardless of Section 5.3, seems to emphasize an uncertain result.

5.0 Discussion

Page 8 Line 25: The result that all glaciers on the east coast receded should be clarified to state 'since 1985'. Does Davies et al. (2012) overlap in terms of area with this study? Page 8 Line 27: Superscript for area is missing. Page 9 Lines 3/4: This is very difficult to ascertain from Figure 4c and seems to be an overreach of the results, the text seems speculative. See the small deviations in the area change trend for the 1995-2005 'blocks'. Page 9 Lines 6-8: Seehaus et al. (2015, Figure 3) shows warming for Marambio for 1998 to 2006 not 1997 to 2007. That time range appears to be from the Oliva et al. (2017) broader analysis who shows the locations of all the available records and their variation over a longer time frame. And it isn't clear what "Unfortunately, no

temperature records are available in sector "East" covering this period." means as all the temperature data appears to be from outside this paper's study area. Page 9 Lines 11-13: Clarify that the 'frames' correspond to ESA conventions for identifying ERS coverage and that frame 4923 covers 'the central and much of the northern part of sector "West"'. Page 9 Lines 14-19: Is this really a 'discovery' since you go on to show that the 'discrepancy' has a logical explanation?

5.1 East ice shelf 'sector' (no reason to capitalize)

Page 9 Line 22: Given Figures S1-13 describe sector "East" why start with the ice shelf loss area basins detailed in S14-26? Please add the date or dates that detail when the basins lost the ice shelf area in front of them (e.g. paragraphs on Page 10). Page 9 Line 26: Here and elsewhere, hyphens are not needed for 'Larsen-A/B'. Page 9 Line 30: It is good that you can resolve differences due solely to methodology but please clarify what 'equal temporal trends' means in this context. Page 10 Lines 2-5: It is difficult to conclude that the stated variation in the behavior of these basins shows they are still 'adjusting to the new boundary conditions' as opposed to responding to purely localized forces acting on them. On Line 3, do you mean 'medial' as opposed to the statistical 'median'? Page 10 Lines 6-15: Some interesting details are discussed here but they seem to be overly specific rather than useful indicators. The discussion of Pyke Glacier vs the composite APPE basin, including Pyke, suggests a concern about this analysis combining individual flow systems in composite basins. Does averaging over multiple smaller glaciers blur a discernable signal? The lack of sufficient temporal coverage of the available velocity data appears to be a common issue here.

5.2 East 'sector' (see comment above on order of discussion)

Page 10 Lines 20-28: It would seem that a good bit of this discussion might fit better in the introductory section. The specific figures in the Supplement would be useful to point out for the named basins. Depending on whether you choose to interpret Turner's or Oliva's figures allows you to vary the point when cooling began in the 21st century, what

specific date do you prefer? Page 11 Lines 1-4: Does the analysis of Oliva et al. (2017) not allow more precision than 'before earliest velocity measurements'? Does the area change time series going back to 1985 (in this sector) not provide additional insight? Page 11 Lines 8-10: Please be more specific as to what/how the visual imagery was used to identify the 'bump'. Page 11 Lines 13-19: Some of this material should be in the introductory material and the analysis seems speculative given the stated need for more observations. Page 11: Also highlights the difficulties in reading Figure 3 for specific locations (or interpreting symbols) even after magnification of the pdf.

5.3 West 'sector'

Page 11 Line 24: See previous comment on Turner vs Oliva temperature studies. Page 11 Lines 24/25: Clarify what is meant by 'constant trend'? Do you mean in both space and time? If so, can the ocean temperature differences be reconciled? Page 11 Lines 25/26: Does 'southern part' apply to both West and East or only 'West"? What abut the coastline makes it 'fractal' and does that aid understanding? Clarify 'These' factors lead (cause?)... Page 11 Lines 28/29: Clarify if the 12 glaciers studied by Kunz et al. (2012) included basins and years overlapping this study. Which 'authors' are being referred to here? Page 11 Line 31: The fact that fjord and glacier geometries may be uncertain should probably be mentioned here, especially for smaller basins. Page 12 to Page 13 Line 13: As indicated above, I find the cluster analysis to be of uncertain value and will refrain from further comment on it. Other reviewers and/or the Editor can decide if it should remain in the paper.

6.0 Conclusions

Page 13 Lines 15/16: The usage of 'northwestern' to define the study area is quite imprecise as is the usage of 'north of 65°S' as was previously commented. Page 13 Line 18: The 'dynamics' were observed most clearly only during ∼1992 to 2014 through the repeated velocity observations. This text should be clarified. Page 13 Line 19: Clarify if 'significantly higher' is simply due to differences in the methodology relative to

Pritchard and Vaughan (2007) for the same period. If so, should this simply say 'higher' velocities were observed? Page 13 Line 22: Be clear that all 'East' glacier fronts re-treated relative to 1985 (or 1995 after shelf losses). Page 13 Line 28: The 'cooling since 2000' depends on how you read the Seehaus et al. (2015), Turner et al. (2016) or Oliva et al. (2017) analyses. Mid-2000s seems to be a more reasonable number for much of your study area. Page 14 Lines 3-5: See previous concerns about how well the cluster analysis with 5 variables can discriminate across such a broad swath of the western AP. It appears that this study needs to include additional parameters rather than attributing groups to basin geometry alone (as is clearly indicted in their next paragraph).

Figures -

Figure 1: This figure needs to be redesigned with a small Antarctic map in the corner of the 'general peninsula region' map showing the specific study area on the ∼1300 km long Antarctic Peninsula. Major landscape features and adjacent water bodies should be clearly labeled on both of the panels especially (c) if mentioned in the text (e.g. Bruce and Detroit plateaus, James Ross Island, Charcot, Charlotte, Andvord, Wilhelmina bays, not just on Figure 5). The LIMA credit is incorrect, should be USGS, NASA, BAS, NSF. Further, the scale of the third panel should be sufficient to clearly discern ice front positions and related color choices of lines (shades of orange, red on red?) may need to be revised. It is appropriate to specify in the caption why ADD 6.0 is being used for glacier fronts instead of the data from the study. Also, areas mostly or totally excluded from the study (e.g. Trinity, Longing, Sobral peninsulas) should be identified here. Also, Bellingshausen Sea is misspelled and inaccurately located.

Figure 2: The caption seems to need to include "for each velocity change category (see Table 3)." And it does seem odd that there is only one example that is not from 'West'. As with S1 to S74, it seems appropriate to ask for both velocity and area change data to be plotted at the same scales or a compelling argument advanced as to why this is not more appropriate. This would likely greatly reduce the size of the error bars that

distract the eye in many instances. Also, as mentioned in text comments, was curve fitting of the velocity data attempted?

Figure 3: Even after magnification of the pdf, Figure 3 is difficult to read for locations and symbols and these also cannot be searched. This makes the text discussion of small features very difficult. Also, see above for the need for locations mentioned in the text to be labeled. Close inspection reveals that smaller areas appear to be excluded along with the larger Sobral and Longing peninsula regions and such areas need to be mapped/explained (also see text comments). Also, discerning the color scale for the HI outlines of each basin are challenging especially where they overlap.

Figure 4: It is positive to note that this figure's caption points out that the left y-axis (not the right one) has different scaling for each of the plots. It is appropriate for the area change y-axis to be consistently scale as that allows the reader to quickly detect the magnitude of change from region to region. It is not clear why the left y-axis doesn't start at zero in all cases and use some distinct maximum thousands value to clearly show that the changes are still small relative to the total area in each sector, especially for 'all glaciers'. The editor may wish to provide guidance here.

Figure 5: See comments on the text regarding the cluster analysis. The caption needs to clarify that all polygons in the figure are colored (see previous comment on overlapping basin outlines) but that the sectors are (somewhat) defined with three colors. Also, 'dA' should apparently be $\Delta$A. This figure finally provides some location pointers to the Trinity Peninsula (partial) and the bays missing from Figure 1 but, oddly, doesn't label any of the glaciers? This figure also highlights that 3 of the 'composite' basins are quite large (APPE, CLM, and DBE) and a fourth (SBG) is much larger than some of the investigated 'west' basins. This makes one wonder why they could not be similarly subdivided. "Laterally- connected' is not clearly explained in the text as the reason to composite these basins (how much of each glacier?).

Figure 6: See comments on the text regarding the cluster analysis. Add numbers for

each cluster group to each red box if the figure is included in the revised paper. The third sentence could be reduced to "(see Section 5.3)" at the end of the caption.

Figure 7: See comments on the text regarding the cluster analysis. Add 'N' to each group in the plot if figure is included in revised paper. Also, the 'FA' plot y axis label needs to be changed to include 'ratio (FA)' at its end. The symbols should probably be removed and only numerical values shown on the y-axes on two of the plots.

Tables -

Table 1: The title should be simplified "Abbreviations of glacier names", delete "Used". Also, ensure that the plural 'glaciers' is used whenever the acronym is used in the text and/or figures (e.g. S27, S57, also S29, S58, others). Table 2: The title should be simplified and limited to the first part of text "Overview of SAR sensors and relevant specification". The second part should be a footnote to the table and specify which columns are relevant. Also, there needs to be a column that shows the spatial resolution of the SAR sensor. Table 3: The title should be limited to the first part of text. The second part should be a footnote to the table and specify which column is relevant. Also, 'Long-term' is not appropriate for a time period that is $\sim$20 years or less in some cases. Table 4: The title should be "Hypsometric Index and glacier basin category descriptions". The part "After Jiskoot et al. (2009)" should be a footnote to the table and should include the full range of HI values in the study (apparently much larger than for the Jiskoot study), including mean and standard deviation. The table could probably use at least a third column with the number of glaciers of each category. Table 5: Similarly, the title should be simplified and much of the header text moved to footnotes. Further, the table needs to be reformatted so that 'Sector' applies to not the first column (Parameters) but the subsequent four columns. Superscripts are missing for area rows. Consistent use of 'd' (italicized) or $\Delta$ for 'delta' would be appreciated through the paper. The mean velocity measurements should have a standard deviation as well given the larger uncertainties of some of the observations. This also applies to Table S1/S2.

Supplement "to:" -

Figures S1 to S74: As with Figure 2, it seems appropriate to ask for both velocity and area change data to be plotted at the same scales or a compelling argument advanced as to why this is not appropriate other than the effort involved. This would likely greatly reduce the size of the error bars that distract the eye in many instances and also clarify the 'patterns' more consistently. Paired and 'acronym' glaciers should be plural and with a lowercase 'g'. Table S1: See comment above, simplify the title, move parameter descriptions to footnotes or a header box as the editor prefers. Also ensure that the related text points to the correct table for specific parameters (Page 8, Line 15). Include a numbering scheme so it is obvious that there are far more 'West' glaciers than in any other category (split composite glaciers as required). Table S2: Add an appropriate title and move parameter descriptions to footnotes or a header box as the editor prefers. The $\Delta t$ values = 1d should be flagged in bold and the reader pointed to a specific text section of the paper and/or a footnote that explains why they need to be flagged.

---

## Referee Comment (RC2) · J. Wuite (Referee) · 5 May 2017

General Comments This paper provides an analysis of comprehensive satellite data sets to study changes in glacier area (over the period 1985-2015) and glacier surface velocity (1992-2014) on the northern Antarctic Peninsula, highlighting the complex temporal pattern of glacier retreat and ice flow dynamics in this region. This is a topic of great relevance for exploring factors that are controlling the varying response to climate change for the glaciers in this region. The hierarchical cluster analysis applied for the west coast glaciers is an inventive effort to provide insight into various flow controlling factors. I have, however, some major concerns that would need to be addressed, more specifically there appear to be some serious deficiencies regarding technical matters, as well as in the presentation of the work and discussion of the results, requiring in depth checks and major revisions and/or re-analysis of data. Referee #1 provides de-

tailed comments and suggestions for improvements regarding the presentation of the study sites, the description of methods, the presentation of results, as well as on the contents in discussion and conclusions sections.

Complementary to this careful and well-founded review, I am addressing below additional critical issues with emphasis on analysis, presentation and discussion of velocity data. I am focussing on the glaciers draining into the embayments of the former Larsen-A and Prince-Gustav-Channel (PGC) ice shelves because published data on these glaciers (based on various data sources) enable comparisons and checks of the various results.

The statement (Abstract P1L18, Results P8L11) "In 2014, the flow speed of the former ice shelf tributaries was 16.8% higher than at the beginning of the study period." implies that the outlet glaciers into the Larsen-A and PGC embayments are close to balance. This is in contradiction to other observations, showing prevailing large mass imbalance of these glaciers derived from geodetic data, and also to the much higher velocities compared to pre-collapse state. For example Rott et al. (2014) report for the period 2011 to 2013 a rate of mass depletion of $4.2 \pm 0.4$ Gt/year based on topographic data of the TanDEM-X satellite mission. The largest contribution is supplied by Drygalski Glacier (deficit $2.2 \pm 0.2$ Gt/year). Scambos et al. (2014) report a mass depletion of 5.6 Gt/year for the same area for the period 2003 to 2008. Analysis of TanDEM-X data from 2013 to 2015 show somewhat reduced mass deficit for these glaciers, but still a large imbalance (Rott et al., 2016), impossible to be maintained by a velocity that is only 16.8 % higher than in the pre-collapse state.

In Section 5.1 (Discussion East-Ice-Shelf) the authors discuss possible reasons for differences in velocities of glaciers in this sector compared to velocities reported by Rott et al. (2014). They argue that these differences are due to different approaches for reporting velocities (location in the centre of the glacier near the front vs. the median velocities at cross profiles close to the glacier fronts). Also, they are claiming that "equal temporal trends are observed in both studies" (P9L30). This is incorrect as evident by

comparing the velocity data in Table 2 of Rott et al. (2014) for several dates between November 1995 and November 2013. On Drygalski Glacier for example velocity near the centre of the 2013 front is reported to be 280% higher in November 2013 than in November 1995, and on Sjögren Glacier 410%. When referring to the pre-collapse state, the increase of velocity on Drygalski Glacier is even higher, because in November 1995 the lower glacier terminus had already accelerated significantly compared to pre-collapse state, as the time series of velocities starting in January 1993 shows (Rott et al., 2015). This acceleration 10 months after ice shelf collapse was already reported by Rott et al. (2002).

In order to clarify the discrepancies addressed above, it is necessary to better explain the methods used, check and revise the error estimates, and provide full traceability on the geographic location of the selected profiles for velocity retrieval and the epochs, and quantify the impact of using median values for quantifying velocities of glacier fronts for the different sensors. It would for example be very valuable to present cross profiles and/or profile time series used to derive the median values (and not only for East-ice-Shelf), in particular for the earlier pre-collapse estimates.

Regarding velocities, these are the main issues to be checked.

Cross sections: Cross section poorly defined and not well visible in Fig.1. Possibly define in supplement the coordinates of profile start/end.

Median value: How does median compare to velocity profiles of glacier cross section near the terminus. From which statistical sample is the median selected (A certain area close to the front? How far inland? Does it vary with sensor & patch size?). Impact of different sensor resolution, impact of different tracking patches to be checked. Table 2: Specify patch size on ground (metre), or specify pixel size (range, azimuth) for each sensor.

Error analysis (Section 3.2 and Supplement Table S2): The procedure applied for estimation of uncertainty seems to refer to the optimum case (smooth velocity fields and

good temporal stability of the surface features). A rather generic procedure is applied for specifying the uncertainty of velocity fields, whereas the uncertainty estimates should be provided for the single numbers (median values) presented in the paper. The velocity cross sections near calving fronts outlet often show strong velocity gradients. For these cases large tracking templates (in particular for the sensors with comparatively low spatial resolution) cause increased uncertainty in velocity. The constant factor (C= 0.2) for specifying the accuracy of the tracking algorithm (P5L26) is a value for the optimum case. McNabb et al. (2012) use C = 1-2. The actual values of C can be quite different, depending on time span, spatial resolution of the sensor, and temporal stability of the surface features. Many data sets were acquired during the summer period (Table S2), when surface melt and possibly also temporary refreeze cause changes of amplitude features, impairing the quality of correlation products. Another point to be reconsidered for the uncertainty estimate (Eq. 1, P5L25) is the oversampling factor z which reduces the uncertainty significantly if independence between (partial) overlapping template patches is assumed (which is not the case). This factor is not clearly explained in the paper.

The specified numbers of uncertainty for image coregistration (Table S2) apparently refer to full images, whereas the velocity data are derived from points near the coastline. Due to the lack of points on the ocean the coregistration accuracy near the coast lines might be impaired. The coregistration accuracy should be determined for the relevant image segments near the coast.

Additional comments: P1L12 'However...missing' -> the statement as written neglects previous research by various authors

P1L17 'Whereat ... trends' -> the statement as written implies that the ice shelf tributary glaciers also decelerated by something in the same order of 69% since 1992 which is not the case.

P8L10 'On...1.6%' -> this is a very surprising number and requires explanation as it

implies on average no change at all.

P13L13 'Group 3' -> I assume Group 4 is meant here.

Additional references (not cited in the manuscript):

Rott, H., Rack, W., Skvarca, P., and De Angelis, H.: Northern Larsen Ice Shelf, Antarctica: Further retreat after collapse, Ann. Glaciol., 34, 277– 282, 2002.

Rott, H., Wuite, J., Floricioiu, D., Scheiblauer, S., Nagler, T.: Synergy of TanDEM-X DEM differencing and input-output method for glacier monitoring. Proc. of 2015 IEEE Int. Geoscience and Remote Sensing Symp. (IGARSS 2015), Milan. Italy, 26-31 July 2015, pp. 5216-5219, 2015.

Rott, H., Wuite, J., Floricioiu, D., Johnson. E., Nagler, T., Scheiblauer. S.: Flow dynamics and mass balance of Antarctic Peninsula outlet glaciers from TanDEM-X and TerraSAR-X data time series. Paper presented at TerraSAR-X/TanDEM-X Science Team Meeting, 17-20 Oct. 2016, DLR Oberpfaffenhofen, Germany, 2016.

---

## Author Comment (AC1) · 1 Jul 2017

**First of all we want to thank the reviewer for constructive comments on our manuscript. All comments have been taken into account and a list of answers and actions undertaken is given below. Answers are indented and in bold face type and changes in manuscript are indented in** *blue*.

General Comments from the paper for the Authors –
The authors are to be appreciated for assembling an extensive array of illuminating data sets for a fairly large portion of the Antarctic Peninsula. By extending and expanding a previous study (Seehaus et al., EPSL 2015), it is clear that the hope was to illuminate many more glacial basins in this area of ongoing response to climate change. The use of the 5- parameter cluster analysis was a brave attempt to derive common themes across the area. Unfortunately, the complexities of the areas being investigated and the shorter/irregular nature of the velocity data appear to have confounded confident conclusions as the authors note on Page 14. A carefully edited paper with improved figures focusing on what is clearly known over the 1985 to 2015 area change period and the ~1992 to 2014 velocity data time frame will likely be publishable in TC.

Specific Comments from the text for the Authors –

Abstract
(Page 1 Line 9): The first three sentences should emphasize that this study will attempt a comprehensive analysis rather than 'other analyses have been lacking/missing' or too focused on the shelf collapse glaciers.

**Thank you for this advice. We changed/adjusted the wording of the respective sections to better emphasize that we are presenting a comprehensive study.**

*The climatic conditions along the northern Antarctic Peninsula have shown significant changes within the last 50 years. Here we present a comprehensive analysis of temporally and spatially detailed observations of the changes in ice dynamics along both the east and west coastlines of this region.*

Page 1 Line 13: The <65∘ latitude limit would include some of the Larsen B's major tributary glaciers so a less ambiguous way of defining the basins chosen for study is needed here and in the Introduction.

**We changed the definition of the study region (here and in the Introduction) in order to avoid ambiguity.**

*Abstract: <65° S along the west coast and north of the Seal Nunataks on the east coast*
*Introduction: (<65° S along the west coast and north of the Seal Nunataks on the east coast, Fig. 1b colored polygons)*

Page 1 Lines 15/16: Here and elsewhere the area changes need to be attributed to a specific year or by 'the end of the study period' or similar text. The Prince Gustav Channel ice shelf's northern limit is from what year? What is the standard deviation of the average velocity for those glaciers? 'Whereat' appears to be an archaic term.

**We added information on the observation periods for area change data and information on the data of the northern limit of Prince Gustav Channel Ice shelf extent.**
**We did not provide a standard deviation of the average velocity of those glaciers, since not all glaciers in this sector showed a similar trend. We intended to provide general information about the ice dynamic trend of each sector to the reader. More details of the individual glaciers are addressed in the Section "Discussion".**
**"Whereat" is replaced by "Whereas"**

*Glaciers on the east coast north of the former Prince Gustav Ice Shelf extent in 1986 receded by only 21.07 km² and decelerated by about 69 % on average (1985-2015).*

Page 1 Line 19: Similarly, what is the standard deviation of the average velocity?

**See comment above**

1.0 Introduction –

Page 1 Line 29: It seems important to have the word 'estimated' before mass balance given that IMBIE was a 'consensus' report.

**We replace "The authors reported..." by the "The authors estimated…."**

Page 2 Line 9: Here and elsewhere it seems more appropriate to put references chronologically from early to later.

**We appreciate the reviewer's comment, and did some editing of the manuscript according to his/her suggestion. However, at some sections it is more appropriate to keep the reference order for better storytelling.**

Page 2 Line 23: 'The collected observations reported in these studies suggest' rather than 'the observations suggest'...

**The sentence was adjusted according to the reviewer's suggestions**

Page 2 Line 28: 'methodologically' rather than 'methodically'

**We exchange the word according to the reviewer's comment**

2.0 Study Site

Page 3: This section MUST explain why a region that is only about 25% of the total AP was chosen for study. This should also include why sections of even the 330 km long area are excluded. Vague phrasing such as 'apart from those that are ice shelf tributaries, nearly all glaciers on the AP are marine-terminating' doesn't explain why much of the west coast + nearby major islands are excluded from this study.

**Thank you for the advice. We added a justification for the definition of the study region, and an explanation for why some sections were excluded. We did not include the nearby islands, since they are also not covered by most other studies and are not included in the basin definitions of IMBIE (Zwally Basins, Rignot Basins)**

*This facilitates the analyses of the long-term response (~20 years) of tributary glaciers to ice shelf disintegration at the former Larsen A and Prince-Gustav ice shelves on the east coast, the investigation of glaciers north of the former Prince-Gustav Ice Shelf, where no information on change in ice flow is currently available, and the comparison with temporal trends in ice dynamics along the west coast at the same latitude….*
*… Due to the sparse data coverage (fewer than three good quality velocity measurements), no*

*time series analysis of the glaciers at the northern tip of the AP or at some capes and peninsulas (e.g. Sobral Peninsula, Cape Longing) is possible.*

Page 3 Lines 3/4: 'high precipitation' and 'orographic barrier' could use numerical support. Does the whole selected study site act as the barrier or just the broad plateaus? Better graphics and labeling will help as noted further below.

**We added information about the typical height of the AP's mountain chain and the extreme rates of precipitation. According to the precipitation fields in van Wessem et al. 2016 the whole study region acts as a barrier.**
**Regarding the revision of the graphics see further down.**

*The AP's mountain chain (typically 1500-2000 m high) acts as an orographic barrier for the circumpolar westerly air streams leading to very high precipitation values on the west coast and on the plateau region of up to 5000 mm we yr −1, as well as frequent foehn type wind occurrences on the east coast (Cape et al., 2015, Marshall et al., 2006, van Wessem et al. 2016).*

Page 3 Line 11: Order the shelf areas chronologically.

**We followed the reviewer's suggestion.**

Page 3 Line 12: The Scambos et al. (2003) sentence needs to be balanced with a more recent reference such as Holland et al. (2015).

**As suggested we added a brief description of the findings of Holland et al.** (2015).

*A more recent study by Holland et al. (2015) discovered that significant thinning of the Larsen C Ice Shelf is caused by basal melting and that ungrounding from an ice rise and frontal recession could trigger its collapse.*

Page 3 Line 14: Insert 'frequently' before 'experiences melting'; other areas in Antarctica experience periodic melt events, especially a number of shelf areas (see just published work in Nature).

**Thank you for this advice. We added "frequently" as suggested.**

Page 3 Line 16: 'Narrow' seems an odd choice given the adjacent/excluded islands and smaller peninsulas and the broad plateaus (named elsewhere) in the study area.

**We removed "narrow"**

Page 3 Line 20: Making composite glaciers because they have 'laterally connected termini' needs to be better justified given the Seehaus et al. (2015) paper on DBE.

**According to the reviewer's advice a justification was added**

*Neighboring basins with coalescing ice flow at the termini are merged (many are already merged in the ADD 6.0), as the delineation of the individual glacier sections is not always possible and the width can vary temporally (due to changes in mass flux of the individual glaciers).*

Page 3 Line 22: 'Sparse data coverage' needs to be clarified.

**We added a statement to clarify the data coverage.**

*Due to the sparse data coverage (fewer than three good quality velocity measurements), no time*

*series analysis of the glaciers….*

Page 3 Line 24: The three sectors being defined by their 'different climatic settings' needs some additional justification. Some of the 'west' glaciers are shielded to some extent by large/high islands?

**The sectors were defined by
the climatic settings and drainage orientation → separation of east and west coast
and the former ice shelf extent → separation of the east coast in 2 sectors.
We adjusted the wording to be more clear.**

*Furthermore, the study region is divided into three sectors, taking into account the different climatic settings and drainage orientation as well as former ice shelf extent: ….*

3.0 Data and Methods –

3.1 Area changes –

Page 4 Line~1: I find sections that begin with no or abbreviated text frequently can be more clearly written. The 'Data and Methods' section needs an introductory paragraph that indicates why these specific data sets in the study are being utilized.

**An introduction for this section "Data and Methods" was added.**

*A large number of various remote sensing datasets are analyzed in order to obtain temporally and spatially detailed information on changes in ice dynamics in the study area. Glacier area changes are derived from satellite and aerial imagery. Repeat-pass Synthetic Aperture Radar (SAR) satellite acquisitions are used to compute surface velocity fields in order to obtain information on changes in glacier flow speed. Auxiliary data from sources such as a digital elevation model and glacier inventory are included in the further analyses and discussion of the results.*

Page 4 Lines 4/5: The two sentences can easily be merged with lines below them.

**We merged the two paragraphs.**

Page 4 Lines7/8: Distinguish sensors and satellites explicitly.

**Thank you for this comment. We removed "sensors"**

*…. using imagery from various satellites (e.g. Landsat, ERS) ….*

Page 4 Line 13: Given the retreat processes for the PG Channel, is limiting all of the glaciers to 1995 appropriate?

**Only one glacier (ADD ID: 2668) was affected by the gradual retreat of PGIS between 1985 and 1995. During this process, the PGIS retreated gradually along the frontal section of this glaciers (see Fig. 1). Therefore we think it is appropriate to refer the area changes to 1995.**

Page 4 Line 20: Were ratings of 4 and 5 not needed or was any such data discarded?

**There were no ice fronts mapped with such ratings within the study region. We changed the wording to be more clear**

*No ice fronts with reliability ratings of 4 and 5 are mapped in the study area.*

3.2 Surface velocities

Page 4 Line 24: Table 2 lacks SAR resolution information.

**See comments on Tables. This information was added.**

Page 4 Line 28: Does the mentioned masking eliminate glacier areas from having their full velocity patterns mapped? I think this and Line 30 could be clarified.

**The glacier areas are just masked out during the co-registration process (tracking was done on the full image), and the concatenation of images improves the co-registration in coastal areas, because more stable areas can be used to perform the co-registration. We adjusted the wording to be more clear.**

*In order to improve the co-registration of the image pairs, we mask out fast moving and unstable regions such as outlet glaciers and the sea during the co-registration processes. Furthermore, single SAR image tiles acquired during the same satellite flyover are concatenated in the along-track direction. This helps to further improve the co-registration in coastal regions (by including more stable areas in the co-registration process) but also simplifies the analysis of the final results as no mosaicking of the results is needed.*

Page 5 Line 7: Put a period after 'topography' and start the next sentence with 'The results are then geocoded...'

**We changed the structure according to the reviewer's advice.**

*…. incidence angle by the topography. The results are then geocoded, orthorectified and converted into …..*

Page 5 Lines 8-10: Some discussion of the limitations of the ASTER DEM is needed (this also potentially impacts the cluster analysis).

**We added a short summary of the quality of the ASTER DEM.**

*It has a mean elevation bias of -4 m (±25m RMSE) from ICESat data and horizontal accuracy better than 2 pixels. It is currently the best available digital elevation model of the Antarctic Peninsula.*

Page 5 Line 11: Are there no reference for the text in this paragraph? Is this a unique approach or are there any similar analyses? Does any of this approach depend on the native resolution of the SAR sensor utilized (add column in Table S2)?

**We added a reference regarding the tracking window size. However, usually only one tracking window size is used to calculate surface velocity fields. Due to the heterogeneous glacier flow, we applied different tracking window sizes and stacked them in order to improve the spatial coverage. Moreover, the window size depends on the SAR sensor resolution. We have changed the wording to be more precise. Regarding Table S2 see further down.**

*Depending on the displacement rate and resolution of the SAR sensor, the tracking window size needs to be adapted (de Lange et al. 2007).*

Page 6 Line 1: Please give the time frame for when the terminus profiles were defined. The phrase "taking into account temporal changes' suggests there is a broad range of profile times rather than a consistent time.

**For each glacier only one profile was defined. "Taking into account the temporal changes of the ice front" means, that the profile was defined behind the glacier front of the maximum retreat state. We changed the wording to be more clear.**

*A profile is defined (red lines in Fig. 1) close to the terminus of each glacier basin, behind the maximum retreat state of ice front position in the observation period.*

Page 6 Lines 2/3: The second sentence needs to be clarified.

**We change the wording to be more clear.**

*The results are visually inspected in order to remove unreliable measurements, based on the magnitude and direction of ice flow along the profiles. Datasets with partial profile coverage or large data gaps, as well as those with still remaining tracking errors, are rejected.*

Page 6 Lines 7-9: Change text to 'three or more' rather than 'more than two' and discuss if 3 observations in 10 years is adequate to 'classify' a basin as in Table 3 (with potential impact to the cluster analysis). Clarify if any of the '74 basins' were specifically excluded or does this apply only to the smaller areas that appear to be excluded (see Figure 5). Also, a plot showing the number of velocity observations as a function of (named) basin size with indications of latitude may be useful given the 'sparse' coverage of the northern Trinity Peninsula (Page 3 Line 22).

**We changed the wording of this section and added more detailed information.**
**The number of velocity measurements is listed in Table S1 and does not depend on the basin size, only on the spatial coverage by the SAR acquisitions. Therefore we did not perform a plot as suggested by the reviewer.**

*Only glaciers with three or more observations and an observation period of more than 10 years are considered in the categorization, resulting in 74 categorized glacier basins (colored polygons in Fig. 1b. There is a minimum of seven velocity measurements per categorized basin and the shortest observation period is 14.83 years (see Table S1; average number of velocity measurements per glacier is 33.8 and average observation period is 19.40 years).*

3.3 Catchment geometries and settings

Page 6 Lines 12-14: It seems appropriate to mention this analysis and how/why it differs from the earlier work led by Cook (Huber et al., 2017) http://www.earth-syst-sci- data.net/9/115/2017/essd-9-115-2017.pdf

**We added the reference to Huber et al. (2017) and mentioned the additional parameters that were derived. Why we derived this attributes is explained in the subsequent paragraphs.**

*In addition to glacier attributes derived by Huber et al. (2017), we calculated the Hypsometric Index and the ratio of the flux gate cross section divided by the glacier catchment area.*

Page 6 Line 17: Does accumulation increase with higher altitude on both sides? Does this apply mostly to the plateaus? Please clarify.

**The accumulation increases towards higher altitudes on both sides and this trend is not only limited to the plateaus (please see also Turner, 2002; van Wessem et al. 2016)**
**We have changed the wording to be more clear.**

*The climatic mass balance at the northern AP shows a strong spatial variability, with very high accumulation rates along the west coast, significantly lower values on the east coast and an increase towards higher altitudes along both coast lines (Turner, 2002; van Wessem et al. 2016).*

Page 6 Line 20: Add the Jiskoot et al. reference(s) here, not just in Table 4.

**We added the reference as suggested.**

Page 6 Lines 23-25: These two sentences need some expansion, perhaps to include the impact of the DEM's uncertainty and or any issues in defining the flux gates.
A plot would be better than just stating 'lower values indicate a channelized outflow'.

**In order to be clearer we have expanded the description of the FA ratios and the definition of the flux gates. We hope the reader will understand it without an additional plot.**

*In order to characterize the catchment shape, the ratios (FA) of the flux gate cross sections divided by the glacier catchment areas are calculated. The flux gates are defined along the profiles used for the glacier flow analysis (Section 3.2). Lower values of FA indicate a channelized outflow (narrowing towards the glacier front), whereas higher FA ratios imply a broadening of the glacier towards the calving front. Ice thickness at the flux gates is taken from the AP Bedmap dataset from Huss and Farinotti (2014).*

3.4 Cluster analysis –

Page 6 Line 26: Given that uncertainties in several of the five variables underlying the cluster analysis have not been explored, it is difficult to accept this approach. If this technique has been utilized practically in other similar glaciologic studies, please provide a reference(s).

**See answer to reviewer comment further down (Results)**

The standardization technique described (Page 7 Lines 2/3) could use some clarification and also a reference.

**We added a reference and extended the description of the standardization.**

*The variables are standardized in the traditional way of calculating their standard scores (also known as z-scores or normal scores). It is done by subtracting the variables mean value and dividing by its standard deviation (Miligan and Cooper, 1988).*

Page 7 Lines 4-7: This is rather unclear and this technique could very much use an analogy or similar technique to make it clearer to the reader what is actually being done to 'sort the basins' into groups with common parameters.

**We are sorry, we do not understand what the reviewer actually wants. We applied a standard statistical analyses method and the reader can find more details regarding this method in the references provided**.

4.0 Results

4.1 Area changes
Page 7 Line 8: This section also needs an introductory paragraph that summarizes what will be discussed in the sub sections.

**We do not think that an introduction is needed, since the sub sections are in the same structure as in the "Data and Methods" Section and the names of the sub sections clearly represent the topic of the sub section and what will be discussed in the sub sections.**

Page 7 Lines 10/11: Explain why these glaciers were chosen (all but one are from the 'West' region). It appears that they illustrate not just the three 'area change groups' but also the six 'velocity change groups' (Table 3). Is this correct? If using 'Figure' within a sentence, please spell it out. Use 'Fig.' as in (Fig. 3).

**The reviewer is right. The glaciers were selected in order to illustrate the three "area change groups" and the six "velocity change groups"(see Section 4.2 "Figure 2 shows by example**

the temporal evolution of the ice flow for each velocity change category"). We changed the wording of this section to be more clear.

It happened by chance that most glaciers are from the west coast.

According to the author guidelines of TC the abbreviation "Fig." should be used in running text.

> "The abbreviation "Fig." should be used when it appears in running text and should be followed by a number unless it comes at the beginning of a sentence, e.g.: "The results are depicted in Fig. 5. Figure 9 reveals that...".

*Area changes relative to the measurements in the epoch 1985-1989 of all observed glaciers are plotted in Fig. S1-S74 (supplement). The glaciers are classified in three groups based on the latest area change measurements, which are illustrated in Fig. 2:....*

Page 7 Line 16: Assume you mean '238 km2'. Also, see comments on Figure 4 that seem designed to greatly accentuate the '2.2%' loss between 1985 and 2015.

**See answer to comment on Fig. 4.**

Page 7 Line 17: You could usefully add the individual loss % values here.

**Thank you for this advice. We added the the area loss values (in %) for each sector.**

*.... of which the area loss by 5.7% at sector "East-Ice-Shelves" clearly dominates. The glaciers in sector "West" and "East" recessed by 0.2% and 1.4%, respectively.*

4.2 Surface velocities

Page 7 Line 22: 'A total of' 282 etc...

**We appreciate this comment. We replaced "In total" by "A total".**

Page 7 Lines 23-26: Are the 'average' uncertainties of the velocity fields meaningful given the array of different sensors used? The text suggests not. Perhaps the average uncertainty of each sensor (and its standard deviation) could be stated instead and also added to Table 2? This information is too deeply buried in Table S2.

**We appreciate this advice and have added the average uncertainty of each sensor to Table 2. We kept the average value of all datasets in the text and included a reference to Table 2.**

*The average total uncertainty of the velocity fields amounts to $0.08 \pm 0.07$ m d$^{-1}$ and the values for each SAR sensor are provided in Table 2.*

Page 7 Lines 26-28: If these data are unreliable, explain how they were or were not used in the study and all the Figures S1-74? This is unclear.

**The ERS datasets with 1 day repetition frequency are not necessarily unreliable or of bad quality. The total intensity tracking accuracies of these datasets was obtained by only considering the mismatch of the coregistration, since the applied approach to estimate the accuracy of the tracking algorithm is strongly biased by the very short temporal baseline of these data sets. This applies only to seven datasets out of 382. We rephrased this section to be more clear.**

*ERS image pairs with time intervals of one day have very large estimated tracking uncertainties, biased by the very short temporal baselines. Therefore, only the errors caused by the mismatch of the coregistration are considered in the total error computations of the seven ERS tracking results with one day temporal baselines.*

Also, was there any attempt to do curve fitting through the data that passed the quality criteria? Given the range of velocity (and area change) axes used, I find it very difficult to visually assess (Page 8 Lines 1-3) the Table 3 categories.

**We attempted to do curve fitting in order to automatically derive the velocity change categories but we were not satisfied with the results. Therefore, we did a manual classification. A statement to clarify this was added in Section 3.2.**

*The glaciers are manually classified in six categories according to the temporal evolution of the ice flow speeds (see Table 3), since automatic classification attempts did not succeed.*

Page 8 Lines 6/7: The 'local clustering' should be identified even if it is explored further in the Discussion section (see comments on location indicators of Figures).

**We added a location reference for the local clustering.**

*…a local clustering of accelerating glaciers can be observed at Wilhelmina Bay.*

Page 8 Line 9: Table S2 should be S1 and there is an error in one of the subscripts and 'd' should apparently be Δ, here. Also see comments on Table 5.

**Thank you for this advice. We have corrected it accordingly. Regarding "d" and "Δ" see further down.**

Page 8 Line 13: You might as well give the longest period for velocity and also the standard deviation.

**According to your advice we added information on the longest period and the standard deviation.**

*The shortest observation period is 14.83 years at DBC31 Glacier, the longest observation period is 21.99 years at TPE31 and Sjögren glaciers and on average velocity changes are analyzed over a period of 19.40 years ($\sigma$ = 1.97 years).*

4.3 Catchment geometries and settings

Page 8 Lines 15/16: The HI values are in Table S1, not S2, and appear to vary quite a bit more than in Jiskoot et al. (2009).

**We corrected the references to the tables in the supplement. We applied the same classification as Jiskoot et al. (2009), in order to be consistent/comparable with/to another study that also applied it at the Antarctic Peninsula (Davis et al. (2012)**

Figure 3 is very difficult to read for both velocity and HI categories. Given that this section is 'Results', perhaps the unmapped areas should be mentioned.

**See answer to comment on Figure 3 further down.**

4.4 Cluster analysis

Page 8 Lines 19-21: In part due to the preceding text (Lines 16/17) "No clear distribution pattern can be identified, reflecting the heterogeneous topography of the AP.", my concerns about the cluster analysis remain unresolved. The limited text here, regard less of Section 5.3, seems to emphasize an uncertain result.

**"No clear distribution pattern can be identified, reflecting the heterogeneous topography of the AP." refers to the HI, which does not need to have a clear distribution pattern.**

**Because it is hard to manually identify clear distribution patterns of individual glacier variables along the west coast or identify relations between the variables, the cluster analysis approach was applied and lead in our opinion to reasonable results. See also answer to reviewer comment on the cluster analyses further down.**

5.0 Discussion

Page 8 Line 25: The result that all glaciers on the east coast receded should be clarified to state 'since 1985'. Does Davies et al. (2012) overlap in terms of area with this study?

**We added "since 1985". The study area of Davies et al. (2012) overlaps with our study area on Trinity Peninsula.**

*Only glaciers along the west coast showed stable or advancing calving fronts and all glaciers on the east coast receded since 1985. This heterogeneous area change pattern was also observed by Davies et al. (2012) on western Trinity Peninsula.*

Page 8 Line 27: Superscript for area is missing.

**We are sorry, but we could not identify the missing superscript, since no variable is mentioned in this section.**

Page 9 Lines 3/4: This is very difficult to ascertain from Figure 4c and seems to be an overreach of the results, the text seems speculative. See the small deviations in the area change trend for the 1995-2005 'blocks'.

**We are aware, that the recession in 1995-2005 was just slightly increased and that the relation between the ice shelf break-up and the increased retreat rates is just a speculation. We adjusted the wording to better emphasis that it is just a slight increase in the retreat rates and that our explanation is speculative.**

*Moreover, slightly increased recession is also found in the time period (1995-2005, Fig. 4) at sector "East". Davies et al. (2012) and Hulbe et al. (2004) supposed that the disintegration of an ice shelf affects the local climate. The air temperatures would rise due to the presence of more ice free water in summers. This might explain the slightly higher retreat rates at sector "East".*

Page 9 Lines 6-8: Seehaus et al. (2015, Figure 3) shows warming for Marambio for 1998 to 2006 not 1997 to 2007. That time range appears to be from the Oliva et al. (2017) broader analysis who shows the locations of all the available records and their variation over a longer time frame. And it isn't clear what "Unfortunately, no temperature records are available in sector "East" covering this period." means as all the temperature data appears to be from outside this paper's study area.

**We corrected the time specification and included only information from Oliva et al. (2017). "Unfortunately….." means, that no temperature data recorded within this sector. We changes the wording to be more clear.**

*At Base Marambio, ~100 km east of this sector, approximately 2°C higher mean annual air temperatures were recorded in the period 1996-2005 as compared to the period 1986-1995 (Oliva et al., 2017). Unfortunately, no temperature data recorded within sector "East" is available covering this period that could be used to validate this theory.*

Page 9 Lines 11-13: Clarify that the 'frames' correspond to ESA conventions for identifying ERS coverage and that frame 4923 covers 'the central and much of the northern part of sector "West"'.

**Thank you for this advice. We changed the wording accordingly.**

*Pritchard and Vaughan (2007) reported an increase in mean flow rate of 7.8% in frame 4923 (the*

*central and much of the northern part of sector "West") and 15.2% in frame 4941 (the southern part of sector "West") for the period 1992-2005 (frame numbers correspond to European Space Agency convention for identifying ERS coverage).*

Page 9 Lines 14-19: Is this really a 'discovery' since you go on to show that the 'discrepancy' has a logical explanation?

**We replaced "discovered" by "derived"**

*However, for the same observation period we derived a mean increase in flow velocity by 18.9 % in sector "West", which is an approximately 1.6 times higher acceleration.*

5.1 East ice shelf 'sector' (no reason to capitalize)

Page 9 Line 22: Given Figures S1-13 describe sector "East" why start with the ice shelf loss area basins detailed in S14-26? Please add the date or dates that detail when the basins lost the ice shelf area in front of them (e.g. paragraphs on Page 10).

**We appreciate this advice and exchanged Section 5.1 and 5.2 ("East" and "East-Ice-Shelf") in order to match the order of Figures S1-S74. We added information on the dates of the loss of the ice shelf area in front of the glaciers.**

*In the sector "East-Ice-Shelf" the tributary glaciers in the Larsen A embayment ("2558", Arron Icefall, DBE, Drygalski, LAB2, LAB32, TPE61 and TPE62; Fig. S14, S17, S19-S22, S25 and S26) and Sjögren-Inlet (Boydell, Sjögren and TPE114; Fig. S18, S23 and S24) lost the downstream ice shelves in 1995....*
*In the 1980s, Prince Gustav Ice Shelf gradually retreated (see Fig. 1) and "2668" Glacier (Fig. S15) has not been buttressed by the ice shelf since the early 1990s..*
*The ice shelf in Larsen Inlet disintegrated in 1987-1988 and earliest velocity measurements are obtained in 1993. Therefore, a potential peak in the flow speed after ice shelf break-up cannot be detected at APPE glaciers.*

Page 9 Line 26: Here and elsewhere, hyphens are not needed for 'Larsen-A/B'.

**We appreciate this advice and removed the hyphens throughout the manuscript.**

Page 9 Line 30: It is good that you can resolve differences due solely to methodology but please clarify what 'equal temporal trends' means in this context.

**"equal temporal trends" means that comparable temporal changes in glacier flow speed were observed in both studies. We adjusted the wording to be more clear.**

*The different approaches result in different absolute values, but comparable temporal trends in glacier flow speeds are observed in both studies.*

Page 10 Lines 2-5: It is difficult to conclude that the stated variation in the behavior of these basins shows they are still 'adjusting to the new boundary conditions' as opposed to responding to purely localized forces acting on them. On Line 3, do you mean 'medial' as opposed to the statistical 'median'?

**We supposed that this glaciers show a prolonged response to the ice shelf break-up caused by the local settings. We extended the discussion to be more clear and removed "median".**

*At "2558", Boydell, DBE and Sjögren glaciers the deceleration is ongoing and Boydell and DBE glaciers still show increased flow speeds at the glacier fronts. We suppose that these tributary glaciers show a prolonged response to ice shelf disintegration, caused by local settings (e.g. bedrock topography or fjord geometry), and are still adjusting to the new boundary conditions, as*

*suggested by Seehaus et al. (2015, 2016).*

Page 10 Lines 6-15: Some interesting details are discussed here but they seem to be overly specific rather than useful indicators. The discussion of Pyke Glacier vs the composite APPE basin, including Pyke, suggests a concern about this analysis combining individual flow systems in composite basins. Does averaging over multiple smaller glaciers blur a discernable signal? The lack of sufficient temporal coverage of the available velocity data appears to be a common issue here.

**The observations by Rott et al. (2014) at Pyke Glacier show the same trend as our measurements. We changed the wording to better emphasize this.**
**The reviewer is right the temporal coverage at Larsen Inlet (APPE) and "2668" Glaciers is a limiting factor. However, there is no data available to obtain reasonable information about glacier flow speeds at this glaciers for the 1980s. A statement on this issue was added.**

*As for "2668" Glacier no sufficient cloud free coverage by Landsat imagery is available which facilitates the computation of surface velocities for the 1980s. The ice flow at APPE glaciers shows a nearly stable trend with short term variations in the order of 0.2-0.5 m d$^{-1}$ between 1993 and 2014. Rott et al., (2014) also found nearly constant flow velocities at Pyke Glacier.*

5.2 East 'sector' (see comment above on order of discussion)

Page 10 Lines 20-28: It would seem that a good bit of this discussion might fit better in the introductory section. The specific figures in the Supplement would be useful to point out for the named basins. Depending on whether you choose to interpret Turner's or Oliva's figures allows you to vary the point when cooling began in the 21st century, what specific date do you prefer?

**According to the reviewer's advice we moved some parts to the "Introduction". The numbers of the specific figures in the Supplement were added in this section and section 5.1.**
**Oliva et al. (2017) stated "Our results also indicate that the cooling initiated in 1998/1999 has been most significant in the N and NE of the AP..." which is nearly similar to Turner et al. (2016) "… to show an absence of regional warming since the late 1990s." Therefore, we decided to use the phrase "However, a recent cooling trend on the AP was revealed by Oliva et al. (2017) and Turner et al. (2016) since the late 1990s." (now in the Introduction)**

Page 11 Lines 1-4: Does the analysis of Oliva et al. (2017) not allow more precision than 'before earliest velocity measurements'? Does the area change time series going back to 1985 (in this sector) not provide additional insight?

**We appreciate this advice and referenced our discussion to the date from Oliva et al. (2017) and Skvarca et al. (1998).**
**The area change time series shows a frontal stabilization after 1985, but every glacier started to maintain its front positions at different periods.**

*Hence, we assume that the initial recessions of the glaciers in sector "East" were forced by the warming observed by Oliva et al. (2017) and Skvarca et al. (1998) since the 1970s. Therefore, this initial frontal destabilization and retreat led to high flow speeds at the beginning of our ice dynamics time series (earliest velocity measurements from 1992) and the subsequently observed frontal stabilization (after 1985) caused the deceleration of the ice flow.*

Page 11 Lines 8-10: Please be more specific as to what/how the visual imagery was used to identify the 'bump'.

**We identified some small rock outcrops that indicate a shallow bedrock bump. The wording was adjusted to be more precise.**

*No nunatak is present at the terminus, but small rock outcrops, indicating a shallow bedrock bump, are identified north of the center of the ice front by visual inspection of optical satellite imagery.*

Page 11 Lines 13-19: Some of this material should be in the introductory material and the analysis seems speculative given the stated need for more observations. Page 11: Also highlights the difficulties in reading Figure 3 for specific locations (or interpreting symbols) even after magnification of the pdf.

**We would like to keep this material in this section, since the description of the surge cycle is quite specific for only these 2 glaciers.**
**We adjusted the wording to emphasize that it is speculative, but we would like keep to his sections in the paper, since it provides a motivation to further continue the observation of glaciers in this region.**

*Diplock and Victory glaciers (Fig. S5 and S13) show a decrease of flow speed during retreat followed by an acceleration combined with frontal advance. Surge-type glaciers, found for example in Alaska (tidewater) (Motyka and Truffer, 2007; Walker and Zenone, 1988) or Karakoram (land terminating) (Rankl et al., 2014), show similar behavior. They are characterized by episodically rapid down-wasting, resulting in a frontal acceleration and strong advance. Regarding tidewater glaciers the advance can be strongly compensated by increased calving rates in deepwater in front of the glacier. It is therefore possible that these glaciers may have experienced a surge cycle in our observation period; however, a longer time series analysis is necessary to prove this hypothesis.*

5.3 West 'sector'

Page 11 Line 24: See previous comment on Turner vs Oliva temperature studies.

*See answer to previous comment*

Page 11 Lines 24/25: Clarify what is meant by 'constant trend'? Do you mean in both space and time? If so, can the ocean temperature differences be reconciled?

**The reviewer is right. The climatic trends on the AP are not constant in space and time. We have changed the wording to be clearer and added a statement on the link between ocean temperatures, sea ice concentration and the deceleration of the warming.**

*However, Cook et al. (2016) reported cool ocean temperatures along the north-western AP for the period 1945-2009, and an absence of the atmospheric warming, especially pronounced at the northern AP, since the turn of the millennium was found by Oliva et al. (2017) and Turner et al. (2016), which correlates with an increase of sea ice concentration and the cool ocean temperatures at the northern AP.*

Page 11 Lines 25/26: Does 'southern part' apply to both West and East or only 'West"? What abut the coastline makes it 'fractal' and does that aid understanding? Clarify 'These' factors lead (cause?)...

**"southern part" refers to sector "West" and "fractal" was replaced by "jagged". We hope to be more clear now. "This factors lead" was replaced by "These factors cause"**

*Moreover the glacier geometries differ strongly, and especially in the southern part of sector "West", the coastline is more jagged. These factors cause the heterogeneous pattern of area and flow speed changes in sector "West" as compared to the eastern sectors.*

Page 11 Lines 28/29: Clarify if the 12 glaciers studied by Kunz et al. (2012) included basins and years overlapping this study. Which 'authors' are being referred to here?

**We included information about the glaciers located in our study area, analyzed by Kunz et**

**al. (2012). We referred to the "authors" of Benn et al. (2007). We change the wording to be more clear.**

*Kunz et al. (2012) observed thinning at the glacier termini along the western AP, by analyzing airborne and spaceborne stereo imagery in the period 1947-2010. Two of the twelve studied glaciers are located within our study area; Leonardo Glacier (1968-2010) and Rozier Glacier (1968-2010). …*
*However, Benn et al. (2007) also….*

Page 11 Line 31: The fact that fjord and glacier geometries may be uncertain should probably be mentioned here, especially for smaller basins.

**According to the reviewer's advice we added a statement on this issue**

*However, Benn et al. (2007) also point out that changes in ice thickness do not necessarily affect the ice flow and that calving front positions and ice dynamics are strongly dependent on the fjord and glacier geometries, derived from modeling results which have higher uncertainties especially for smaller basins.*

Page 12 to Page 13 Line 13: As indicated above, I find the cluster analysis to be of uncertain value and will refrain from further comment on it. Other reviewers and/or the Editor can decide if it should remain in the paper.

**We would like to keep the cluster analysis in the paper, since it significantly helped to categorize the glaciers along the west coast and led to reasonable results(in our opinion). This work was also presented at the EGU General Assembly 2017 and we received positive feedback by the community also regarding the cluster analysis. Therefore, we think this approach might be a useful tool for the analysis of long-term chances in ice dynamics in combination with glacier geometry parameters at other study sites. Time series calculations are becoming more feasible with better temporal and spatial coverage of the cryosphere by the current sensors like TerraSAR-X/TanDEM-X and Sentinel-1A/B and future missions.**

6.0 Conclusions

Page 13 Lines 15/16: The usage of 'northwestern' to define the study area is quite imprecise as is the usage of 'north of 65∘S' as was previously commented.

**We adjusted the wording to be more precise.**

*Our analysis expands on previous work on ice dynamic changes along the west coast of AP between TPE8 and Bagshawe-Grubb Glacier, both in regard to temporal coverage and analysis methods. It also spatially extends previous work on changes in ice dynamics along the east coast between Eyrie Bay and the Seal Nunataks.*

Page 13 Line 18: The 'dynamics' were observed most clearly only during ~1992 to 2014 through the repeated velocity observations. This text should be clarified.

**According to the reviewer's advice, we added information on the study periods for each method.**

*The spatially and temporally detailed analysis of changes in ice flow speeds (1992-2014) and ice front positions (1985-2015) reveal varying temporal trends in glacier dynamics along the northern AP.*

Page 13 Line 19:Clarify if 'significantly higher' is simply due to differences in the methodology relative to Pritchard and Vaughan (2007) for the same period. If so, should this simply say 'higher' velocities were observed?

**As mentioned in the "Discussion", differences could be caused by the different methodologies. We removed "significantly".**

Page 13 Line 22: Be clear that all 'East' glacier fronts retreated relative to 1985 (or 1995 after shelf losses).

**We adjusted the wording to be more clear.**

*On the east side all glacier fronts retreated in the study period (relative to 1985), with highest retreat rates observed at former tributaries of the Prince Gustav, Larsen Inlet and Larsen A ice shelves (relative to the year of ice shelf disintegration)..*

Page 13 Line 28: The 'cooling since 2000' depends on how you read the Seehaus et al. (2015), Turner et al. (2016) or Oliva et al. (2017) analyses. Mid-2000s seems to be a more reasonable number for much of your study area.

**According to the reviewer's suggestion we change the wording.**

*Based on the observed warming trend since the 1960s and the subsequent cooling since the mid-2000s in the northern AP ….*

Page 14 Lines 3-5: See previous concerns about how well the cluster analysis with 5 variables can discriminate across such a broad swath of the western AP. It appears that this study needs to include additional parameters rather than attributing groups to basin geometry alone (as is clearly indicted in their next paragraph).

**We tried to include a broad variety of data, but also to keep the focus on the remote sensing part and the ice dynamics analysis. Therefore, we gave the suggestion in the next paragraph how the results of this study could be used to further investigate the processes at the Antarctic Peninsula.**

Figures -

Figure 1: This figure needs to be redesigned with a small Antarctic map in the corner of the 'general peninsula region' map showing the specific study area on the ~1300 km long Antarctic Peninsula. Major landscape features and adjacent water bodies should be clearly labeled on both of the panels especially (c) if mentioned in the text (e.g. Bruce and Detroit plateaus, James Ross Island, Charcot, Charlotte, Andvord, Wilhelmina bays, not just on Figure 5). The LIMA credit is incorrect, should be USGS,NASA, BAS, NSF. Further, the scale of the third panel should be sufficient to clearly discern ice front positions and related color choices of lines (shades of orange, red on red?) may need to be revised. It is appropriate to specify in the caption why ADD 6.0 is being used for glacier fronts instead of the data from the study. Also, areas mostly or totally excluded from the study (e.g. Trinity, Longing, Sobral peninsulas) should be identified here. Also, Bellingshausen Sea is misspelled and inaccurately located.

**We appreciate the reviewer's comment and revised the figure. Additional labels of landscape features and water bodies were included as far as possible, in order to keep the figure clear. The color of some layers were also revised. We used only the "coastline dataset" from ADD 6.0 to display the ice shelf extents. We adjusted the caption to be more clear and corrected the LIMA credit. The regions/glaciers which were excluded from the study are not included in the polygons indicating the three sectors.**

Figure 2: The caption seems to need to include "for each velocity change category (see Table 3)." And it does seem odd that there is only one example that is not from 'West'. As with S1 to S74, it seems appropriate to ask for both velocity and area change data to be plotted at the same scales or a compelling argument advanced as to why this is not more appropriate. This would likely greatly reduce the size of the error bars that distract the eye in many instances. Also, as mentioned

in text comments, was curve fitting of the velocity data attempted?

**We revised the caption. Regarding the selection of glaciers and the curve fitting see answer to review comment further up. Of course some error bars of e.g. area changes (e.g. in Fig. 1b) seems to be quite large compared to error bars of glaciers with large area changes (e.g. Fig. 1c). However, due to the large diversity and variability of glacier velocities and area changes, we do not want to used fixed scales for all glaciers.**

Figure 3: Even after magnification of the pdf, Figure 3 is difficult to read for locations and symbols and these also cannot be searched. This makes the text discussion of small features very difficult. Also, see above for the need for locations mentioned in the text to be labeled. Close inspection reveals that smaller areas appear to be excluded along with the larger Sobral and Longing peninsula regions and such areas need to be mapped/explained (also see text comments). Also, discerning the color scale for the HI outlines of each basin are challenging especially where they overlap.

**As for Fig. 1 we added additional labels landscape features and water bodies. Regarding the size and scale of Fig.3, we tried different labeling options and increased the size of the glacier labels. We could ask the editor if it might be possible to spread it over 2 pages in order to magnify it.**
**Excluded area are not covered by the HI polygons. See also answer to comment on Fig.1.**
**We changed the HI outline color scale and removed the overlap by using buffered polygons.**

Figure 4: It is positive to note that this figure's caption points out that the left y-axis (not the right one) has different scaling for each of the plots. It is appropriate for the area change y-axis to be consistently scale as that allows the reader to quickly detect the magnitude of change from region to region. It is not clear why the left y-axis doesn't start at zero in all cases and use some distinct maximum thousands value to clearly show that the changes are still small relative to the total area in each sector, especially for 'all glaciers'. The editor may wish to provide guidance here.

**The reviewer is right. The area changes are quite small compared to the total area, but this is usually the case, since glacier area changes are mostly in the order of a few %. We did not start the left y-axis at 0 because we want present the temporal trend of area change, which can not be seen, if we start the y-axis at 0. If it is OK for the reviewer's and the editor we would like to keep the figure as it is. Another option could be, that we just show the "Area change".**

Figure 5: See comments on the text regarding the cluster analysis. The caption needs to clarify that all polygons in the figure are colored (see previous comment on over lapping basin outlines) but that the sectors are (somewhat) defined with three colors. Also, 'dA' should apparently be ΔA. This figure finally provides some location pointers to the Trinity Peninsula (partial) and the bays missing from Figure 1 but, oddly, doesn't label any of the glaciers? This figure also highlights that 3 of the 'composite' basins are quite large (APPE, CLM, and DBE) and a fourth (SBG) is much larger than some of the investigated 'west' basins. This makes one wonder why they could not be similarly subdivided. "Laterally- connected' is not clearly explained in the text as the reason to composite these basins (how much of each glacier?).

**According to the reviewer's advices, we revised the caption to be clearer, and removed the overlap of the sector outlines. Moreover, we added location and glacier labels.**
**Regarding the "composite" basins please see answer to reviewer comment further up.**
**Regarding "d" vs. "Δ" see answer to reviewer comment on Table 5.**

Figure 6: See comments on the text regarding the cluster analysis. Add numbers for each cluster group to each red box if the figure is included in the revised paper. The third sentence could be reduced to "(see Section 5.3)" at the end of the caption.

**We appreciate the comment and revised the caption and added numbers for each cluster group**

Figure 7: See comments on the text regarding the cluster analysis. Add 'N' to each group in the plot if figure is included in revised paper. Also, the 'FA' plot y axis label needs to be changed to include 'ratio (FA)' at its end. The symbols should probably be removed and only numerical values shown on the y-axes on two of the plots.

**We adjusted the figure according to the reviewer's suggestions. We would like to keep the symbols on the y-axes (velocity change and FA). We guess it helps the reader to a better understand/interpret the graphs. Moreover we added numerical values to the y-axes of the FA plot.**

Tables

Table 1: The title should be simplified "Abbreviations of glacier names", delete "Used". Also, ensure that the plural 'glaciers' is used whenever the acronym is used in the text and/or figures (e.g. S27, S57, also S29, S58, others).

**Thank you for this comment, we revised the title and checked the manuscript for the plural "glaciers".**

Table 2: The title should be simplified and limited to the first part of text "Overview of SAR sensors and relevant specification". The second part should be a footnote to the table and specify which columns are relevant. Also, there needs to be a column that shows the spatial resolution of the SAR sensor.

**According to the reviewer's advices (see also above) we added a column that shows the nominal spatial resolution and the mean uncertainty of the tracking results.**

Table 3: The title should be limited to the first part of text. The second part should be a footnote to the table and specify which column is relevant. Also, 'Long-term' is not appropriate for a time period that is ~20 years or less in some cases.

**We appreciate the comment and put the second part of the title in the footnote. "Long-term" was replaced by "general".**

Table 4: The title should be "Hypsometric Index and glacier basin category descriptions". The part "After Jiskoot et al. (2009)" should be a footnote to the table and should include the full range of HI values in the study (apparently much larger than for the Jiskoot study), including mean and standard deviation. The table could probably use at least a third column with the number of glaciers of each category.

**We revised the table following the reviewer's suggestion and added a column listing the number of glaciers of each category. We decided to not show the range of HI values, mean and standard deviation in the footnotes, but added this information in the results (Section 4.3).**

*The HI values range between -4.6 and 9.1 (mean: 0.88, σ: 2.10).*

Table 5:

Similarly, the title should be simplified and much of the header text moved to footnotes. Further, the table needs to be reformatted so that 'Sector' applies to not the first column (Parameters) but the subsequent four columns. Superscripts are missing for area rows. Consistent use of 'd' (italicized) or Δ for 'delta' would be appreciated through the paper. The mean velocity measurements should have a standard deviation as well given the larger uncertainties of some of the observations. This

also applies to Table S1/S2.

**According to the reviewer's suggestion we moved most of the title to footnotes, and re-formatted the table to better indicate that sector applies to the subsequent for columns. We are sorry, but we do not understand which superscripts are missing for area rows. we used subscripts to indicate the observation intervals. We checked the paper and used "d" for "delta" through the paper. Table S1/S2 were also revised accordingly.**

Supplement "to:" -
Figures S1 to S74: As with Figure 2, it seems appropriate to ask for both velocity and area change data to be plotted at the same scales or a compelling argument advanced  as to why this is not appropriate other than the effort involved. This would likely greatly reduce the size of the error bars that distract the eye in many instances and also clarify the 'patterns' more consistently. Paired and 'acronym' glaciers should be plural and with a lowercase 'g'.

**Please see answer to reviewer comments further up (regarding the scale). We revised the glacier labels according to the reviewer's advice.**

Table S1: See comment above, simplify the title, move parameter descriptions to footnotes or a header box as the editor prefers. Also ensure that the related text points to the correct table for specific parameters (Page 8, Line 15). Include a numbering scheme so it is obvious that there are far more 'West' glaciers than in any other category (split composite glaciers as required).

**According to the reviewer's suggestion we simplified the title and checked the cross references in the text. We decided to not include a numbering scheme in Table S1, but to add a row to Table 5 which shows the number of glaciers in each sector.**

Table S2: Add an appropriate title and move parameter descriptions to footnotes or a header box as the editor prefers. The $\Delta t$ values = 1d should be flagged in bold and the reader pointed to a specific text section of the paper and/or a footnote that explains why they need to be flagged.

**We moved the parameter description to the footnotes. We added "*" to highlight the dt values =1 and linked the footnote to the text section in the paper.**

---

## Author Comment (AC2) · 1 Jul 2017

**J. Wuite Referee #2**

**First of all we want to thank the reviewer for constructive comments on our manuscript. All comments have been taken into account and a list of answers and actions undertaken is given below. Answers are indented and in bold face type and changes in manuscript are indented in blue.**

General Comments This paper provides an analysis of comprehensive satellite data sets to study changes in glacier area (over the period 1985-2015) and glacier surface velocity (1992-2014) on the northern Antarctic Peninsula, highlighting the complex temporal pattern of glacier retreat and ice flow dynamics in this region. This is a topic of great relevance for exploring factors that are controlling the varying response to climate change for the glaciers in this region. The hierarchical cluster analysis applied for the west coast glaciers is an inventive effort to provide insight into various flow controlling factors. I have, however, some major concerns that would need to be addressed, more specifically there appear to be some serious deficiencies regarding technical matters, as well as in the presentation of the work and discussion of the results, requiring in depth checks and major revisions and/or re-analysis of data.

Referee #1 provides detailed comments and suggestions for improvements regarding the presentation of the study sites, the description of methods, the presentation of results, as well as on the contents in discussion and conclusions sections. Complementary to this careful and well-founded review, I am addressing below additional critical issues with emphasis on analysis, presentation and discussion of velocity data. I am focusing on the glaciers draining into the embayments of the former Larsen-A and Prince-Gustav-Channel (PGC) ice shelves because published data on these glaciers (based on various data sources) enable comparisons and checks of the various results.

The statement (Abstract P1L18, Results P8L11) "In 2014, the flow speed of the former ice shelf tributaries was 16.8% higher than at the beginning of the study period." implies that the outlet glaciers into the Larsen-A and PGC embayments are close to balance. This is in contradiction to other observations, showing prevailing large mass imbalance of these glaciers derived from geodetic data, and also to the much higher velocities compared to pre-collapse state. For example Rott et al. (2014) report for the period 2011 to 2013 a rate of mass depletion of 4.2±0.4 Gt/year based on topographic data of the TanDEM-X satellite mission. The largest contribution is supplied by Drygalski Glacier (deficit 2.2±0.2 Gt/year). Scambos et al. (2014) report a mass depletion of 5.6 Gt/year for the same area for the period 2003 to 2008. Analysis of TanDEM-X data from 2013 to 2015 show somewhat reduced mass deficit for these glaciers, but still a large imbalance (Rott et al., 2016), impossible to be maintained by a velocity that is only 16.8 % higher than in the precollapse state.

**We understand the reviewer's concerns, but here we present the average of the changes in flow speed of all glaciers in this sector (ignoring the different size or mass discharge). Therefore, this value is biased by the small glaciers, which were not so strongly affected by the disintegration of the ice shelf disintegrated, compared to the larger more inertially glaciers (like Drygalski, DBE, Boydell, Sjögen glaciers). Moreover, the changes in flow speeds do not directly reflect the changes in ice discharge, which also strongly depends on the spatial distribution of the ice thickness at the flux gate. However, a rough approximation of the ice discharge (using our median velocity values and average ice thickness information from Huss and Farinotti 2014 along the profiles) leads to an ice discharge of**

**~9.4 Gt/a in this sector, whereof 16.8% (our observed average increase of flow speed) correspond to 1.6 Gt/a. This number is lower than the values reported by Rott et al. (2014) or Scambos et al. (2014), but at a comparable level (assuming no change in SMB). Moreover, a recent study by Hogg et al. (2017) points out that only ice discharge across the grounding line can not necessarily explain the deflation. They attribute 35% of the imbalance to increased ice discharge, and hypothesized that ocean driven melting may have forced the dynamical thinning of the glaciers at Western Palmer Land.**
**The presented study has put the focus on the temporal changes in ice dynamics. However, a detailed study on the changes in ice discharge and mass balance using the "Input-Output" Method or "Flux-Gate" Method is currently in preparation and We change the wording in Section 4.2 to provided additional information to the reader regarding the average flow speed changes.**

*The presented average flow speed change values are based on the observed changes of all glaciers in the respective sector (Table S1), ignoring the different size of the individual glaciers.*

In Section 5.1 (Discussion East-Ice-Shelf) the authors discuss possible reasons for differences in velocities of glaciers in this sector compared to velocities reported by Rott et al. (2014). They argue that these differences are due to different approaches for reporting velocities (location in the centre of the glacier near the front vs. the median velocities at cross profiles close to the glacier fronts). Also, they are claiming that "equal temporal trends are observed in both studies" (P9L30). This is incorrect as evident by comparing the velocity data in Table 2 of Rott et al. (2014) for several dates between November 1995 and November 2013. On Drygalski Glacier for example velocity near the centre of the 2013 front is reported to be 280% higher in November 2013 than in November 1995, and on Sjögren Glacier 410%. When referring to the pre-collapse state, the increase of velocity on Drygalski Glacier is even higher, because in November 1995 the lower glacier terminus had already accelerated significantly compared to precollapse state, as the time series of velocities starting in January 1993 shows (Rott et al., 2015). This acceleration 10 months after ice shelf collapse was already reported by Rott et al. (2002). In order to clarify the discrepancies addressed above, it is necessary to better explain the methods used, check and revise the error estimates, and provide full traceability on the geographic location of the selected profiles for velocity retrieval and the epochs, and quantify the impact of using median values for quantifying velocities of glacier fronts for the different sensors. It would for example be very valuable to present cross profiles and/or profile time series used to derive the median values (and not only for East-ice-Shelf), in particular for the earlier pre-collapse estimates.

**The difference between measuring velocities at one point at the center of the terminus compared to averaging along a profile at the terminus were already reported in Seehaus et al. (2015) The authors also observed deviations between velocity measurements at the center of the terminus and measurements along profiles across the terminus. They attribute it to the the fact, that highest flow speeds are found at the center of terminus, but this maximum values are suppressed by averaging across the terminus. We added the lit. reference at the description of this issue. See also the figure below. It shows the surface velocity across the terminus of Drygalski Glacier at similar dates as shown in Rott et al. (2015). We observed comparable velocities at the center of Drygalski Glacier's terminus as reported by Rott et al. (2015) (unfortunately the position of the profile is not plotted/provided in this paper). This plot is also added to the "Supplement" and an statement on this issue is added in the manuscript. Regarding the error estimates, geographic location see answers to reviewer comments below.**

*The velocities reported by Rott et al. (2014) at Sjögren, Pyke, Edgeworth and Drygalski glaciers are generally higher than our findings. The authors measured the velocities at locations near the center of the glacier fronts, where the ice flow velocities are typically highest, whereas we measured the median velocities at cross profiles close to the glacier fronts (Seehaus et al. 2015). The different approaches result in different absolute values, but comparable temporal trends in glacier flow speeds are observed by both author groups. For example Rott et al. (2015) presented surface velocity measured along a central flow line of Drygalski Glacier. Figure S75 shows our*

*surface velocity measurements across the terminus of Drygalski Glacier. Both studies show comparable values at the center of the terminus.*

[Figure]

*Figure S75: Surface velocity across the terminus of Drygalski Glacier*

Regarding velocities, these are the main issues to be checked.
-Cross sections: Cross section poorly defined and not well visible in Fig.1. Possibly define in supplement the coordinates of profile start/end.

**Since some of the profiles are kinked and not straight (coordinates of more than 2 points needs to be provided), we could provide the profiles as shapefiles in the supplement or upload them on PANGAEA (which is planed for some of the obtained results after acceptance of this manuscript)**
**(as the editor prefers).**

-Median value: How does median compare to velocity profiles of glacier cross section near the terminus. From which statistical sample is the median selected (A certain area close to the front? How far inland? Does it vary with sensor & patch size?). Impact of different sensor resolution, impact of different tracking patches to be checked.

**The median is calculated based on the velocity measurements extracted along the profiles, including a 200 m buffer zone around each profile for all sensors. We changed the wording to be clearer.**

*In order to reduce the number of data gaps along the profile due to pixel size data voids in the velocity fields, the velocity data is extracted within a buffer zone of 200 m around the profiles.*

**The results of the different patch sizes were stacked ("The results of each image pair are stacked by starting with the results of smallest tracking window size and filling the gaps with the results of the next biggest tracking window size.") and the step size of the tracking process and the geocoding parameters were adjusted in order to obtain velocity field with**

**100 m pixel spacing for all sensor. Information on this issue was added.**

*The results are then geocoded, orthorectified and converted into velocity fields (with 100m pixel spacing for all sensors) by means of the time span between the SAR acquisitions.*

-Table 2: Specify patch size on ground (metre), or specify pixel size (range, azimuth) for each sensor.

**As also requested by reviewer #1 we added a the nominal ground resolution of the sensors**

-Error analysis (Section 3.2 and Supplement Table S2): The procedure applied for estimation of uncertainty seems to refer to the optimum case (smooth velocity fields and good temporal stability of the surface features). A rather generic procedure is applied for specifying the uncertainty of velocity fields, whereas the uncertainty estimates should be provided for the single numbers (median values) presented in the paper. The velocity cross sections near calving fronts outlet often show strong velocity gradients. For these cases large tracking templates (in particular for the sensors with comparatively low spatial resolution) cause increased uncertainty in velocity. The constant factor (C= 0.2) for specifying the accuracy of the tracking algorithm (P5L26) is a value for the optimum case. McNabb et al. (2012) use C = 1-2. The actual values of C can be quite different, depending on time span, spatial resolution of the sensor, and temporal stability of the surface features. Many data sets were acquired during the summer period (Table S2), when surface melt and possibly also temporary refreeze cause changes of amplitude features, impairing the quality of correlation products. Another point to be reconsidered for the uncertainty estimate (Eq. 1, P5L25) is the oversampling factor z which reduces the uncertainty significantly if independence between (partial) overlapping template patches is assumed (which is not the case). This factor is not clearly explained in the paper.

**The reviewer is right, that the error due to the tracking algorithm depends on the template size as well as other parameters and can vary spatially, which is difficult to assess. McNabb et al. (2012)'s analysis is mainly based on optical satellite data and using manual feature tracking, which explains the different value of C. We selected the value of C=0.2 according to personal communication with the software provider (GAMMA Remote Sensing). Rignot et al. (2011) applied a value of "1/128th" (0.0078125) for speckle tracking, which has typically an order of magnitude higher accuracies as intensity tracking (Gray et al. 1998; DOI 10.1080/07038992.2001.10854936). Therefore, our estimate is quite conservative. Wuite et al. (2015) estimated the uncertainty to 0.2 to 0.3 pixels and their resulting uncertainties are quite similar to our estimations (Table 2).**
**We applied an oversampling factor of two (Seehaus et al. 2015) for the tracking windows (tracking chips), which is suggested by the software provider in order to increase the accuracy of the tracking process. We added a statement on this issue to be clearer. Finally there is to say, that in several studies in this region no error estimates are provided for glacier velocities. Thus, to provide just "a rather generic" error estimation, addressing the ascertainable values, is better than no estimation (in our opinion).**

*The accuracy of the tracking algorithm is estimated to be 0.2 pixels and an oversampling factor z=2 is applied to tracking patches in order to improve the accuracy of the tracking process.*

-The specified numbers of uncertainty for image coregistration (Table S2) apparently refer to full images, whereas the velocity data are derived from points near the coastline. Due to the lack of points on the ocean the coregistration accuracy near the coast lines might be impaired. The coregistration accuracy should be determined for the relevant image segments near the coast.

**we stated in the paper (Section 3.2, P5) "single SAR image tiles acquired during the same satellite flyover are concatenated in the along-track direction.... to further improve the co-registration". At coastal regions this helps to increase the area land masses used for the co-registration. Moreover, the still remaining co-registration offset is measured on stable ground close to the coastline (glacier fronts) where most of the rock out crops and**

**nunataks are found. We adjusted the wording to be more precise.**

*Furthermore, single SAR image tiles acquired during the same satellite flyover are concatenated in the along-track direction. This helps to further improve the co-registration in coastal regions (by including more stable areas in the co-registration process) but also simplifies the analysis of the final results as no mosaicking of the results is needed......*
*The mismatch of the coregistration $\sigma_v{}^C$ is quantified by measuring the displacement on stable reference areas close to the coast line, such as rock outcrops and nunataks.*

Additional comments:
P1L12 'However...missing' -> the statement as written neglects previous research by various authors

**This section was revised and this statement removed. See also comments by reviewer #1**

*The climatic conditions along the northern Antarctic Peninsula have shown significant changes within the last 50 years. Therefore we present a comprehensive analysis of temporally and spatially detailed observations of the changes in ice dynamics along both the east and west coastlines of this region.....*

P1L17 'Whereat … trends' -> the statement as written implies that the ice shelf tributary glaciers also decelerated by something in the same order of 69% since 1992 which is not the case.

**We revised this section to be clearer.**

*A dramatic acceleration after ice shelf disintegration with a subsequent deceleration is observed at most former ice shelf tributaries on the east coast, combined with a significant frontal retreat.*

P8L10 'On ...1.6%' -> this is a very surprising number and requires explanation as it implies on average no change at all.

**This is because some glaciers/sectors showed significant increase, whereas others showed a decrease. We added a statement, to point out that the changes in the individual sectors were significant. In the next lines the change values of the individual sectors are presented.**

*On average the ice flow in the study region increased by 1.6%, but the glaciers in the individual sectors showed on average significant change. Along the west coast an average acceleration by 41.5% occurred and the former ice shelf tributaries on the east coast accelerated by 16.8%. In the sector "East" the glaciers decelerated resulting in a mean velocity change of -69%. The presented average flow speed change values are based on the observed changes of all glaciers in the respective sector (Table S1), ignoring the different size of the individual glaciers.*

P13L13 'Group 3' -> I assume Group 4 is meant here.

**Thank you for this advice. The reviewer is right. We meant group 4 and corrected it.**

---

## Referee Report (RR1)

**Re-Review of The Cryosphere Discuss., doi:10.5194/tc-2017-50, 2017**
**Title: Changes in glacier dynamics in the northern Antarctic Peninsula since 1985**
**Authors: Thorsten Seehaus, Alison Cook, Aline B. Silva, Matthias Braun**

**General Comments**

The comments here concern the second revised version of the manuscript by Seehaus et al. on changes in glacier dynamics in the northern Antarctic Peninsula since 1985. The authors have made substantial revisions to the original and revised manuscript in response to the reviewer's comments. These revisions include clarifications, corrections, additional references, as well as an added section and new figures in the supplemental material. The additional scrutiny appears to have led to significant changes in some of the reported velocity changes (individual glaciers & regions) and also to the discovery of errors in some of the figures/tables.

I appreciate the efforts by the authors to clarify their adopted approach, but would like to point out here that I am still rather unconvinced about the exact implementation of the across flow median velocity values in the presented analysis and my earlier worries on several issues still stand. The profiles presented in the manuscript's supplement illustrate to some degree the difficulties and problems that arise. If, as the authors mention in their reply document (1st reply), the erroneous/incomplete profiles were not used for the analysis there is little reason to include them, but in the way it is written currently it looks like all depicted profiles passed the quality check (Pg. 9 Ln. 12-15). Therefore, I mention here specifically: unrealistic fluctuations appear to occur along some of the profiles (e.g. Fig S155) and some profiles are incomplete (and were apparently not rejected as mentioned in the text – Pg. 6 Ln. 29/Pg. 7 Ln. 1). A crucial issue, however, is the treatment of velocity in the shear margins, some profiles seem to be cut off rather abruptly, some are smoother and go down to zero velocity, some appear to cover more glaciers and include intermediate (stationary?) areas, some have very high values at the margins (or do not seem to include the margin – e.g. S150). The sensor capabilities and to a certain degree the algorithm settings largely determine how well these margins can be captured. One should therefore be cautious when interpreting extracted values as they could reflect sensor limitations instead of real velocity changes.

That said, in concordance with my main wish in the previous review rounds, at least the methods are now better documented and the inclusion of ice velocity cross profiles in concordance with calculated velocity median values (in the supplement) provide some means of traceability. I likewise welcome the inclusion of additional ice velocity maps for this reason as it shows the source material of some of the numbers and its potential or limitation. Also, the reported ice velocity changes for Drygalski (which I took as a primary example in my previous review) has been revised considerably.

A few more **specific comments** for consideration:

- The IV profiles are extracted from close to the terminus, but it is not mentioned in the text from which year nor is it very clear from fig 1.

- In the conclusion it is now mentioned "Upcoming sensor [SIC] probably facilitate the region wide measurement of recent surface elevation, since current estimates have got only partial coverage or have got some issues due to the complex topography of the AP. ". Just a thought, what type of sensor is going to overcome the partial coverage and issues due to complex topography, or is this wishful thinking (in which case 'hopefully' is more apt than 'probably')?

- As mentioned above, please clarify whether profiles depicted in supp. material passed the quality check and if they were used or not used in the analysis.

**Grammar**

Pg. 2 Ln. 27: ") The" → missing point

Pg. 7 Ln. 17&18: on average

Pg. 12 Ln. 2: van der Veen

Pg. 13 Ln. 8: observeD

Sup Mat.

Pg. 2 2nd paragraph: 2256 measurementS

Pg. 2 3rd paragraph:  using obtained by the second approach → seems like a word is missing here

Pg. 2 3rd paragraph: This mismatch does not influencE the subsequent

Pg. 2 5th paragraph: 'is little' → is small

---

## Author Response (AR2)

**Response to the Reviewer comments on**

"Changes in glacier dynamics at the northern Antarctic Peninsula since 1985"

By Thorsten Seehaus et al.

**Anonymous Referee #1 Received: 04 August 2017**

First of all we want to thank the reviewer for constructive comments on our manuscript. All comments have been taken into account and a list of answers and undertaken actions is given below. Answers are indented and in bold face type and changes in manuscript are indented in *italic*.

 Because this study confines itself to a rather specific portion of the whole Antarctic Peninsula 'region', the authors are cautioned to avoid vague terms such as 'region' (see Abstract, page 1, line 11) when referring to only the very northern part of the contiguous AP and none of the adjacent islands.

**We appreciate the comment and carefully revised the usage of "region" throughout the manuscript.**

2) 'Seal Nunataks' is used in the abstract to provide a geographic limit to the study area (page 1 line 13) but should refer to Fig. 1 as it is not labeled on the other maps.

**It is quite uncommon to refer to figures in the abstract. Therefore, we do not want to do it here. In the description of the study site in the Introduction we refer to Fig. 1.**

3) The authors are similarly cautioned to be specific when referring to the 'study period' given that the title states that the study period is 'since 1985' but the velocity data covers a much shorter period, especially for some glaciers, and some regions only had 'area changes' since ~1995.

**We revised the usage of "study period" and year numbers provided according to the reviewer's comment.**

4) Similar to #1, the authors must be clear that the warming that has been observed on the 'Antarctic Peninsula' must be limited to the 'northern' AP and its outlying islands given where the majority of the data sets are located (page 1 line 26).

**We rephrased this sentence to be clearer.**

**During the last century, the northern Antarctic Peninsula (AP) and its outlying islands haves undergone significant warming (Turner et al., 2005), leading to substantial glaciological changes.**

5) The use of a single reference to document the loss of ice shelves, page 2 line 10, appears a little uncharitable to the many researches who have contributed a great deal of insight and analysis to this particular topic along and across the AP. Even though additional references are added below, the list still seems inadequate and does not give the reader a useful sense of what shelf areas were lost and when.

**We revised this sentence and added reference and year numbers.**

Numerous ice shelves along the AP have retreated widely (e.g. Müller, Wilkins, Wordie) or disintegrated in recent decades (e.g. Braun and Humbert, 2009; Cook and Vaughan, 2010; Doake and Vaughan, 1991; Rack et al., 1998; Rack and Rott, 2003; Wendt et al., 2010)

6) Related to #5, close inspection of Figure 1 helps somewhat with this issue but the placement of the 'keys' on top of the area changes is unhelpful. Further, dashed lines would help differentiate some of the colors that are very similar. Oddly, it is only by closely reading the text that one realizes that the northern end of the contiguous AP was studied but lacked enough velocity data to be included (those basins are included in Figure 5 for area change). It isn't clear why the basins are cropped off.

According to the reviewer's advice, we revised the "keys" and replaced some solid lines by dashed or dotted lines of Fig.1. We revised Fig. 5 and removed the "cropped off" basins at the northern end, which are not included in our study area.

7) On page 2, line 23, given the number of studies in this section, it isn't clear which 'authors' are being discussed.

**We rephrased this sentence to be clearer.**

Observations by Kunz et al. (2012) support this supposition. They analyzed surface elevation changes of 12 glaciers on the western AP based on stereoscopic digital elevation models (DEM) over the period 1947-2010.

8) Similar to #1, the term 'north-eastern' appears to mean the northern AP's eastern glacier basins on page 2, line 25. Also, 'as a consequence to' on line 26 is unclear.

**This section was revised to be clearer.**

Frontal surface lowering was found at all glaciers. Whereas, glacier area-wide surface lowering of ice shelf tributaries was observed along the north-eastern AP by various author groups (e.g. Berthier et al., 2012; Rott et al., 2014; Scambos et al., 2014).

9) The phrase 'not homogeneous' (page 2, line 28) seems rather obvious. One assumes that the authors mean this in both space and time and this should be linked back to the different times that glaciers in the area have become marine- as opposed to shelf-terminating.

**We would like to keep the sentence as it is, since all details are already provided above.**

10) The phrase "Previous studies often only cover a specific period or region, or focus on one particular aspect of glacier change." On page 2 lines 29-30 also appears to be uncharitable to other researchers who worked with what data sets were available to them at the time. In time, this study will also be superseded by new data and techniques so I suggest rephrasing the intended meaning.

**The reviewer is right and we revised this section accordingly.**

Previous studies often cover a specific period or area, or focus on one particular aspect of glacier change. By now, the availability of remote sensing data time series data and other data sets in this region facilitates the comprehensive analysis of glacier change.

11) Similar to previous concerns, 'northern-most' (page 3, line 5) is quite imprecise and requires all outlying islands of the AP to be ignored. The analysis of Huber et al. 2017 makes it quite clear how much area of the AP has been excluded in this study.

**We refer her to the whole AP, since it is the introductory sentence to the section "Study site". Further down, we specify our study area more precisely. See also answer to comment #16.**

12) Please review all superscripts for consistency (page 3, line 8). Also, should 'yr' or 'a' -1 be used?

**Thank you for this advice. We revised the superscripts and replaced "yr" by "a" to be consistent.**

13) Given their prominence on the figures and also as the source areas for many glaciers, the 'plateau regions' should be referenced (page 3, line 12).

**We added the names of the plateau regions.**

Aside from those that are ice shelf tributaries, almost all glaciers on the AP are marine terminating, and the majority of the glacier catchments extend up to the high elevation plateau regions (north to south: Laclavère, Louis Philippe, Detroit, Herbert, Foster, Forbidden, Bruce, Avery, Hemimont, Dyer).

14) The Prince Gustav (no hyphen) shelf had left the channel before 1995 according to ADD and also the USGS I-2600A map (page 3, line 15).

**The ADD and USGS I-2600A map shows the extent of Prince Gustav ice shelf in 1993 (across the channel) and 1995 (the channel is open). However, the date of the ice shelf break up cannot be determined based on these maps.**

The ice shelf quite likely broke up in early 1995 according to Rott et al. 1996: "A National Oceanic and Atmospheric Administration (NOAA) image from 9 January 1995 shows that although the channel was open, major icebergs were close to the previous ice shelf, which indicated that the shelf had broken only a few days earlier."

**To be clearer we added a literature reference. All hyphens between "Prince" and "Gustav" were removed.**

15) The phrase 'long-term' is not appropriate for ~20+ year records (see previous comment on 'study period').

**We removed the phrase "long-term" in this sentence.**

16) Given the use of percentages in a number of places in the paper is seems appropriate to contrast the '11,000 km2' area (page 3, line 26) against the whole area of the AP including its islands. Also 'altitudes' should probably be 'elevations'.

**We appreciate the reviewer's suggestion and added information on the percentage of AP area coverage by our study area. Altitudes were replaced by elevations.**

The study area covers an area of ~11,000 km2 (~11% of the whole AP including islands, Cook et al., 2014; Huber et al., 2017) with elevations stretching from sea level up to 2220 m.

17) There are too many 'word choices' to point them all out but 'substituted' should be 'replaced' (page 3, line 29).

**According to the reviewer's advice, we changed the wording.**

18) Please revise capitalization on page 4, lines 8-9. Also line 26.

**We revised the capitalization.**

19) One assumes that the '100m' (please check spacing for units throughout the text) pixel spacing requires resampling given the native resolutions now listed in Table 2 as 'nominal ground resolution'\* although one has to wonder if this is also due to incidence angle on the AP's 'jagged' topography. Please clarify.

The resolution of the velocity fields depends on the combination of the SAR data resolution in slant-range-geometry (not the ground range resolution in Table 2) and the tracking stepsize (Table 2) of the tracking process. We applied step-sizes in order to achieve approx. 100 m pixel spacing of the velocity results. The geocoding and orthorectification algorithms include a resampling process.

Only for the high resolution sensors (TerraSAR-X and TanDEM-X), tracking results of 50 m resolution (based on SAR resolution and step-size) were calculated and resampled to 100 m resolution, in order to use the same resolution for all sensors.

We added information on this issue in the manuscript.

As described in the manuscript, we correct for effects on the local incidence angle by the AP's "jagged" topography. (see also next answer to next comment)

20) A requested 'summary' of the uncertainty in the ASTER DEM has been inserted on page 5, line 25. Unfortunately, a quick examination of Cook et al. (2012) Figure 5, the paper that is the source of the elevation bias, shows that the bias number is itself biased towards the much more extensive ICESat coverage far to the south (to 70°S) of this paper's study area. With the vast majority of the ICESat to DEM comparisons apparently coming from lower slope areas of the AP, it is highly unlikely that the given numbers apply to the 63-65°S portion of the AP. Further, no attempt appears to have been made to show if the bias varies as a function of slope in the study area. One has to wonder why this was not done given the importance of the DEM to the geometric aspects of the study as well as the potential to impact the velocity data. In short, a much more realistic assessment of the DEM's accuracy in the study area is not yet available. See also Huber et al's (2017) more conservative estimates.

We appreciate the reviewer's comment and added information on Huber et al's estimates in the manuscript. Our velocity measurements are all located close to the calving front where the slope of the glacier is typically quite low and the quality of the DEM high. Therefore, we assume the impact on the velocity data to be insignificant (see also Seehaus et al. 2015, supplemental material, and Scambos et al. 2014).

It is currently the best available digital elevation model of the Antarctic Peninsula. It has a mean elevation bias of -4 m (±25 m RMSE) from ICESat data and horizontal accuracy better than 2 pixels. However, Huber et al. 2017 estimated the uncertainty to be ±50 m, since it varies regionally. Velocity data is analyzed close to the calving front (see further down) where the slope of the glaciers at the AP is typically quite low. Thus, the impact of the DEM accuracy on the velocity fields is insignificant (see Seehaus et al., 2015 supplemental material).

21) The text on page 6, line 15 needs clarification: "close to the terminus of each glacier basin, behind the maximum retreat state of ice front position in the observation period". It is pretty clear what is meant but this phrasing is awkward.

**We rephrased this sentence.**

A profile is defined (red lines in Fig. 1) close to the terminus of each glacier basin, considering the maximum retreat state of ice front position in the observation period.

22) It would be useful to quantify what is meant by 'very high' and 'significantly lower' on page 6, lines 6-8. Also, does the mass input depend at all on basin orientation or only on the hypsometry and elevation (lines 8-9).

As requested, we added CMB values for the east and west coast. The reviewer is right; the mass input also depends on the basin orientation (east coast vs. west coast, as stated in the same section of the manuscript). We changed the wording to be more precise.

The climatic mass balance at the northern AP shows a strong spatial variability, with very high accumulation rates along the west coast (3769 mm we a-1 in average in sector "West", 1992-2014,

RACMO2.3), significantly lower values on the east coast (1119 mm we a-1 in average in sector "East", 1992-2014, RACMO2,3) and an increase towards higher altitudes along both coast lines (Turner, 2002; van Wessem et al. 2016). Consequently, the mass input depends on the basin orientation (east coast or west coast), elevation range and the hypsometry.

23) The sentence at the end of page 7 and top of page 8 still needs some sort of analogy or further explanation to make it accessible to the average reader. Even the most dedicated readers will be unlikely to dive as deep as appears to be needed to see what is being done to the raw input numbers. In addition, it still seems relevant to point to any other study in glaciology that has derived useful results from a related 'sorting' technique.

**According to the reviewer's suggestion we rephrased this sentence and added literature reference of studies, which used the cluster analyses to group glaciers based on a set of variables.**

This is a proven method to classify glaciers based on a set of variables (Lai and Huang, 1989; Sagredo and Lowell, 2012)

. . . . .

At the start, the most similar glaciers (samples) are grouped. The resulting clusters are iteratively joined based on their similarities (distances) until only one cluster is left, resulting in a dendrogram (see Section 4.4).

24) Area changes for the 'ice shelf loss glaciers' needs to be separately called out on page 8, line 5, given that are 'after 1995' not since '1985-1989'. Interestingly, this date range suggests the variability in temporal resolution of the area change values going into the cluster analysis, also an issue with the temporally variable velocity values as shown in S1-74.

**We appreciate this comment and revised the date specification in this section and also in Table S1 and Table 5. We are sorry, but we do not understand the second part of the comment.**

25) See previous comment on 'percentages' but the actual area change values should be given on page 8, lines 11-13.

**We added the actual area change values.**

In total, 238.81 km2 of glacier area was lost in the survey areas in the period 1985-2015, which corresponds to a relative loss of 2.2%. All sectors show glacier area loss (Table 5), of which the area loss by 5.7% (208.59 km2) at sector "East-Ice-Shelves" clearly dominates. The glaciers in sector "West" and "East" recessed by 0.2% (9.14 km2) and 1.4% (21.07 km2), respectively

26) The use of the word 'trends' when referring to what are simply plots of the velocity data is problematic for a number of reasons. In some case, there simply isn't enough data to even estimate a trend (e.g. S4, S6, S8) and even when there is more data, it is often so unevenly temporally sampled (S27, S28, S56) as to be impossible to discern a trend (or as is discussed later, a pattern). Further, there is also a great deal of concern that signal vs noise (apparent pattern vs error bars) is not being taken into account in the results section of this paper.

**We appreciate this comment and revised the usage of the word "trends" and "pattern" throughout the manuscript.**

We added some information regarding the signal to noise ratio in Section 4.2.

In order to analyze the quality of obtained velocity change signal, the ratio of the maximum measured velocity difference (maximum velocity minus minimum velocity) divided by the average error of the velocity measurements is calculated for each glacier. An average signal to noise ratio of 14.6 is found. At three glaciers (DGC14, DGC22 and Orel) a signal to noise ratio of less than 2 is observed. These glaciers are characterized as "stable", which justifies the low signal to noise

ratio.

27) Please clarify "On average the ice flow in the study region increased by 1.6%, but the glaciers in the individual sectors showed on average significant change." on page 9, lines 7-8. Please give velocity change values as well as % so as to save to save the reader from having to also read the supplement.

**We changed the wording to be clearer ad added the velocity change values.**

On average the ice flow in the whole studied area increased by 1.6% (0.008 m/d), but the average changes of the individual sectors are more pronounced. Along the west coast an average acceleration by 41.5% (0.177 m/d) occurred and the former ice shelf tributaries on the east coast accelerated by 16.8% (0.081 m/d). In the sector "East" the glaciers decelerated resulting in a mean velocity change of -69% (-0.688 m/d).

28) The sentence "The presented average flow speed change values are based on the observed changes of all glaciers in the respective sector (Table S1), ignoring the different size of the individual glaciers." leads to wondering if size classes in each sector might provide more insight?

**In our opinion an average value weighted by the flux-gate size rather the catchment size could be applied. However, the ice thickness data at the AP (e.g. from Huss and Farinotti 2014) also has got significant uncertainties, which might bias the calculation. Therefore, we decided to keep it simple and comprehensible.**

29) Please change the term 'shrinkage' throughout as it suggests a 3-dimensional change in volume rather than a change in area alone (page 10, line 3).

**We revised the usage of "shrinkage" throughout the manuscript. Thank you for this advice.**

30) The term 'theory' should be replaced with 'hypothesis' and given how speculative this is, given the large distances to the nearest met stations, I think the editor should consider excising this speculative section. Note, unpublished (as of yet) studies are suggesting that the 'cooling' was a problematic sampling of the longer-term record now that 2016 and 2017 data is becoming available.

**We replaced "theory" with "hypothesis". Well, in this section we are not talking about the cooling trend in the 21st century. We talk about a warming between 1986 and 2005. However, if the editor wants to excise this section, we can remove it.**

31) The 'clear positive velocity trend' (page 10, lines 16-17) does not appear to be supported by the figures in S57-74 in my opinion. See the previous comments on 'trend' and signal vs noise'.

**We rephrased this sentence, in order to not over-interpret the results and observed trends. Regarding signal vs noise; see answer to comment 26.**

This spatial trend corresponds to our observations, since most of the glaciers which accelerated are located at the southern end of sector "West".

32) Please explain how Skvarca et al. (1998) saw a cooling trend in the 21st century (page 11, line 6).

**Sorry, this reference was displaced.**

33) The 'peaked' trend for TPE10 Glacier (page 11, line 15, is a very clear example of overinterpreting insufficient data as is Aitkenhead Glacier (S3).

**The reviewer is right TPE10 Glacier has got only a few velocity measurements. However, we**

added the graphs obtained from the 2nd velocity measuring approach (measured at maximum ice thickness at the terminus profiles) in the revised version of the supplement. These results also support the velocity change classification of TPE10 Glacier as "peaked". As well for Aitkenhead Glacier, the same classification is obtained by both measuring approaches.

**We added a reference to the respective figures in the manuscript.**

34) Perhaps I don't understand what 'frontal advance' means (page 11, line 22) given 5-year averaging but the glaciers mentioned both show continued area losses in the referenced plots.

**Both glaciers accelerated and gained area (frontal advance) in the period 2010-2015. We added data specification to be clearer.**

Diplock and Victory glaciers (Fig. S5 and S13) show a decrease of flow speed during retreat (1995-2010) followed by an acceleration combined with frontal advance (2010-2015).

35) Please give the 'comparable values' for the two analyses on page 12, line 11.

**We added the observed values of both analyses exemplarily for two dates.**

36) A 'potential peak in flow speed' (page 12, line 25) appears to be unnecessary speculation as it 'cannot be detected'.

**According to the reviewer's suggestion we removed this sentence**

37) Pyke Glacier is within the APPE group so a reference to Table 1 seems useful (page 12, line 28).

**We appreciate this advice and added a reference to Table 1.**

**Rott et al., (2014) also found nearly constant flow velocities at Pyke Glacier (part of the APPE basin, Table 1).**

38) The opening paragraph on page 13 seems muddled. Also, the term 'jagged' for the western coastline seems inadequate given the point is to give context for the 'heterogeneous' changes observed in sector 'West'. See the previous review for a concern about the potential orographic impact of large islands on the west side as well as an earlier comment here on slope aspect vs solely elevation/hypsometry.

**We rephrased and condensed this paragraph to be clearer. Moreover, we added a statement on the potential impact of the islands on the climate on AP's mainland.**

We are sorry, but we do not exactly understand the part regarding "slope aspect vs. solely elevation/hypsometry". The general aspect of the glaciers (east coast vs. west coast) is taken into account by dividing the study site in sectors. In our opinion, to use the average slope of the glaciers at the AP as a geometry variable is less reasonable, since the glacier tongues are usually dynamically separated by steep cliffs from the plateau section. Therefore, we decided to use maximum elevation and Hypsometric Index since the accumulation (which influences the ice dynamics) shows a significant elevation gradient (especially along the west coast).

The glacier geometries differ strongly along the west coast. In the southern part of sector "West" the shoreline is more ragged and islands are near the coast. An impact of the islands on the climatic conditions at the AP mainland's coastline (e.g. orographic barrier) is not obvious (visual inspection of RACMO2.3 5.5 km grid cell model results (Van Wessem et al., 2016)). However, the climatic conditions on the AP show strong spatial and temporal variability (see Section 1.2 and 3.3). These factors cause the heterogeneous spatial pattern of area and flow speed changes in sector "West" as compared to the eastern sectors.

39) Group1 'needs some space' (page 13, line 21).

**Corrected**

40) There is something missing around 'dissimilarities' on page 13, line 18, perhaps 'matrix analysis' would be appropriate to add here.

**We changed the wording to be clearer.**

The large number of glaciers in this sector is analyzed by means of a hierarchical cluster analysis (Section 3.4) and assorted into four groups based on the resulting dendrogram (Fig. 6).

41) It seems odd not to reference the previous study by the lead author at the end of the first sentence of the Conclusions.

**With previous work (along the west coast) we mean the work by Pritchard and Vaughan (2007). We added the lit. reference to be clearer.**

42) It would appear to be more accurate to say higher 'overall' glacier flow given the heterogeneous response (page 15, line 7) and there is also a problematic use of 'trends' here as well.

**We changed the wording according to the reviewer's suggestion. The use of "trends" was also revised.**

The results are in general in line with findings of the previous studies, however along the west coast higher overall glacier flow was determined and on the eastern side temporal evolution of ice dynamics of 21 glaciers were observed for the first time.

43) It was my understanding from the paper that 'Larsen Inlet, Larsen A' glaciers had area changes assessed since ~1995 (not relative to 1985), page 15, lines 10-11.

**The Larsen A and Prince Gustav Ice Shelf tributaries had area changes assessed since 1995. We added this information to be clearer.**

On the east side all glacier fronts retreated in the study period (relative to 1985, relative to 1995 for former Larsen-A and Prince Gustav Ice Shelf tributaries, see also Section 5.2), with highest retreat rates observed at former tributaries of the Prince Gustav, Larsen Inlet and Larsen A ice shelves.

44) The phrase 'cooling since the mid-2000s' (page 15, line 16) is inconsistent with what Turner and Oliva et al. published. Also, '1960s' seems incorrect. Please check.

**In the previous review, the reviewer suggested to use the term "mid-2000s", considering Turner and Oliva et al.**

1960s refer to Skvarca et al. (1998) (see also Section 1). The authors analyzed temperature measurements at the norther-eastern AP between 1961 and 1996.

45) There would usefully be some discussion of what is and is not possible with the data sets available to date in the last paragraph. It is clear to this reader that even with a very serious effort to understand the variability of this region, there are pretty significant deficiencies in our data sets.

We appreciate the reviewer's comment. In our opinion, most significant is the lack of recent high quality region wide surface elevation change data at the AP, since measurements obtained by Cryosat are strongly limited due to the complex topography and measurements using TanDEM-X data is only available for some sections. We added a statement on this issue. Upcoming sensor probably facilitate the region wide measurement of recent surface elevation, since current estimates have got only partial coverage or have got some serious issues due to the complex topography of the AP.

Figures showing LIMA need not have a copyright symbol, just a credit to the agencies involved.

**We removed the copyright symbol, according to the reviewer's advice.**

Figure 2 has an order of magnitude of velocity difference between panels C and E and I remain concerned that this makes it very difficult to interpret these plots.

We understand the concerns of the reviewer, but we analyzed a large variety of glaciers of different sizes and geometries, therefore the variability of the velocity magnitudes is large. In our opinion it is more useful to adjust the y-axis scale to the individual magnitude in order to better interpret the temporal evolution.

Figure 3 is improved but is still very difficult to read even with much magnification of the pdf files.

**If the editor agrees, we could also upload a high resolution version of the image as a supplement.**

Figure 4, check the caption for a typo 'lest'? Also, please darken the area labels on the Y1 axis to emphasize how much area remains in each sector.

**We appreciate the reviewer's advice and changed the left y-axis (and labels) color to black. We replaced "lest" by "left".**

Figure 5's key should not have dashes and minus signs, please find some other way to show the ranges and consider adding a '+' for the one positive color. Also remove 'regional' before 'sector' in the caption.

**According to the reviewer's advice we revised the keys and caption**

Figure 7, one presumes 'Group N' is 'Group Number' or simpler 'Group'.

**To add "N" to Group was requested by the reviewer in the first review. Well, we removed it from the graph.**

Table 3 shows two categories 'stable' and 'fluctuating' as having the same numeric rating which makes one wonder even further about the cluster analysis. Did I miss discussion of 'fluctuating' in the text? Are these distinctions meaningful given the temporal resolution of the velocity data for many glaciers? Please clarify.

**We decided to use the same numeric rating for "stable" and "fluctuating" glaciers, since the difference is that the variability of "stable" glaciers is less than 0.25 m/d. For both types no clear temporal evolution of the flow speed is obvious. We added a statement in the manuscript to be clearer.**

Glaciers categorized as "stable" showed a temporal variability in flow speeds of less than 0.25 m d-1. Therefore, we used the same rating for the velocity change categories "stable" and "fluctuating" to perform the cluster analysis.

Supplement, page 1, Figure S74 should be S75 for the Drygalski

**Thank you, we corrected it.**

S17, add a space in Arron Icefall's label, consider increasing font sizes for axis labels

According to the reviewer's suggestion we add a space in "Arron Icefall" and increased the font size of the axis labels.

**Referee #2: Jan Wuite Received: 08 August 2017**

**General Comments**

The comments here concern the revised version of the manuscript by Seehaus et al. on changes in glacier dynamics in the northern Antarctic Peninsula since 1985. While I am glad to see that some of the issues seem adequately addressed, some important issues still stand that need to be addressed and clarified. Specifically, those related to methods & interpretation of results.

To illustrate my concerns, I focus here on Drygalski Glacier, because it is the largest glacier in the study region and with a significant reported mass loss, and one with ample velocity data providing a means of comparison. Across-flow velocity profiles of the terminus region are added in the authors' reply and updated manuscript (Fig. S75). While the quality of some of the profiles are up for debate, it shows very clearly that the later velocities (those from 2009/2010 in the plot) are well above pre-collapse values by orders of magnitude. Nevertheless, the authors write (Pg. 12 Ln. 13-15): "Most glaciers (Arron Icefall, Drygalski, LAB2, TPE61, TPE62) decelerated towards pre-collapse values and show almost constant flow speeds in recent years, indicating that the glaciers adjusted to the new boundary conditions." This is, in this example, clearly not the case.

Regarding the quality of some profiles: We did not apply any smoothing or interpolation on our obtained velocity fields. Therefore, some data gaps are present in our profiles. However, profiles with erroneous measurements (noise), large data gaps and partial coverage were sorted out and not used for further analysis (see Section 3.2)

Regarding the flow speed and slow down: We understand the reviewer's concerns. In the respective section we are talking about the median values of flow speed along the profiles, and we are aware that the flow speed at a large section of Drygalski's terminus is still significantly higher than before the disintegration of the Larsen A Ice Shelf, however we decided to use the median values for our analysis and discussion (see also the discussion of the velocity measuring approaches which is now in the supplement). With "decelerated toward pre-collapse values" we meant that the glaciers now slowed down but not necessarily reached pre-collapse values.

We changed the wording of this section to be clearer

Most glaciers (Arron Icefall, Drygalski, LAB2, TPE61, TPE62) strongly decelerated after the initial acceleration and show almost constant flow speeds in recent years, indicating that the glaciers adjusted to the new boundary conditions, albeit significant higher flow speeds can be observe at the central sections of the terminus (see Section S1 and Fig. S149 and S150 in the supplement).

It is, to me, unclear how the velocity profiles of this transect (Fig. S75) translate into the velocities depicted in Fig. 2C and values of Table S1. So, while the cross profiles of 2010 show a significant higher velocity than 1993 and 1995, Fig. 2C shows the median velocity of 1993 to be in the same order of magnitude and for 1995 even higher than those from 2010 and in Table S1 we find for Drygalski Glacier a reported increase in velocity of only ~15% between 1993 and 2010. It seems something is not right. Therefore, to restate my earlier comment, it is certainly not a 'comparable temporal trend' to that from Rott et al. 2014 who report a 280% higher velocity in 2013 compared to 1995, although this statement is still present in the revised version (Pg. 12 Ln 8-10).

We appreciate the reviewer's comment and added a detailed discussion in the supplement on measuring velocities (median along profile vs. point measurement). We concluded to apply the median values in our study. Moreover we carefully revised all data sets which were selected in order to improve the quality. This led to some changes in the obtained results. Now we observe an increase in flow speed between 1993 and 2010 by ~73% at Drygalski Glacier. Considering the different measuring approaches, the results of both studies show comparable temporal trends (acceleration followed by deceleration), but different absolute values as stated in the preceding sentence and also the added discussion in Section S1 (supplement)

**Regarding Fig. S75 (now Fig. S149).**

We are sorry, but somehow not the whole profile was plotted in the previous version of the manuscript. We also added the median values to the graph (right side) in order to better compare the values.

We hope these additions (see also answer to next comment) will help to better interpret the results and explain the differences between point measurements and median values along a profile.

Perhaps there is an issue in the calculation of the median values but it could also be the quality of the velocity data. However, as Drygalski Glacier is a rather wide/large and relatively fast glacier, the quality of velocity data for the many smaller & much slower glaciers is likely worse and in particular for the early period using ERS-1/2 feature tracking (although InSAR should provide better results, was this tested?). How do other cross profiles look like, for instance for the more extreme acceleration and deceleration cases in sector West and East? Do they have the same issue?

I stress this point, as I believe the figures (Fig. 2 & Fig. S1-S74 and the Table) form the core of the manuscript and the basis for subsequent discussions and conclusions. However, since the Drygalski example is the only velocity profile time series given in the manuscript, I think it makes it difficult to convince the reader about the validity of at least some of these plots/numbers. For a paper dealing primarily with glacier velocity we get to see very few velocity maps and/or profiles to get any real confidence in the data.

Following the reviewer's comment and advises and to show the quality of the obtained velocity data, we added velocity maps (for different sensor) of our study site in the supplement. Some more velocity profile time series were also added to the manuscript (for all different velocity categories and for the largest (Drygalski) and smallest (DGC14) catchments as well).

The reviewer is right, regarding ERS-1/2 data. These data sets have got the highest uncertainties (see also last rows of Table S3) as well as the most tracking errors and data gaps (about 50% of the generated results were discharged during the quality checks, as mentioned above we carefully revised the data sets in order to improve the quality of the results). However, we did not apply InSAR to measure surface displacement. Only during ERS mission phases with 1 d and 3 d repetition cycles sufficient coherence can be retained (due to the glacier flow and changes of surface conditions). Just a small amount of the available ERS-data is suitable for this technique, which partially covers the study site. Moreover, the surface displacement can only be measured in range direction using InSAR (except at areas where ascending and descending orbits cross, which further limits the spatial coverage). By using a DEM to project the range displacement in the direction of glacier flow, certain errors can be introduced (e.g. various studies report significant surface elevation changes at the AP. Changes in the surface slopes between the date of the DEM and ERS data impact the projection of the range displacements). Thus, we decided not to use InSAR methods, in order to consistently (temporal and spatially) apply only one technique to calculate surface velocity fields.

**Changes in glacier dynamics in the northern Antarctic Peninsula since 1985**

Thorsten Seehaus1, Alison J. Cook2, Aline B. Silva3, Matthias Braun1

[revised manuscript text omitted]

---

## Author Response (AR3)

**Response to the editor and reviewer comments on**

"Changes in glacier dynamics at the northern Antarctic Peninsula since 1985"

By Thorsten Seehaus et al.

First, we want to thank the editor and reviewer for constructive comments on our manuscript. All comments have been taken into account and a list of answers and undertaken actions is given below. Answers are indented and in bold face type and changes in manuscript are indented in *italic*.

**Comments by editor Etienne Berthier Received: 19 December 2017**

Your revised manuscript and your responses to the second round of comments have now been evaluated by one of the two initial referees. Although he found some clear improvements and clarification, your manuscript still need to be improved before it can be accepted for final publication in TC.

You will find the reviewer assessment below or attached. I also have myself several comments. My main comment is in line with the scepticism expressed by the referee about the use of the median along incomplete transverse velocity profiles as a metric to evaluate glacier velocity fluctuations.

**We understand the editor's concerns. The profiles have an average data coverage of 97% (now mentioned in the revised manuscript; see also answer to comment further down). Therefore, we think that it is justified to average along the profiles.**

I appreciated the huge effort you made to extract the velocity variations at the thickest point along the transverse profile. However this alternative way of qualifying glacier velocity change is currently buried in the supplement although it is important to back up your results. I think that this alternative analysis would deserve to be better included in the manuscript than just by stating "Two approaches to measure and analyze the temporal changes in ice flow of the studied glacier are evaluated and the differences are discussed in the supplement Section S1. The favored measuring approach is explained in the following and its results are used for the subsequent analysis." This does not need to be long I think: A few line of description in the method section and providing the results for the mean velocity change of the three sectors would do the job.

**We agree with the editor and revised the methods and results section accordingly. More details and results of the second approach are now in the manuscript.**

To facilitate the review process (possibly by one of the chief editors due to my 2.5-month absence), please attach to your revised manuscript a cover letter detailing the changes you made in response to all comments.

Best regards,

**Etienne Berthier**
* * *
Abstract: Currently, the sentence "By applying a hierarchical cluster analysis we show that this is associated with the geometric parameters of the individual glacier basins." is a bit enigmatic to the reader. At least authors could list in parenthesis the geometric parameters examined.

**According to the editor's comment, we listed the geometric parameters.**

By applying a hierarchical cluster analysis, we show that this is associated with the geometric parameters of the individual glacier basins (Hypsometric Indexes, maximum surface elevation of the basin, flux gate to catchment size ratio).

2.16 "at the northern part of the peninsula" why not using "northern AP" here

**We changed the wording accordingly.**

2.27 Structure of the sentence starting with "Whereas" does not seem to be OK.

**We appreciate this advice and restructured this sentence.**

Whereas, glacier-wide surface lowering was observed by various author groups (e.g. Berthier et al., 2012; Rott et al., 2014; Scambos et al., 2014) at former ice shelf tributaries along the northeastern AP.

3.7 "-" between "sub" and "regions" (I think)

**Thank you for this advice, we looked it up in a dictionary, and removed the space.**

4.1 delete "an area of"

**We removed "an area of".**

4.25 delete "in the area studied region"

**We deleted "in the area studied".**

5.31 "taking into account the effects on the local incidence angle by the topography" was not clear to me (I am not a SAR specialist as many others TC readers).

**We rephrased this sentence to be clearer to the reader.**

Finally, the displacement fields are transferred from slant range into ground range geometry, taking into account the contortion caused by the topography (topographic effects on the local SAR incidence angle).

6.5 Huber et al. (2017). Tell where they performed their accuracy assessment of the AP-DEM? Northern AP? Whole peninsula?

**Huber et al. (2017) estimated the uncertainty, also based on their experiences with other DEMs. Thus, we conclude their accuracy value was used for the whole AP-DEM. We changed the wording accordingly.**

Since the accuracy varies regionally, Huber et al. 2017 estimated the uncertainty to be  $\pm 50$  m for the AP-DEM, based on their experiences with other DEMs.

7.4 Indicate that this a transverse profile (right?)

**Thank you for this advice. We changed the wording to be clearer**

An across glacier profile is defined (red lines in Fig. 1) close to the terminus of each basin, considering the maximum retreat state of ice front position in the observation period.

7.7 I find the statement "Datasets with partial profile coverage or large data gaps, as well as those with still remaining tracking errors, are rejected." rather vague. What is exactly "partial" (less than 80%, 50% coverage?) and "large data gaps"? Can authors make it clearer and reproductible?

We understand the editor's concerns. The selection of the datasets was done manually (visual inspection) and no certain threshold was applied. However, we analyzed the profile coverage by velocity data and added this information in the manuscript.

The resulting average coverage by velocity measurements along the profiles is 97% and 90% of all extracted profiles have got a data coverage of more than 93%.

Moreover, we restructured the respective paragraph to be clearer.

7.10 The use of the median is a bit problematic (see also the referee statement). Authors should think twice about it in their future studies. Imagine that 40% of the profile had a drastic velocity change and not the rest of the profile. The median would not show any change in velocity where as the ice flow has clearly changed. This is why their alternative method should be included more in the manuscript.

The editor is right, regarding the limitation of median values. However, we also tested to use mean values, but the obtained results were noisier. Small scale outliers (single pixels or just a few pixels, which are remaining tracking errors after filtering) can bias the mean values measured along the profiles, whereas they are filtered out using median values.

"To minimize the impact of potential outliers (still remaining small scale tracking errors), median velocities along the profiles are calculated ...." (Section 3.2)

We did not want to reject the profiles which have some small-scale outliers. We wanted to keep the number of velocity measurements per glacier high, in order to obtain reasonable velocity change time series. Therefore, we decided to use median instead of mean values.

7.14 close parenthesis.

**We closed the parenthesis.**

9.22 Seems that "2" should be deleted before "256"

Please see also Table S1. In total 2256 profile measurements were used in this study. We changed the wording in order to not confuse this value with the number of obtained velocity fields.

...inspected and in total 2256 profile measurements passed the quality check...

11.5 I have a hard time understanding the statement "Moreover, slightly increased recession is also found in the time period (1995-2005, Fig. 4) at sector "East"." in view of the two previous sentences. And then the next sentence deal with ice shelf disintegration but authors discuss here a sector where not ice shelf existed. Overall, the causal link between the successive sentences in this paragraph is unclear. Or did I miss something?

**We understand the editor's concerns and rephrased this paragraph to be clearer.**

Davies et al. (2012) also reported higher retreat rates for most of the glaciers in this sector in the period 1988-2001 than in the period 2001-2009. However, another significant recession is also found at sector "East" after 1995 (Fig. 4). Davies et al. (2012) and Hulbe et al. (2004) supposed that the disintegration of a nearby ice shelf affects the local climate. The air temperatures would rise due to the presence of more ice free water in summers. Thus, the higher retreat rates at sector "East" after 1995 could be indirectly caused by the disintegration of Prince Gustav and Larsen A Ice Shelf.

12.8 why deleting Skvarca et al. (1998) and keeping it two lines later?

Here it was misplaced, since this sentence is about the observed cooling in the 21th century (cannot be observed in 1998) and the next sentence is about the warming since 1970s, which was reported by Skvarca et al. (1998) and Oliva et al. (2017).

12.25. Some textbook reference for surging glacier may be more appropriate (or a generic paper such as Sevestre and Benn, 2015). I do not really see the point at picking Alaska and Karakoram as examples here.

We appreciate the editor's comment. We just picked two regions which are famous for surge-type glaciers. According to the editor's comment we adjusted the wording and reference to two generic paper.

Surge-type glaciers (tidewater as well as land terminating), found in various regions worldwide, show similar behavior (Meier and Post, 1969; Sevestre and Benn, 2015).

13.14. Rott et al. (2017) is just published in TCD and has not passed the peer-review process. Should not be cited.

**We removed the citation.**

13.22 " higher flow speeds" higher than what?

**We refer to pre-ice-shelf-collapse conditions. We added this information.**

....albeit significant higher flow speeds (compared to pre-ice-shelf-collapse conditions) can be observed at the central sections of the terminus ....

14.5 delete "," in "Rott et al., (2014)"

**Deleted**

16.15 "along the west coast higher overall glacier flow" Do the author mean higher flow increase? In the rest of the paper they never compared their absolute flow magnitude to Pritchard & Vaughan (2007) but only compared the % of velocity increase.

**The editor is right, we mean higher flow increase. We rephrased this sentence to be clearer.**

The results are in general in line with findings of the previous studies; however, along the west coast a more accelerated glacier flow is determined and

16.26 "conclude" is very strong in this context. I think "suggest" or "speculate" would better reflect the level of understanding that we have here on these "East" glaciers.

**We replaced "conclude" by "suggest" according to the editor's suggestion.**

17.14 "M.B." instead of "MB"

**Corrected**

Ref to Lai. Volume/page numbers?

**We revised the reference and added volume and page numbers.**

Lai, Z., & Huang, M. 1989. Numerical Classification of Glaciers in China by Means of Glaciological Indices at the Equilibrium Line. In Snow Cover and Glacier Variations. Proceedings of a Symposium held in Baltimore, Maryland May 10-19, 1989, IAHS Publication 183, 103-111

Supplement.

Avoid so much parenthesis in the legend.

**According to the editor's suggestion, we revised the overview legend.**

S1. "along across glacier profiles" is not clear; " using obtained by the second approach" or " This mismatch does not influencing" is not correct. Check the supplement VERY carefully to make sure it is understandable. Proof reading may be useful for this supplementary text.

We are sorry for the errors. We checked the supplement and revised it. Additionally, we send it to a proofreading agency (therefore, most of the "tacked-changes" done by the authors, were removed by the agency and are not visible in the "tracked-changes" version of the revised supplement. We are sorry for this).

**Comment by Referee #2: Jan Wuite Received: 18 December 2017**

**General Comments**

The comments here concern the second revised version of the manuscript by Seehaus et al. on changes in glacier dynamics in the northern Antarctic Peninsula since 1985. The authors have made substantial revisions to the original and revised manuscript in response to the reviewer's comments. These revisions include clarifications, corrections, additional references, as well as an added section and new figures in the supplemental material. The additional scrutiny appears to have led to significant changes in some of the reported velocity changes (individual glaciers & regions) and also to the discovery of errors in some of the figures/tables.

I appreciate the efforts by the authors to clarify their adopted approach, but would like to point out here that I am still rather unconvinced about the exact implementation of the across flow median velocity values in the presented analysis and my earlier worries on several issues still stand. The profiles presented in the manuscript's supplement illustrate to some degree the difficulties and problems that arise. If, as the authors mention in their reply document (1 st reply), the erroneous/incomplete profiles were not used for the analysis there is little reason to include them, but in the way it is written currently it looks like all depicted profiles passed the quality check (Pg. 9 Ln. 12-15). Therefore, I mention here specifically: unrealistic fluctuations appear to occur along some of the profiles (e.g. Fig S155) and some profiles are incomplete (and were apparently not rejected as mentioned in the text – Pg. 6 Ln. 29/Pg. 7 Ln. 1).

Not all profiles with data gaps and remaining tracking errors were rejected, only those with partial coverage, large data gaps or large-scale tracking errors. However, the average profile coverage was kept at a high level of 97%, to facilitate a reasonable averaging along the profiles. In order to suppress the effect of the small-scale tracking errors we calculated median values instead of mean values (see also answer to editor comment above). We are sorry, that it was not clearly descripted. We revised the methods section (also taking in to account the comments of the editor) and hope that it is now clearer.

The results are visually inspected in order to remove unreliable measurements, based on the magnitude and direction of ice flow along the profiles. Datasets with partial profile coverage, large data gaps and large-scale tracking errors are rejected. The resulting average coverage by velocity measurements along the profiles is 97% and 90% of all extracted profiles have got a data coverage of more than 93%. To minimize the impact of potential outliers (still remaining small scale tracking errors), median velocities along the profiles are calculated

A crucial issue, however, is the treatment of velocity in the shear margins, some profiles seem to be cut off rather abruptly, some are smoother and go down to zero velocity, some appear to cover more glaciers and include intermediate (stationary?) areas, some have very high values at the margins (or do not seem to include the margin – e.g. S150). The sensor capabilities and to a certain degree the algorithm settings largely determine how well these margins can be captured. One should therefore be cautious when interpreting extracted values as they could reflect sensor limitations instead of real velocity changes.

We understand the reviewer's concerns. Some of the glacier basins consist of more than one major ice stream, which join towards the terminus. For some cases the intermediate areas have low flow speeds (e.g. Fig. S152 or S155). However, the delineation of the individual sections is not always clearly possible, and its width can vary temporally (as mentioned in Section 2, page 4, line 3-5). Therefore, we decided to merge coalescing glaciers for our analysis and use only one across terminus profile. Moreover, measuring velocities of a glacier system at a fix point can also lead to discrepancies, since the peak position and the shape of the across glacier velocity profile can vary (as discussed in Section S1 supplement). Regarding the velocity data in the margin of the glaciers: The reviewer is right, that sensor capabilities and the tracking algorithm determine the capturing of the flow velocities in the lateral glacier sections. In order to minimize the limitations caused by the tracking algorithm, we used different tracking window sizes (small windows for the slow moving lateral section and larger windows to capture the displacements in the fast-flowing central sections (Section 3.2 page 6, line 9-16). The results were iteratively stacked to obtain the best spatial coverage. However, at some small and narrow glaciers the capturing of the velocity gradient in the margins is still mainly limited by the sensor resolution. Therefore, we present also the velocity profiles of small glaciers in the manuscript (e.g. TPE61 Glacier Fig. S 150, DGC14 Fig. S153, TPE8 Glacier Fig. S156) to illustrate the limitations. Moreover, we added a statement on this issue in the manuscript (Section 4.2 Results)

For small and narrow glaciers, the capturing of the flow velocity gradients in the margins is still mainly limited by the sensor resolution, even applying different tracking window sizes.

That said, in concordance with my main wish in the previous review rounds, at least the methods are now better documented and the inclusion of ice velocity cross profiles in concordance with calculated velocity median values (in the supplement) provide some means of traceability. I likewise welcome the inclusion of additional ice velocity maps for this reason as it shows the source material of some of the numbers and its potential or limitation. Also, the reported ice velocity changes for Drygalski (which I took as a primary example in my previous review) has been revised considerably.

A few more specific comments for consideration:

•The IV profiles are extracted from close to the terminus, but it is not mentioned in the text from which year nor is it very clear from fig 1.

We are sorry, but we do not know to which paragraph/section this comment refers. In Section 3.2 in the main manuscript we state: "A profile is defined (red lines in Fig. 1) close to the terminus of each glacier basin, considering the maximum retreat state of ice front position in the observation period."

In the Supplemental Material Section S1 we link to this section: For the first approach, the flow velocities are extracted along across glacier profiles (see. Fig. 1 in the manuscript) close to the terminus and the median values along the profiles are calculated (see also Section 3.2 in the manuscript).

However, we rephrased this sentence in the Supplemental Material to be clearer:

For the first approach, the flow velocities are extracted along across glacier profiles, defined for each basin close to the terminus and considering the maximum frontal retreat state (see. Fig. 1 in the manuscript), and the median values along the profiles are calculated (see also Section 3.2 in the manuscript).

• In the conclusion it is now mentioned "Upcoming sensor [SIC] probably facilitate the region wide measurement of recent surface elevation, since current estimates have got only partial coverage or have got some issues due to the complex topography of the AP. ". Just a thought, what type of sensor is going to overcome the partial coverage and issues due to complex topography, or is this wishful thinking (in which case 'hopefully' is more apt than 'probably')?

We understand the reviewer's concerns, and "hope" that sensors like e.g. ICESat-2 would overcome the partial coverage and the issues due to complex topography. However, this needs to be validated once it is in operation. Therefore, we replaced "probably" by "hopefully".

• As mentioned above, please clarify whether profiles depicted in supp. material passed the quality check and if they were used or not used in the analysis.

**All profiles shown in Fig. S149-S156 are used in the further analysis. See answer to comment above.**

Grammar

Pg. 2 Ln. 27: ") The"  $\rightarrow$  missing point Pg. 7 Ln. 17&18: on average Pg. 12 Ln. 2: van der Veen Pg. 13 Ln. 8: observeD Sup Mat. Pg. 2 2 nd paragraph: 2256 measurementS Pg. 2 3 rd paragraph: using obtained by the second approach  $\rightarrow$  seems like a word is missing here Pg. 2 3 rd paragraph: This mismatch does not influencE the subsequent Pg. 2 5 th paragraph: 'is little'  $\rightarrow$  is small

Thank you for these advices. We corrected the manuscript and supplemental material accordingly.

[revised manuscript text omitted]

---

## Author Response (AR4)

Dear Olaf Eisen,

Thank you for taking over the final check of our manuscript and for considering it to be accepted for publication in TC. We revised the manuscript and fixed the issues you raised (see answer to your comments below). We hope that you are satisfied with our corrections.

Kind regards

Thorsten Seehaus

Editor Comments (Authors answers are indented in **bold face type**):

2.27
EB: Structure of the sentence starting with "Whereas" does not seem to be OK.
We appreciate this advice and restructured this sentence.
Now reads "Whereas, glacier-wide surface lowering was observed by various author groups (e.g. Berthier et al., 2012; Rott et al., 2014; Scambos et al., 2014) at former ice shelf tributaries along the northeastern AP."

OE: Sorry, the grammar is still wrong! This is a main sentence, no reason to start with „whereas" - delete.

**According to the editor's suggestion we deleted "whereas".**

7.10: OE: This sentence is wrongly written, please rewrite or split.
"The resulting average coverage by velocity measurements along the profiles is 97% and 90% of all extracted profiles have got a data coverage of more than 93%."

**Thank you for this advice, we revised this sentence:**
**The resulting profile coverage by velocity measurements is in average 97% and data coverage of more than 93% is obtained for 90% of all extracted profiles.**

7.16:
"resulting evolutions of the flow speed"
I'd say "evolution of flow speeds" is more consistent, as you use „speeds" later.

**We appreciate this advice and corrected this sentence accordingly.**

17.26: Fix: add „…s will"
Upcoming sensors will hopefully fix …

**Thank you for this advice. We added "..s".**

Figures:
7 and S169: graph „velocity change category": what are the three symbols for between legend and labels? Typesetting error? Please check. Likewise for „Flux gate", where it is however more clear what is meant. Mention these symbols in caption.
Add a) to e) to graphs.

**We used the symbols for the "velocity change categories" already in Fig. 3. In order to facilitate a better comparison and to better illustrate the meaning of the velocity change category ratings (numbers), we added the symbols to the plot. For the Flux gate/ catchment area ratio plot, we also added the pictograms in order to better illustrate the meaning of the numbers. We added information regarding these issues to the caption and added a) to e) to the graphs:**
**Figure 7. Boxplots of cluster analysis input variables (Sector "West") for each group. Whiskers extend to the most extreme data points. Panel (b): The symbols used for the velocity change categories (see Table 3) are the same as in Fig. 3. Panel (d): The pictograms illustrate the catchment shape (see Section 3.3).**

S149: At least mention in the caption of the first of these figures what the date in the legend stands for.

**According to the editor's suggestion, we revised the figure captions (Fig. S149-S156):**
**Surface velocity across the terminus of XXX Glacier (left) and median values of each profile (right). Dashed line: maximum ice thickness of across glacier profile; Dates in legend: mean dates of SAR acquisitions used to calculate the surface velocity fields.**